# Towards Estimation of Seasonal Water Dynamics of Winter Wheat from Ground-Based L-Band Radiometry: A Concept Study

Thomas Jagdhuber[1,2], François Jonard[3,4], Anke Fluhrer[1,2], David Chaparro[5], Martin J. Baur[6], Thomas Meyer[4] and María Piles[7]

[1] German Aerospace Center, Microwaves and Radar Institute, Münchener Strasse 20, 82234 Wessling, Germany
[2] Institute of Geography, University of Augsburg, Alter Postweg 118, 86159 Augsburg, Germany
[3] Geomatics Unit, Université de Liège (ULiege), Allée du Six Août 19, 4000 Liège, Belgium
[4] Agrosphere (IBG-3), Institute of Bio- and Geosciences, Forschungszentrum Jülich GmbH, 52428 Jülich, Germany
[5] Universitat Politècnica de Catalunya, CommSensLab & IEEC/UPC, Jordi Girona 1-3, 08034 Barcelona, Spain
[6] University of Cambridge, Department of Geography, Philippa Fawcett Dr, CB3 0AS, Cambridge, U.K.
[7] Image Processing Lab, Universitat de València, 46980 Valencia, Spain

*Correspondence to:* Thomas Jagdhuber (Thomas.Jagdhuber@dlr.de), Tel.: +49-8153-28-2329

**Abstract.** The vegetation optical depth ($VOD$) variable contains information on plant water content and biomass. It can be estimated alongside soil moisture from currently operating satellite radiometer missions, such as SMOS (ESA) and SMAP (NASA). The estimation of water fluxes, such as plant water uptake ($PWU$) and transpiration rate ($TR$), from these earth system parameters ($VOD$, soil moisture) requires assessing water potential gradients and flow resistances in the soil, the vegetation and the atmosphere. Yet water flux estimation remains an elusive challenge especially on a global scale. In this concept study, we conduct a field-scale experiment to test mechanistic models for the estimation of seasonal water fluxes ($PWU$ and $TR$) of a winter wheat stand using measurements of soil moisture, $VOD$, and relative air humidity ($RH$) in a controlled environment. We utilize microwave L-band observations from a tower-based radiometer to estimate $VOD$ of a wheat stand during the 2017 growing season at the Selhausen test site in Germany. From $VOD$, we first extract the gravimetric moisture of vegetation and then determine the relative water content ($RWC$) and vegetation water potential ($VWP$) of the wheat field. Although the relative water content could be directly estimated from $VOD$, our results indicate this may be challenging for the phenological phases, when rapid biomass and plant structure development take place within the wheat canopy. We estimate water uptake from the soil to the wheat plants from the difference between the soil and vegetation potentials divided by the flow resistance from soil into wheat plants. The transpiration rate from the wheat plants into the atmosphere was obtained from the difference between the vegetation and atmosphere water potentials divided by the flow resistances from plants to the atmosphere. For this, the required soil matric potential ($SMP$), the vapor pressure deficit ($VPD$) and the flow resistances were obtained from on-site observations of soil, plant

and atmosphere together with simple mechanistic models. This pathfinder study shows that the L-band microwave

radiation contains valuable information on vegetation water status that enables the estimation of water dynamics

(up to fluxes) from the soil via wheat plants into the atmosphere, when combined with additional information of

soil and atmosphere water content. Still, assumptions have to be made when estimating the vegetation water

potential from relative water content as well as the water flow resistances between soil, wheat plants and

atmosphere. Moreover, direct validation of water flux estimates for the assessment of their absolute accuracy could

not be performed due to a lack of in situ $PWU$ and $TR$ measurements. Nonetheless, our estimates of water status,

potentials and fluxes show the expected temporal dynamics, known from literature, and intercompare reasonably

well in absolute terms with independent $TR$ estimates of the NASA ECOSTRESS mission, which relies on a

Priestly-Taylor type of retrieval model. Our findings support that passive microwave remote sensing techniques

qualify for the estimation of vegetation water dynamics next to traditionally measured stand-scale or plot-scale

techniques. They might shed light on future capabilities of monitoring water dynamics in the soil-plant-atmosphere

system including wide-area, remote sensing-based earth observation data.

**1 Introduction**

The monitoring of water dynamics between soil, vegetation and atmosphere and therefore the water availability

for plants requires a system-driven and holistic approach integrating these three water storage compartments. The

soil-plant-atmosphere system (SPAS) represents such an approach, linking the water, energy and carbon cycles

(Reichardt and Timm, 2014; Manzoni et al., 2013a; Ritchie, 1981; Cowan, 1965). SPAS is understood as a physical

continuum, in which water dynamics occur as interdependent transfer processes between the three system

compartments (Nobel, 2020). Gardner (1960; 1965) was among the first to study the system holistically and to

point out that the water transport in the system follows the direction of decreasing energy leading to the concept of

water potential in soil, plant and atmosphere (Slayter & Taylor, 1960). A gradient in water potential induces a flow,

for instance, from soil via roots and plant parts into the atmosphere mediated by the resistance of the traversed

system compartments (Oosterhuis and Walker, 1987; Choudhury and Idso, 1985; Yan and Jong, 1971).

Water potential ($WP$) refers to the potential energy of water and is a key variable in plant hydrology (Reichardt

and Timm, 2014). Since plants are the central component of the SPAS, the decreasing gradient in $WP$ from soils

to the atmosphere drives the movement of water through plants (Elfving et al., 1972; Pearcy et al., 2012). Thus,

the vegetation water potential ($VWP$) changes according to the water availability in soils and the uptake capabilities

of the plant root and xylem systems, according to the regulation of transpiration through the stomata in leaves, and according to the vapor pressure deficit in the atmosphere (Lambers et al., 2008; Jonard et al., 2020).

The water transport through the different compartments of the SPAS is coordinated by adaption of resistances, in a way that water use efficiency is maximized by the plants, avoiding conditions with potentially plant-damaging water potentials (Tyree and Zimmermann, 2013). According to Manzoni et al. (2014), the coordinated behavior is an outcome of plant adaption to their habitat by optimizing water use and gain in carbon assimilation for plant growth.

Under drought conditions, $VWP$ might change due to more negative water potentials in soil and atmosphere (Choat et al., 2018; Johnson et al., 2018). Under these conditions, $VWP$ is regulated by stomatal closure and the general plant hydraulic strategy (isohydric to anisohydric behavior) and thereby mechanistically relates water, energy and carbon cycles (Lambers et al., 2008; Martinez-Vilalta and Garcia-Forner, 2017). Estimating $VWP$ from satellites would be interesting for regional and global ecological studies addressing drought impacts,

especially on forest and agricultural ecosystems (e.g. Feldman et al., 2020). Such retrievals could inform or update global vegetation, earth system, or atmospheric boundary layer models (Bonan et al., 2014; Matheny et al., 2017; Moment et al., 2017; Momen and Bou-Zeid, 2017).

Microwave remote sensing retrieval approaches up to satellite observation missions (e.g. SMOS (ESA) or SMAP (NASA)) often deal with the estimation of earth system parameters (like soil moisture) rather than with the

assessment of dynamic flow processes, like water uptake or transpiration of plants (Entekhabi et al., 2010; Kerr et al., 2010; Portal et al., 2020). An exception in terms of estimation of water dynamics is the GRACE mission (Sadeghi et al., 2020a), assessing total water storage change, and especially its combination with the SMOS and SMAP missions (Sadeghi et al., 2020b). The estimation of water dynamics from remote sensing requires assessing the potential (suction tension) gradients of water rather than the (storage) filling status of soils or plants with water.

Passive microwave remote sensing techniques should be capable of obtaining $VWP$-estimates, provided that plant hydraulic trait information is available (Konings et al., 2019). To do so, the vegetation optical depth ($VOD$) parameter, that measures the degree of attenuation of microwaves as they pass through vegetation, needs to be first extracted from the radiometer observations by model-based parameter retrievals. Afterwards the biomass and water contributions to the $VOD$ parameter need to be disentangled (Martinez-Vilalta et al., 2019). As proposed in Bonan

et al. (2014), the relative water content ($RWC$) can be a valid metric for plant water status and can be utilized to estimate $VWP$. Konings et al. (2019) explained how microwave remote sensing can be applied for monitoring of plant parameters, like $RWC$. Furthermore, Rao et al. (2019) showed that $VOD$ from X-band satellite radiometry

can be converted into $RWC$-estimates with the caveat that both, water and biomass, influence the $VOD$ especially in seasonally growing agricultural species, like winter wheat. Therefore, only periods with constant biomass should be evaluated in order to isolate the water component directly from $VOD$. Fink et al. (2018) found a way to extract the gravimetric moisture of vegetation ($m_g$) from the $VOD$-signal. $m_g$ can be converted into $RWC$ and is not affected by biomass dynamics. The $m_g$-estimation methodology of Fink et al. (2018) was further developed and validated at the field-scale in Meyer et al. (2019).

For the present study, we estimate water potential and water flux along SPAS for the 2017 growing season of a winter wheat field based on the dataset of Meyer et al. (2019). A wheat monoculture has the advantage that growth stages between individual plants are nearly completely synchronized and the canopy develops very homogenously. The benefit here is that measurements of individual plants are very likely representative for all other plants and can be scaled to the whole canopy. Most notably, a main motivation for analyzing wheat comes from its importance for food production being one of the major crop types cultivated around the globe.

The winter wheat field was monitored with the ground-based L-band radiometer instrument ELBARA-II. Simultaneously, in situ measurements for key plant (e.g. $m_g$, height, biomass, leaf area index ($LAI$), and phenology), soil (e.g. relative permittivity and moisture) and atmosphere (e.g. air temperature, wind speed, net radiation and relative humidity) properties were recorded directly at or around the test field.

The objective of this concept study is to research the feasibility of estimating water potentials as well as water uptake from soil and transpiration rates into the atmosphere for a winter wheat field from ground-based L-band radiometry and on-site measurements of soil and atmosphere. With our research we aim to shed light on whether the combination of L-band radiometry and a confined set of on-site measurements are sufficient for the derivation of reasonable water potential dynamics and water flux rates in the SPAS of a winter wheat field.

**2 Test site and experimental data**

The research study was carried out at the Selhausen remote sensing field laboratory, Germany (Jonard et al., 2015; Jonard et al., 2018). In 2017 winter wheat (*Triticum aestivum L.*) was grown in the crop rotation of the farmers at the field laboratory. Key developmental stages of winter wheat are published by Bruns & Croy (1983) and indicate that this agricultural crop has a distinct phenological cycle in the yearly growing period. Detailed information on distribution, botany, growth and physiology of winter wheat are presented in Curtis et al. (2002). The winter wheat at Selhausen grew well without irrigation or fertilization. It was also not affected by plant diseases.

The experimental setup consists of a 12 [$m$] x 20 [$m$] plot covered by a mesh reflector (metal grid) on the ground with winter wheat plants growing through it. We performed L-band (1.4 [$GHz$]) passive microwave measurements using an ELBARA-II radiometer of Forschungszentrum Jülich (FZJ) fixed at 4 [$m$] height. ELBARA-II features a dual-mode conical horn antenna with an absolute accuracy of 1 [$°K$] and relative sensitivity of < 0.1 [$°K$] (Meyer et al., 2018). We repeated the radiometer measurement twice a week at incidence angles between 40° and 60° in 5° increments.

Solely the vegetation microwave radiation is measured using a mesh reflector on the ground and the radiation from the soil is blocked by the reflector (Jonard et al., 2015). We conducted in situ measurements of soil texture (silt loam with 13 [%] sand, 70 [%] silt, and 17 [%] clay), soil permittivity and soil temperature every 15 [$min$] (and every 30 [$min$]) using 15 capacitance sensors installed at 5 [$cm$] (and using 30 time-domain reflectometry (TDR) sensors installed at 30 [$cm$]) depth in direct vicinity of the radiometer footprint within the field. Soil permittivity measurements can be converted to soil moisture $\theta$ according to the well-established dielectric mixing model of Topp et al., (1980). Precipitation, net solar radiation, air temperature, wind speed and relative humidity data are available in 10 [min]-resolution (except for wind speed at 30 [min]-resolution) from two TERENO climate station, located next to the field laboratory (6.449° E, 50.869° N) and on a neighboring field (6.447° E, 50.865° N). In order to be less dependent on in situ measurements, L-band radiometer-derived soil moisture data could also have been used from a non-meshed area of the winter wheat field instead of the in situ soil permittivity measurements. However, the in situ data have a significantly higher temporal resolution than the radiometer observations. But similar to the soil permittivity, these data could be derived from remote sensing-based approaches in a future, more in situ-independent and larger-scale research study.

We measured vegetation height [$m$], leaf area index ($LAI$) [$m²/m²$], above ground biomass ($AGB$) [$kg/m²$], and vegetation water content ($VWC$) [$kg/m²$] destructively every week around the radiometer measurements for comparison and validation. Following Meyer et al. (2019), we used in situ measured $VWC$ to obtain in situ $m_g$ by calculating first the dry matter fraction ($m_d$) as defined by Mätzler (1994), i.e., $m_d$= dry mass/ fresh mass, and subtracting it afterwards from one (i.e., $m_g$= 1 - $m_d$).

All vegetation-related measurements of the growing season in 2017 and the different phenological phases are presented in Figure 1 and Table 1 showing a distinct correlation between $VWC$ and $LAI$ (R=0.94). For a detailed description of the trends and dynamics of these on-site measurements as well as a full sketch of the experimental setup, the reader is referred to Meyer et al. (2018) and Meyer et al. (2019).

**Table 1: Overview of growth stages of the winter wheat between 10th of April (DOY 100) and 14th of August 2017 (DOY 226) and their corresponding phenological phase (after BBCH (*Biologische Bundesanstalt, Bundessortenamt und CHemische Industrie*) code) (Meyer et al., 2018).**

| DOY 2017 | Growing stage | BBCH code |
|---|---|---|
| 100 | Tillering | 26 |
| 108 - 122 | Stable stem elongation | 30 |
| 128 - 142 | Further stem elongation (increased plant growth rate) | 35 |
| 142 - 149 | 50 % of inflorescence visible | 55 |
| 157 | Beginning of flowering | 61 |
| 163 | Grain development started | 71 |
| 180 | Grain fully developed and start of ripening process | 77 and 83 - 89 |
| 190 - 226 | Early senescence until late senescence | 92 -99 |

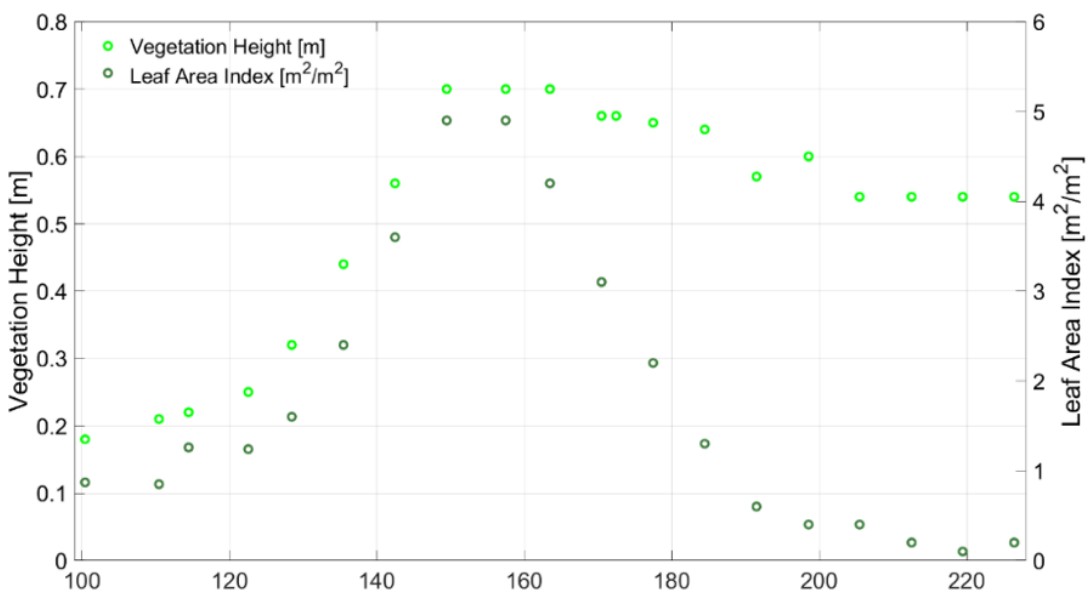

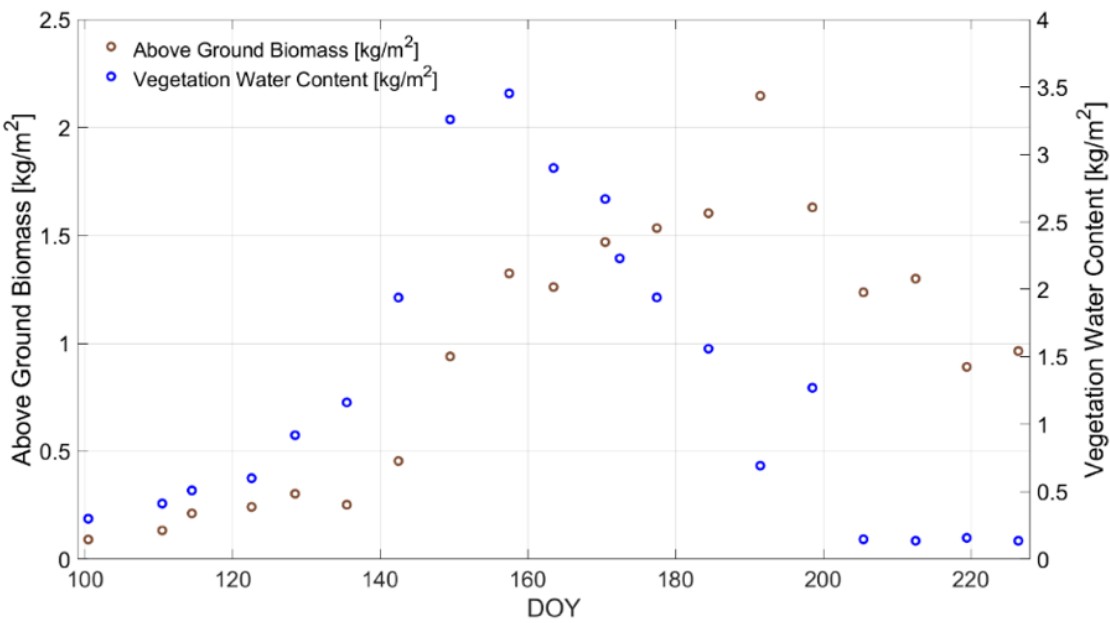

**Figure 1: In situ measurements of the wheat field along growing season for days of year (DOY) in 2017 at the Selhausen field laboratory, Germany: Vegetation height [*m*] & leaf area index (*LAI*) [*m²/m²*] (top); above ground biomass (*AGB*) [*kg/m²*] & vegetation water content (*VWC*) [*kg/m²*] (bottom) (Meyer et al., 2018).**

For later comparison of the transpiration estimates in Selhausen with benchmark information, we apply data from

160 the NASA ECOsystem Spaceborne Thermal Radiometer Experiment on Space Station (ECOSTRESS) mission (https://ecostress.jpl.nasa.gov/). The ECOSTRESS mission was launched in 2018 and carries on board a thermal infrared radiometer with high spatial resolution (70 m). The radiometer measurements are used to estimate the earth's surface temperatures and to derive evapotranspiration (*ET*). Two ECOSTRESS L3 products provide ET data: the L3_ET_PT-JPL product, based on L2 information, and the L3_ET_ALEXI product, based on the

165 ALEXI/disALEXI algorithm. In our case, we focus on the L3_ET_PT-JPL product, which also uses MOderate resolution Imaging Spectroradiometer (MODIS) data as auxiliary information and leads to partitioning of *ET* into canopy transpiration and soil evaporation (Halverson et al., 2019).

We compare our transpiration estimates in Selhausen (2017) with ECOSTRESS L3_ET_PT-JPL transpiration data for years 2019, 2020 and 2021, for DOY between 100 and 200. Note that no ECOSTRESS data is available for

this DOY period in 2018, and that the number of data available varies for each year and is irregularly distributed, which hinders obtaining regularly sampled time series. The L3_ET_PT-JPL dataset provides instantaneous information (i.e., transpiration at MODIS time over pass: 10.30 a.m. local time, approximately) and daily information (i.e., a daily integral of *ET* built from a sinusoidal model mimicking radiation intensity; Halverson et al., 2019). Uncertainty information is available for the instantaneous *ET* dataset: when uncertainty was higher

than 50% of the transpiration value, we excluded the respective date. In addition, note that nightly overpasses of

the ECOSTRESS sensor are excluded. Moreover, transpiration estimates equal to zero are screened out due to unlikely (probably wrong) 100% vs. 0% partition of $ET$ (between soil evaporation and canopy transpiration respectively). Finally, at some days two or more transpiration estimates are available. Thus, we compute the median for each day to ensure we finally have one transpiration estimate per day. Transpiration estimates of L3_ET_PT-JPL are in units of [$W/m^2$]. We transform these estimates into units of [$mm/s$] as follows (Halverson et al., 2019): $TR\left[\frac{mm}{s}\right] = TR\left[\frac{W}{m^2}\right] \cdot 10^{-6} \cdot LH$ where $LH$ is latent heat and is computed with $LH = 2.501 - 0.002361 \cdot T$. $T$ is temperature in degree Celsius. $T$ is obtained in 30-minutes timesteps from a meteorological station next to the Selhausen test field. For the instantaneous transpiration data, $T$ at 10:30 a.m. is used. For the daily transpiration data, $T$ is averaged between 7 a.m. and 8 p.m..

**3 Methodology of water dynamics estimation**

The methodology for water dynamics estimation - water uptake from soil to wheat vegetation ($PWU$) & transpiration from wheat vegetation to atmosphere ($TR$) - is conceptualized in the workflow of Figure 2. First, the water status of soil ($\theta$), vegetation ($m_g$; cf. section 2) and atmosphere ($RH$) need to be known from remote sensing estimates or on-site measurements. From the water status, the water potentials for the three environmental compartments (soil, vegetation & atmosphere) are calculated (sections 3.1 to 3.3) and then $SMP$ and $VWP$ are used to retrieve the water uptake from the soil into the wheat vegetation (section 3.4). In addition, the atmospheric water potential, usually expressed by the vapor pressure deficit ($VPD$), is applied together with the $VWP$ to calculate the transpiration rate of the wheat vegetation into the atmosphere (section 3.5).

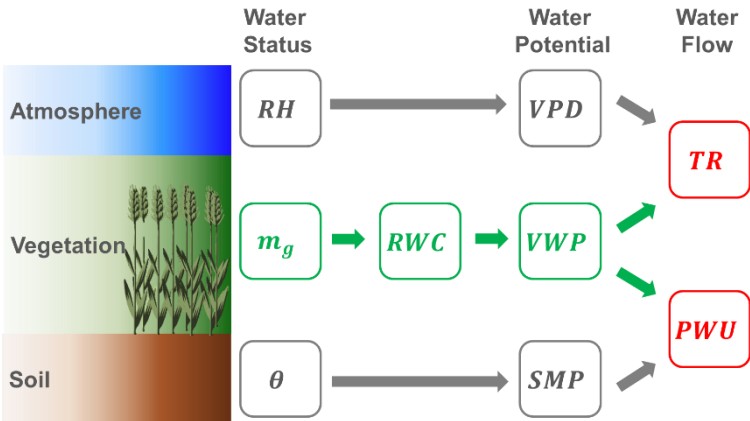

**Figure 2: Processing workflow for estimation of soil, vegetation and atmosphere water potentials ($SMP$= soil matric potential, $RWC$ = relative water content of vegetation, $VWP$ = vegetation water potential, $VPD$ = vapor pressure deficit) and water fluxes ($PWU$ = plant water uptake, $TR$ = transpiration rate) from storage variables ($\theta$ = soil moisture, $m_g$ = vegetation water content (gravimetric), $RH$ = relative air humidity); Green variables are derived from radiometer observations, while gray ones are calculated from in situ measurements; Red variables are derived jointly from radiometer and in situ observations.**

### 3.1 Soil matric potential (*SMP*) from soil moisture (*θ*)

The fundamental force describing the overall state and movement of water within the soil is energy: kinetic or potential (Hillel, 1980). The kinetic energy is assumed to be negligible in these agricultural (wheat-covered) soils around the test site due to the overall slow movement of soil water in fields with no to moderate slopes (Hillel, 1980; Shukla, 2014). The potential energy describes the movement of water in soils and the water retention forces against percolation towards ground water level. Hence, the rate of decrease in potential energy with distance is actually the driving force which causes water to flow within soils (Hillel, 1980). The difference in energy states between the soil water and pure free water (i.e. reference potential) is defined as the total soil water potential. In unsaturated soils various forces, such as capillary or adsorption, act on soil water, which causes soil water potential to be lower than that of pure water in reference conditions (known as suction tension).

Due to the different forces influencing the soil water, the total soil water potential is mainly the sum of pressure, gravitational and osmotic potentials (Hillel, 1980; Shukla, 2014). Since the pressure potential is defined as the water potential resulting from capillary and adsorptive forces acting on the soil matrix, it is generally called the *SMP* or matric suction (Hillel, 1980; Shukla, 2014); it can be expressed in energy per unit mass [*J/kg*], in energy per unit volume (pressure: [*bar*] or [*Pa*]), or energy per unit weight (hydraulic head: [*cm*] or [*pF*]) (Hillel, 1980; Shukla, 2014, Ward and Robinson, 2014). Water potential values in this study will be presented in units of pressure [Pa].

The *SMP* is dependent on *θ* and vice versa. The relationship between both parameters is described by the soil water retention curve. Water retention subsumes all mechanisms and processes related to changes of soil moisture and its energy state (Gupta and Wang, 2006). The shape of the soil water retention curve is dependent upon various soil characteristics (e.g., texture, and particularly clay fraction), as well as on the current and previous states of *θ* (Hillel, 1980; Ward and Robinson, 2014).

When investigating plant growth, two common values of *SMP* are of major interest: the permanent wilting point and the field capacity. The wilting point (around *SMP* of -1.5 [*MPa*]) is defined as the minimum soil water content at which most crop plants can still extract water from the soil (Nobel, 2020, p. 542). The field capacity, reported for instance at -0.01 [*MPa*] in Ward and Robinson (2014) or at -0.033 [*MPa*] in Gupta and Wang (2006), corresponds to the amount of remaining water after a saturated soil drained under gravity for one to two days following a precipitation event. Exact values for each soil are dependent on the individual soil characteristics. The difference between field capacity and wilting point is regarded as available water that can be taken up by plants

(Ward and Robinson, 2014; Gupta and Wang, 2006). Besides tensiometer or thermocouple psychrometer, which can directly measure $SMP$ (Hillel, 1980; Gupta and Wang, 2006), several soil water retention models exist to estimate $SMP$ from $\theta$. Widely used ones are from Brooks and Corey (1964), Campbell (1974), or Van Genuchten (1980). In this study we apply the Campbell model to estimate $SMP$ from $\theta$ measurements:

$$SMP = \left( SMP_s \ast \left( \frac{\theta}{n} \right)^{-b} \right), \tag{1}$$

with $SMP_s$ as matric potential at field capacity (saturated suction), $n$ representing soil porosity, and $b$ as empirically determined constant characterizing the pore-size distribution of the soil (Margulis, 2017; Campbell, 1974). The values for $SMP_s$ and $b$ for (1) are provided by Clapp and Hornberger (1978), where representative values for hydraulic parameters are presented for various soil textures. The values for the soil type silty loam at

Selhausen are $SMP_s$ = -0.786 [$m$], $n = \theta_s \cdot f_s$ (including soil moisture content at field capacity $\theta_s$= 0.485 [-] from Clapp and Hornberger (1978) and silt fraction from soil surveys at the test site $f_s$=0.7 [-]) and $b$ = 5.3 [-]. Values for $\theta$ originate from *in situ* relative permittivity measurements from 5 [$cm$] and 30 [$cm$] soil depth (cf. section 2).

### 3.2 Relative water content ($RWC$) and vegetation water potential ($VWP$) from vegetation moisture ($m_g$)

Meyer et al. (2019) first estimated the $m_g$ of the winter wheat field [$kg/kg$] from ground-based L-band radiometer data, acquired between 9 am and 2 pm at 40° incidence angle. In this process, they retrieved $VOD$ via radiative transfer model inversion from the V-polarized brightness temperature measurements of ELBARA-II. Afterwards $m_g$ was estimated from $VOD$ by inversion of the forward model proposed by Schmugge & Jackson (1992) (Meyer et al., 2019; Ulaby and El-Rayes, 1987). They assumed vertical stalks as mainly affecting plant component of winter wheat for L-band emission and within radiometer-based $m_g$-calculus for subsequent analysis in this study.

Full estimation details are provided in Meyer et al., 2019.

$m_g$ is a metric of kilogram water per kilogram wet biomass and can be converted into a change metric, called relative water content $RWC$ [%] by putting boundaries on upmost (maximum $m_{g_{max}}$) and lowest (minimum $m_{g_{min}}$) $m_g$ (Pearcy et al., 2012; Easmus et al., 2016; Smart and Bingham, 1974):

$$RWC = \frac{m_g - m_{g_{min}}}{m_{g_{max}} - m_{g_{min}}} \cdot 100, \tag{2}$$

As these boundary conditions are found for this study along the measured growing season in 2017 within the winter wheat field, the $RWC$ serves as a relative metric referring to the water dynamics of the recorded season ($RWC_{Season}$). Since $m_g$-estimates are obtained only once on a measuring day, it limits the chance of capturing the true minimum and maximum of the 2017 growing season. However, the $m_{g_{min}}$ is found in the senescence phase where the water content drops to a minimum (see low and constant level of $VWC$ in Figure 1 for the senescence phase). The detection of the maximum $m_{g_{max}}$ of the season is even more challenging due to the temporally sparse measurements. Therefore, the maximum of the $m_g$-timeseries $m_{g_{max}}$ is used. Note that $RWC_{Season}$ by definition is not representative of water dynamics on shorter time scales than seasons, like on weekly or even diurnal scales or for single phenological phases (Passioura, 1982).

For the next step, Zweifel et al. (2000; 2001) described a semi-empirical model linking the $RWC$ [%] to its $VWP$ [$MPa$]:

$$VWP = \frac{VWP_{min}}{e^{\frac{-k_1 + RWC}{k_2}} + 1},$$ (3)

where $k_1$ and $k_2$ are empirical parameters representing the inflection point of the function, and the rate of change between $RWC$ and $VWP$, respectively. $VWP_{min}$ [$MPa$] is the minimum of $VWP$ assumed for the relationship. All three parameters are plant type specific and need plant-specific adaption, since the parameterization of Zweifel et al. (2000; 2001) was done for trees. In this study we adapted the parameterization for winter wheat with $VWP_{min}$ of -2.5 [$MPa$] (Frank et al., 1973; Gupta et al., 1989; Kameli and Lösel, 1993; Rascio et al., 1994, Siddique et al., 2000). Figure 3 illustrates the $RWC$-$VWP$ relationship for slow ($k_1$=55; $k_2$=10), intermediate ($k_1$=68; $k_2$=7.5) and rapid ($k_1$=81; $k_2$=5) change dynamics according to literature studies (Turner and Long, 1980; Turner, 1988; Zweifel et al., 2001; Pearcy et al., 2012; Konings et al., 2019). The sigmoidal dynamics in Figure 3 should cover the potential variation of dynamics for wheat, as understood from literature. It is in no means exhaustive or fully precise.

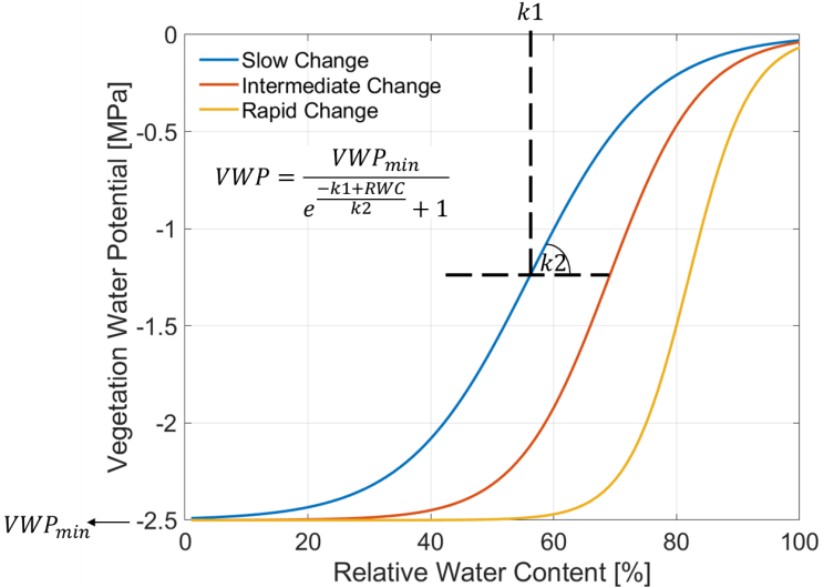

**Figure 3: Modelled relation between relative water content ($RWC$) [%] and vegetation water potential ($VWP$) [$MPa$] adapted for winter wheat and assuming different rates from slow (blue color) to intermediate (red color) until rapid (orange color) change. The inset equation, adopted from Zweifel and Häsler (2000) and Zweifel et al. (2001), integrates a minimum $VWP$ ($VWP_{min}$) and specifies an inflection point ($k_1$) and a rate ($k_2$) of the change dynamic between $RWC$ and $VWP$. $k_1$ and $k_2$ are also indicated schematically with dashed black lines.**

### 3.3 Atmosphere water potential (vapor pressure deficit - $VPD$) from relative humidity ($RH$) and air temperature ($T_{Air}$)

In unsaturated air conditions the atmosphere shows a deficit of water vapor indicating the water potential of the atmosphere. Atmospheric vapor pressure deficit ($VPD$) is defined as the difference between the actual water vapor pressure and the saturation water vapor pressure at a particular temperature. It can be calculated as follows (Reichardt and Timm, 2014; Castellvi et al., 1996):

$$VPD = P_{sa}(1 - RH), \tag{4}$$

Including the relative humidity of the air $RH$ [-] and saturation water vapor pressure of the air $P_{sa} = 0.61094 \cdot \exp(17.625 \cdot T_{Air}/(T_{Air} + 243.04))$ [$kPa$], where $T_{Air}$ represents the air temperature [$°C$] (Alduchov and Eskridge, 1996). In this study $RH$ and $T_{Air}$ are measured at $2$ [$m$] height above ground at the on-site meteorological station. Time series of $VPD$ and its input variables ($RH$, $T_{Air}$) for the period of study are presented in Figure 4.

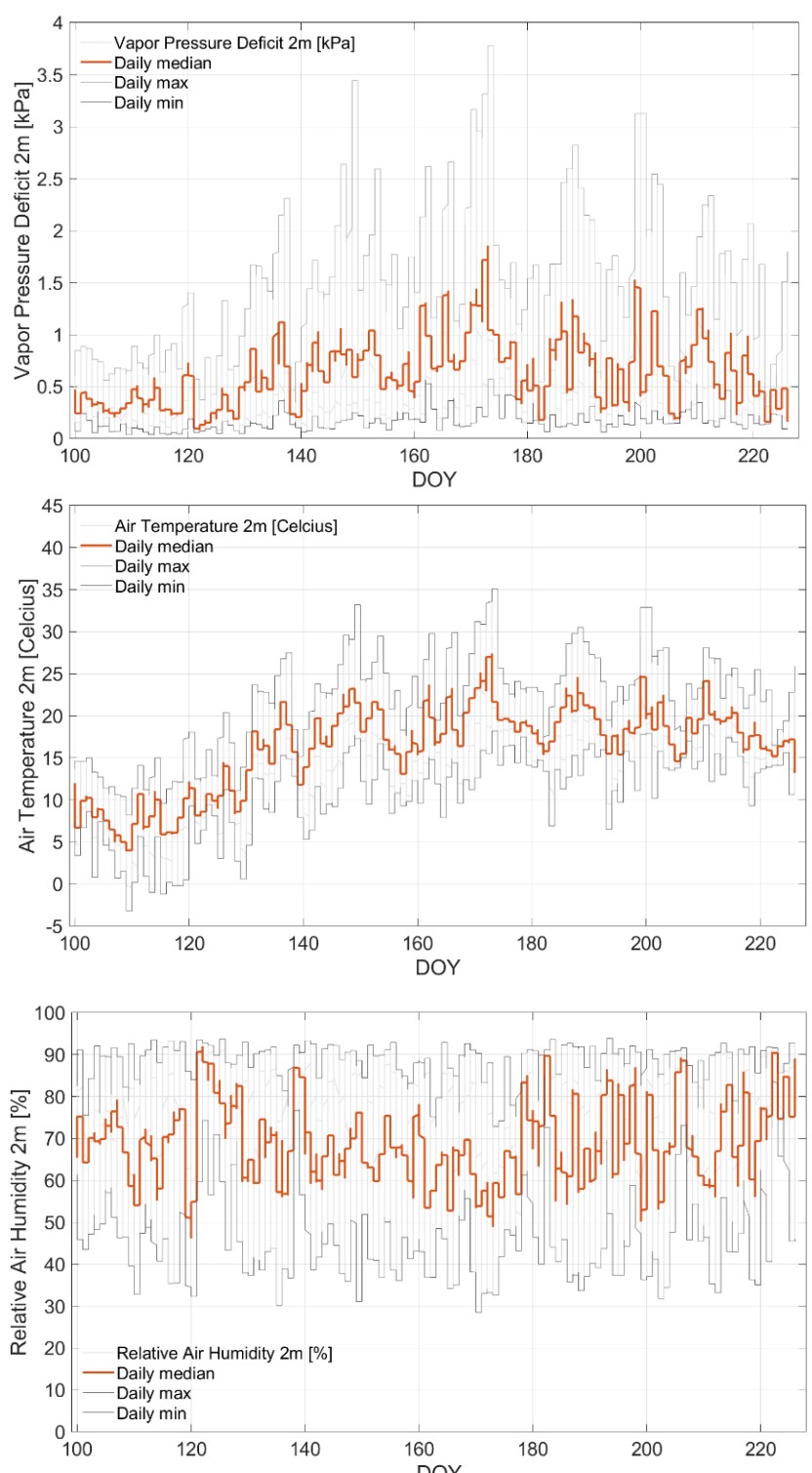

300

**Figure 4: Calculated vapor pressure deficit ($VPD$ [$kPa$]) at 2 [$m$] height (top plot) from measurements of air temperature ([$T$ [°$K$]) (middle plot) and relative humidity ($RH$ [%]) (bottom plot) using Equation (4) in the growing period of 2017 along days of year (DOY) in 2017 at the winter wheat field in Selhausen, Germany. Colored and gray curves indicate daily median, minimum and maximum of measurements and estimates.**

305

### 3.4 Water uptake ($PWU$) from soil into the wheat plants

The water uptake process follows from hydraulic potential gradients and flow resistances in the SPAS. The principle of potential difference by flow resistance is motivated from the field of electricity by Ohm's law. Van den Honert (1948) was one of the first who showed this connecting concept (Cowan, 1965, Monteith and Unsworth, 2013; Nobel, 2020).

The $PWU$ [$mm/s$] from the soil into the winter wheat plant can be defined as the potential difference (converted to [$mm$]) between the soil ($SMP$) and the vegetation ($VWP$) divided by the resistance (e.g. in the rhizosphere, roots and xylem along SPAS (Van den Honert, 1948; Wallace, 1978; Wallace and Biscoe, 1983)):

$$PWU = \frac{SMP - VWP}{R_{RX} + R_S}, \qquad (5)$$

where $R_{RX}$ [$s$] is the resistance to water flow in roots and xylem of the wheat plants (Lynn & Carson, 1990) and $R_S$ [$s$] is the resistance to water uptake from the soil into the wheat roots. Since $R_S$ and $R_{RX}$ cannot be measured in situ at field scale along the growing season (Wallace, 1978; Wallace and Biscoe, 1983; Ruggiero et al., 2007), all resistances in this study are effective values with no claim on absolute accuracy.

We adopted the approach of Feddes and Rijtema (1972) estimating the $R_S$ from soil hydraulic conductivity [$m/s$], based on the Campbell model (Campbell, 1974; Choudhury and Idso, 1985; Lynn and Carlson, 1990; Dingman, 2015; Meyer et al., 2018). We used soil moisture [$m^3/m^3$] (converted from soil permittivity) and soil parameters for silty loam from Clapp and Hornberger (1978) and Steenpass et al. (2010), and a rooting depth of one meter according to literature (Fan et al., 2016). Figure 5 shows estimates of soil resistance [$s$] from Feddes and Rijtema (1972) based on soil hydraulic conductivity [$m/s$] at 5 [$cm$] and 30 [$cm$] soil depth.

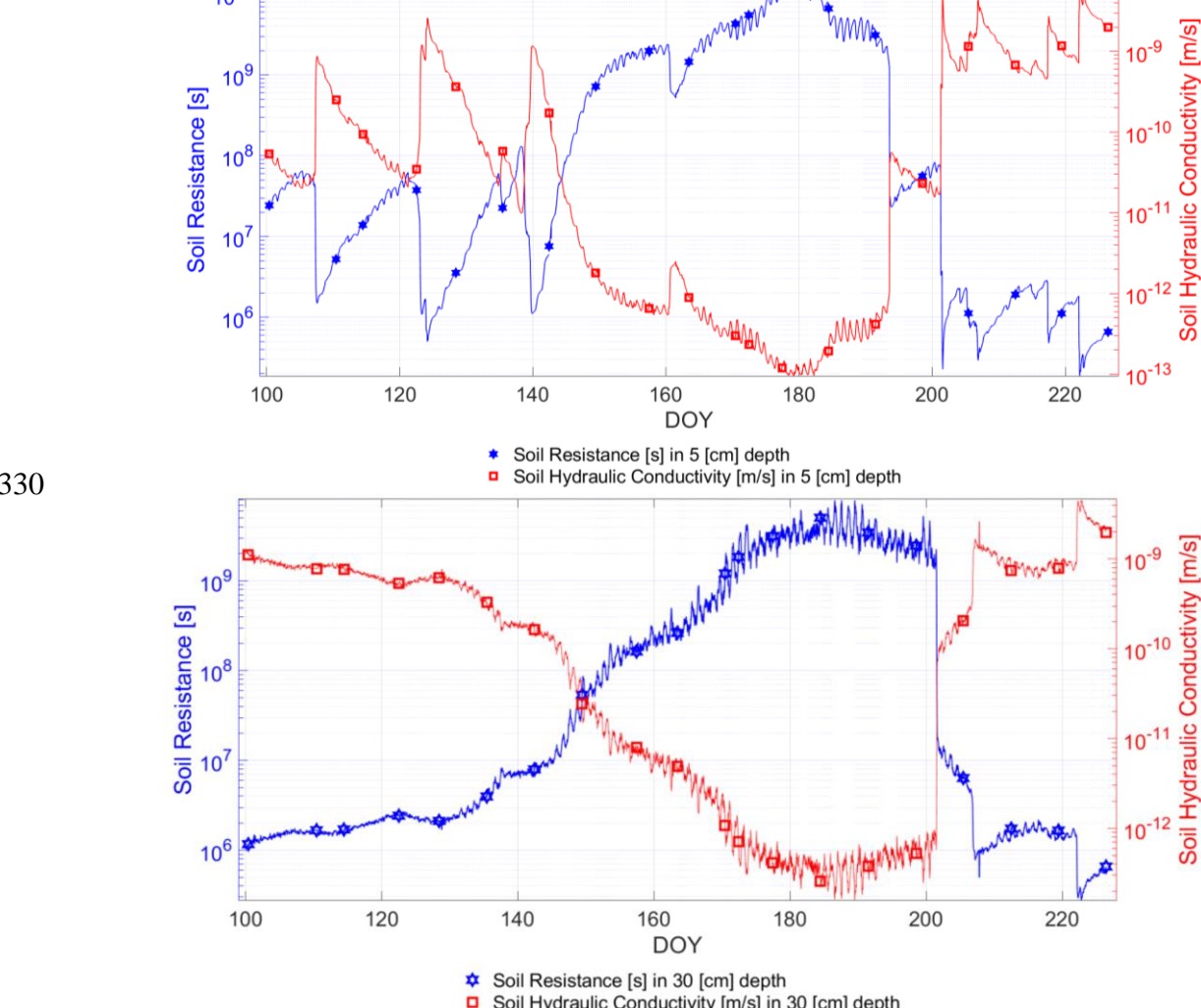


**Figure 5: Comparison of the soil permittivity-derived and Campbell (1974) model-based soil hydraulic conductivity [*m/s*] and the calculated soil resistance $R_S$ [*s*] from the model of Feddes and Rijtema (1972) implemented according to Choudhury and Idso (1985): at 5 [*cm*] (top) and 30 [*cm*] (bottom) soil depth; symbols indicate radiometer observation dates along days of year (DOY).**


For the root-xylem resistance $R_{RX}$ we deduced a *LAI*-based linear model from an approach combining in situ

measurements with boundary layer modelling, detailed in Lynn and Carlson (1990, Fig. 16):

$$R_{RX} = (-LAI \cdot 0.007 + 0.05) * C_U, \tag{6}$$

where $LAI$ $[m^2/m^2]$ was measured in situ (cf. Fig. 1) and $C_U = 0.4 \cdot 10^{10}$ represents the unit conversion from

$[bar \, (W/m^2)^{-1}]$ to $[s]$. The coefficients in (6) are empirically derived from the study of Lynn and Carlson (1990) on corn. It is to note that the relationship for winter wheat may be different than for corn. However, due to its simplicity (linear correlation with *LAI*) it allows dynamizing the root-xylem resistance along the growing

season, while keeping the amount of needed input variables constant. Figure 6 shows $\boldsymbol{R_{RX}}$ along the growing period and its linear relation to $\boldsymbol{LAI}$.

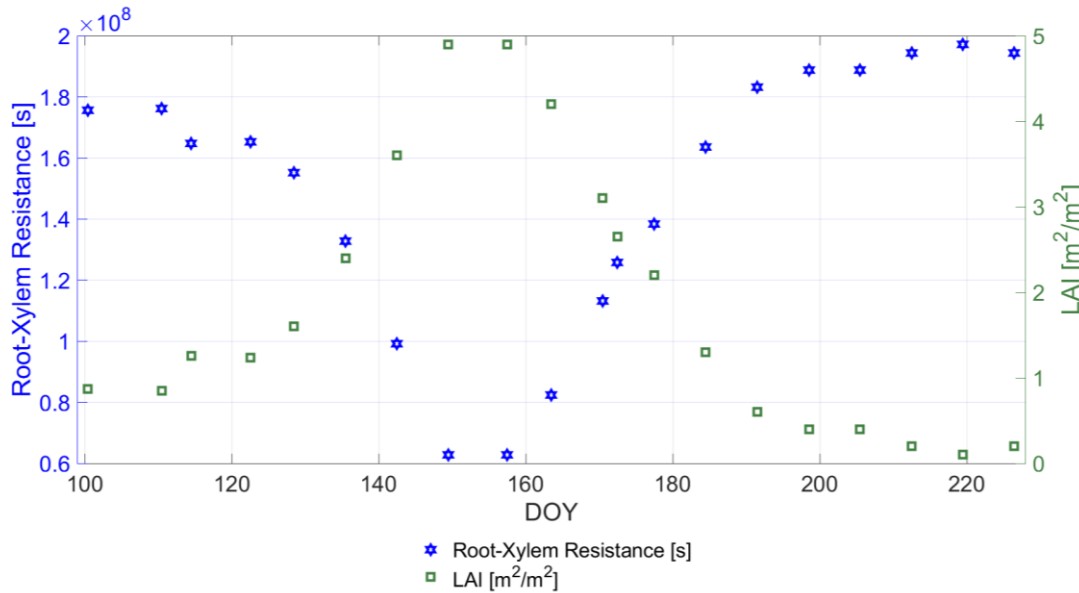


**Figure 6: Comparison of the in situ-measured leaf are index $\boldsymbol{LAI}$ $[\boldsymbol{m^2/m^2}]$ and the calculated root-xylem resistance $\boldsymbol{R_{RX}}$ $[\boldsymbol{s}]$ deduced from Lynn & Carlson (1990) along days of year (DOY).**

### 3.5 Transpiration ($TR$) from wheat plants into atmosphere

The wheat transpiration rate $TR$ $[mm/s]$ from the wheat plants into the surface-proximate atmosphere is expressed as the potential difference between the vegetation ($VWP$) and the atmosphere ($VPD$) (at 2 $[m]$ height above ground) divided by the resistances against transpiration. These latter are the resistance of the upper (adaxial) and lower (abaxial) side of the leaves and their respective stomata to water vapor outgassing and the resistance of the surrounding atmosphere (Reichardt and Timm, 2014, p.277ff; Pearcy et al., 2012, p175ff; Nobel, 2020, p.415ff):


$$TR = \frac{VPD - VWP}{R_A + 2R_C + 2R_{ST}}, \tag{7}$$

where $R_A$ $[s/m]$ is the aerodynamic resistance of the proximate atmosphere to the absorption of the vaporized moisture (Allen et al., 1998; Monteith, 1965; Chouhury and Idso, 1985). $R_C$ $[s/m]$ is the cuticular resistance of the leaf surface for direct transpiration through epidermis (fixed to $R_C = 4 \cdot 10^3$ $[s/m]$) (Monteith and Unsworth,

2013; Nobel, 2020, p.415). $R_C$ is of minor influence in this study due to non-drought conditions (Duursma et al., 2019). It is included for simplicity as an additive resistance component in (7) (Nobel, 2020, p.427ff). $R_{ST}$ $[s/m]$ is the resistance of the wheat canopy stomata to transpiration (Gallardo et al., 1996). Damour et al. (2010) provide a comprehensive overview of stomatal resistance models. Choudhury and Idso (1985) proposed an empirical model

for canopy stomatal resistance of field-grown wheat. It is adopted in the following to provide estimates of time-

varying stomatal resistance $[MPa\,s/m]$ values for the winter wheat field (Choudhury and Idso, 1985), where

density of liquid water (997 $[kg/m^3]$) and latent heat of water vaporization ($2.2564 \cdot 10^6$ $[J/kg]$) are used to convert

units from $[s/m]$ to $[MPa\,s/m]$ to match the calculus in (7) (Monteith & Unsworth, 2013). On-site

measurements of solar net radiation $[W/m^2]$ (cf. Fig. 1) and $VWP$ were used as inputs to the model. The net radiation

as well as the stomatal resistance $R_{ST}$-values are displayed in Figure 7 and compared to values reported in Nobel

(2020, p. 421). Details of the implementation are provided in Choudhury and Idso (1985).

The aerodynamic resistance $R_A$, shown in Figure 8, is a function of the wind speed (measured at 2 $[m]$ height) and

the vegetation height (cf. Fig. 1). The $R_A$-calculus was adopted from Monteith,1963. This definition was also

adopted by Allen et al. (1998) and Choudhury and Idso (1985).

Both resistance values ($R_{ST}$, $R_A$) are used together with $R_C$ in (7) to calculate time-dynamic transpiration rates $TR$.

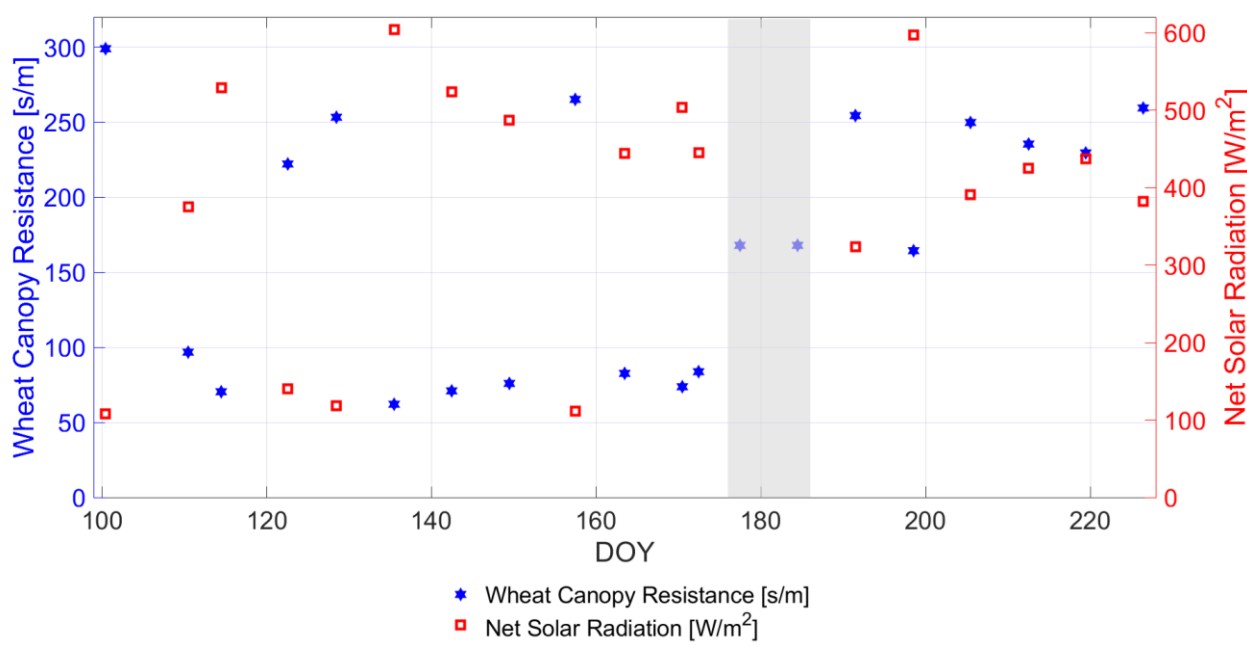


**Figure 7: Comparison of the in situ measured net solar radiation $[W/m^2]$ and the calculated stomata resistance $R_{ST}$ $[s/m]$ of the winter wheat canopy from the model of Choudhury and Idso (1985) under incorporation of $VWP$-observations (cf. Figure 1) along days of year (DOY); The gray area indicates two dates where no net solar radiation**

**measurement were available and the resistance values for the canopy are set to fixed values of 168 $[s/m]$.**

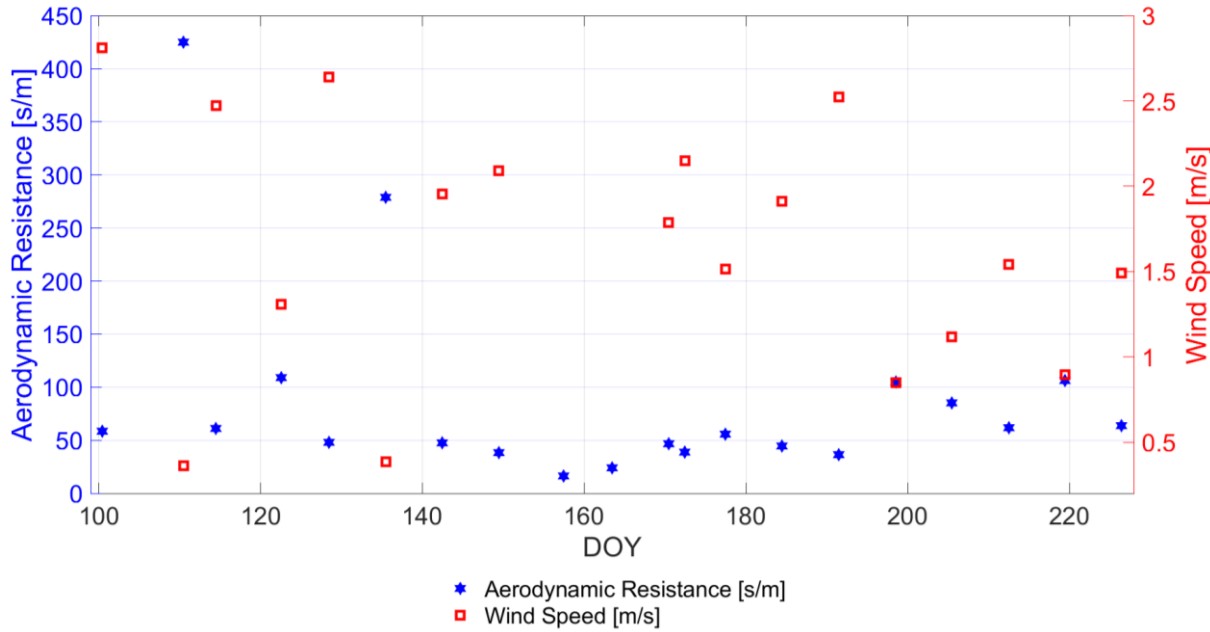

**Figure 8: Comparison of the in situ measured wind speed $[m/s]$ and the calculated aerodynamic resistance $R_A$ $[s / m]$ of the winter wheat canopy from the model of Monteith (1963) (adopted in Choudhury and Idso (1985)) under incorporation of vegetation height observations (cf. Figure 1) along days of year (DOY).**

Since $R_A$, $R_C$ and $R_{ST}$ are effective values (Wallace, 1978; Wallace and Biscoe, 1983) and not directly measurable in situ for an entire wheat field (Selhausen test site) along the full growth period (April to August 2017) (Ruggiero et al., 2007; Blizzard and Boyer, 1980), they were retrieved in a model-based way as introduced above.

Table 2 provides a summary on the non-measured variables for estimation of plant water dynamics that were obtained from parameterizations and models in this study.

**Table 2: Summary of parameterizations and models for the non-directly measured variables in the estimation of plant water dynamics of winter wheat.**

| Variable | Parameterization | Model / Reference |
|---|---|---|
| **Soil** | | |
| Soil matric potential ($SMP$) [$MPa$] | Soil moisture, soil porosity $n$, soil texture (sand, silt & clay fractions) & pore size distribution $b$ | Campbell, 1974; Clapp and Hornberger, 1978 |
| **Vegetation** | | |
| Relative water content ($RWC$) [%] | Gravimetric water content of winter wheat $m_g$ | Pearcy et al., 2012; Smart and Bingham, 1974 |
| Vegetation water potential ($VWP$) [$MPa$] | $RWC$ & empirical calibration parameters ($k_1$, $k_2$) | Zweifel and Häsler, 2000; Zweifel et al., 2001 |
| Plant water uptake ($PWU$) [$mm/s$] | $VWP$, $R_S$, $R_{RX}$ & $SMP$ | Monteith & Unsworth, 2013; Wallace, 1978; Van den Honert, 1948 |
| Soil resistance ($R_S$) [$s/m$] | Soil moisture, soil texture & rooting depth | Feddes and Rijtema, 1972; Campbell, 1974; Clapp and Hornberger, 1978 |
| Root-xylem resistance ($R_{RX}$) [$s/m$] | $LAI$ | Lynn & Carlson, 1990 |
| Transpiration rate ($TR$) [$mm/s$] | $VWP$, $R_C$, $R_{ST}$, $R_A$, & $VPD$ | Reichardt and Timm, 2014, p.277ff; Pearcy et al., 2012, p.175ff; Monteith & Unsworth, 2013, p.191 |
| Stomatal resistance ($R_{ST}$) [$s/m$] | Solar net radiation & $VWP$ | Choudhury and Idso, 1985 |
| Aerodynamic resistance ($R_A$) [$s/m$] | Wind speed & vegetation height | Allen et al., 1998; Choudhury & Idso, 1985, Monteith, 1965 |
| **Atmosphere** | | |
| Vapor pressure deficit ($VPD$) [$kPa$] | Relative air humidity & $P_{sa}$ | Castellvi et al., 1996; |
| Saturation water vapor pressure ($P_{sa}$) [$kPa$] | Air temperature | Alduchov and Eskridge, 1996 |

## 4 Results

This section shows the determined water status in the soil and wheat plants, moving on to, the estimated water

 dynamics from the soil via wheat plants into atmosphere.

**4.1 Water status in the soil**

In Figure 9, we show the estimated $SMP$s at 5 [$cm$] and 30 [$cm$] soil depth together with the soil permittivity of the top soil (5 [$cm$] below surface) and in the root zone (30 [$cm$] below surface) as well as the daily sum of precipitation. Highest permittivity values are reached in the top soil after the early senescence with ~15.5 [-], corresponding to a

soil moisture value of ~28.4 [$vol.\%$]. From DOY 160 to DOY 190 the soil permittivity stays at a low level of ~9 [-] (~16.8 [$vol.\%$]), since during that period almost no precipitation occurred. The high correlation between the $SMP$ and soil permittivity curves at 5 [$cm$] as well as at 30 [$cm$] soil depth is a result of the fact that soil permittivity is one of the main input parameters within the Campbell model to determine $SMP$, and, in addition, due to the water retention characteristics between $\theta$ and $SMP$ (cf. section 3). After every rain event when soil permittivity

increases, the $SMP$ also increases towards less negative values. This is more pronounced for the top soil layer than for the deeper one.

As the empirically fitted parameter $b$ in (1) was set to 5.3, the obtained soil water retention curve is non-linear. This effect is visible during the dry period between DOYs 140 to 190, where the $SMP$ constantly decreases to more and more negative values until DOY 180 and then increases towards less negative values again. While the soil

permittivity decreases, the $SMP$ takes on more negative values. For $SMP$ at 30 [$cm$] soil depth, the drying period starts at DOY 150, which is slightly delayed in time compared to the start of the drying period of $SMP$ at 5 [$cm$] soil depth, as indicated in Fig. 9. The drying period affects the deeper soil layers longer (until DOY 200). Furthermore, at around DOY 185 the maximum suction tension with -0.15 [$MPa$] at around DOY 185 is lower (by 0.05 [$MPa$]) for soil in the root zone than for $SMP$ in the top soil.

If $SMP$ at -0.033 [$MPa$] is considered as field capacity, the estimated results always indicate enough available water for plants to be extracted from the soil across the entire period of investigation. However, if one considers the $SMP$ at -0.01 [$MPa$] as field capacity, the winter wheat may have less available water some times during the stem elongation and after the early senescence phases. However rooting depth was not measured along the growing season of 2017, even though it would provide information about the respective water reservoirs assessible for the

wheat plants during the different phenological phases. During the period between DOY 178 and DOY 182 the results for $SMP$ at 5 [$cm$] depth show the highest suction tensions with values down to -0.2 [$MPa$], whereas the permanent wilting point is around -1.5 [$MPa$]. This indicates a drier period for the wheat plants, although water from the soil was available for the plants at any time during the growing season (cf. section 3).

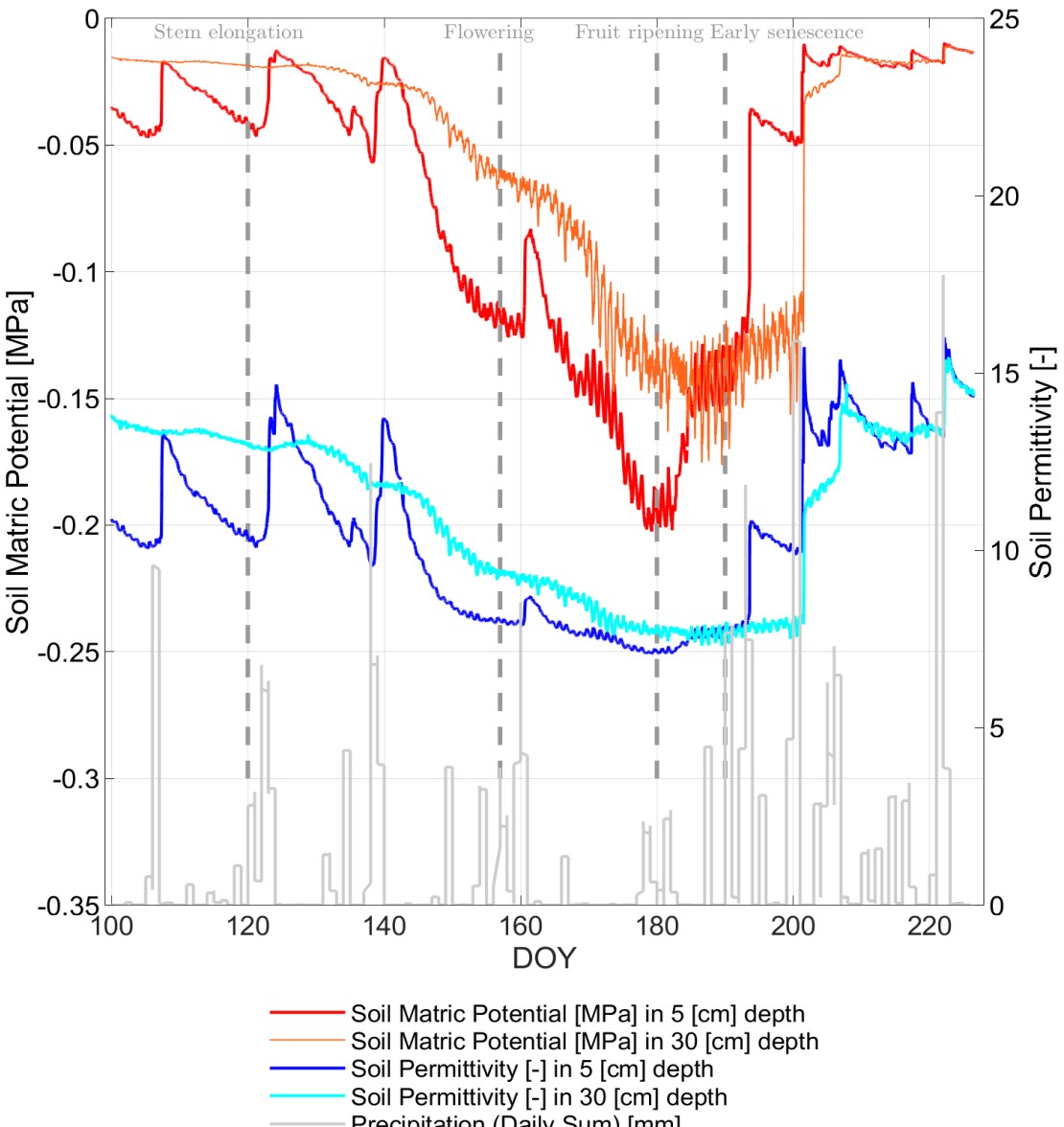

**Figure 9: Comparison of the temporal evolution of soil matric potential (*SMP*) [*MPa*], calculated after Campbell (1974) & Clapp and Hornberger (1978) in 5 [*cm*] (red curve) and 30 [*cm*] (orange curve) soil depth from in situ measured soil permittivity [-] in 5 [*cm*] (blue curve) and 30 [*cm*] (cyan curve) throughout growing season of 2017 in days of year (DOY). Gray bars indicate the daily sum of precipitation [*mm*] at the winter wheat field in Selhausen, Germany.**

## 4.2. Water status in wheat plants

In order to understand the vegetation status along growing season, $m_g$ [*kg/kg*] of the winter wheat is calculated from radiometer observations and presented in Figure 10 (Meyer et al., 2019). Figure 10 indicates the in situ measurements (gray crosses), described in section 2, and the radiometer-based estimates (blue circles) of $m_g$ over the wheat growing period (senescence from DOY 191). The overall dry down of the winter wheat along growing season accompanied by the ripening of the plants until senescence is tracked by in situ measurements as well as by the radiometer-based $m_g$-estimates, despite both are statistically independent entities.

From Figure 10, it becomes apparent that the estimates yield a higher $m_g$-level in comparison with the in situ measurements in the senescence phase (cf. Fig. 10, DOY 200-226), which can be explained by assuming in the retrieval of Meyer et al. (2019) that the vegetation volume fraction stays constant along the growing season. This is a strong, but unavoidable assumption as long as auxiliary information on vegetation volume fraction, as seen by a microwave L-band radiometer, is not at hand. At the moment, no measurements for the vegetation volume fraction from L-band radiometers exist according to the authors' knowledge.

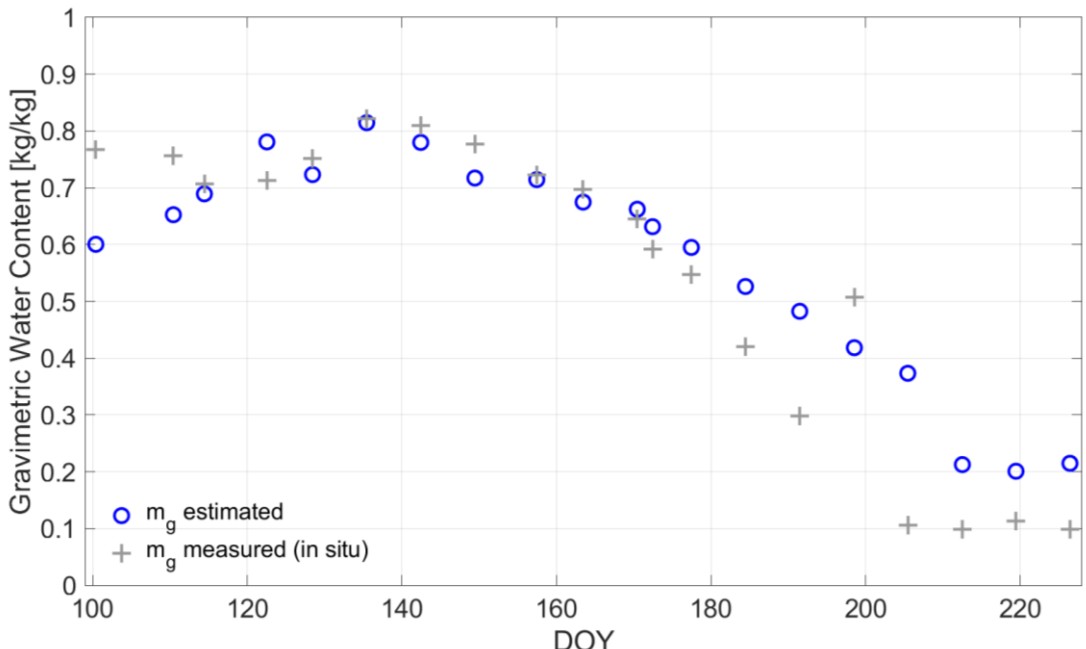

**Figure 10: Measured (gray crosses) and radiometer-derived (blue circles) gravimetric vegetation water content ($m_g$) [$kg/kg$] of winter wheat along days of year (DOY) in the growing season of 2017 at the field laboratory in Selhausen, Germany (Meyer et al., 2019).**

In Figure 11, the $RWC_{Season}$ is shown for the entire growing period after calculus from (2) based on $m_g$. A contrasting way of calculating $RWC_{Season}$ [%] from radiometer-based methods is reported in Rao et al. (2019), but adopted by using the extremes of the $VOD$ along growing season:

$$RWC_{Season,VOD} = \frac{VOD - VOD_{min}}{VOD_{max} - VOD_{min}} \cdot 100, \tag{8}$$

where $VOD_{min}$ and $VOD_{max}$ are the minimum and maximum of $VOD$ within the recorded time series. By calculating $RWC_{Season}$ in this way, it is assumed that $VOD$ is a direct indicator for $m_g$ in wheat and does not depend on biomass or vegetation structure. But, as can be seen from the course of $RWC_{Season,VOD}$ along growing period (cf. Fig. 11), the water content is low for DOY 100 (tillering), since $RWC$-values are below 10 [-].

Afterwards, they increase until DOY 157 (onset of flowering with peak of vegetation height at DOY 149) to the

level of $RWC_{Season}$, calculated from (2). In this phase from tillering to flowering, also the main biomass and plant

structure developments of the wheat plants took place (cf. Fig. 1) and are also included in the $VOD$-signal (Momen

et al., 2017). This indicates one shortcoming of assessing the water content directly on basis of $VOD$, like in (8)

(Fink et al., 2018; Meyer et al., 2019). However, in periods of constant biomass, meaning times where only the

water content in the plants would change, $RWC_{Season}$ could be directly estimated from $VOD$ (Rao et al., 2019;

Holtzman et al., 2020; Xu et al., 2021).

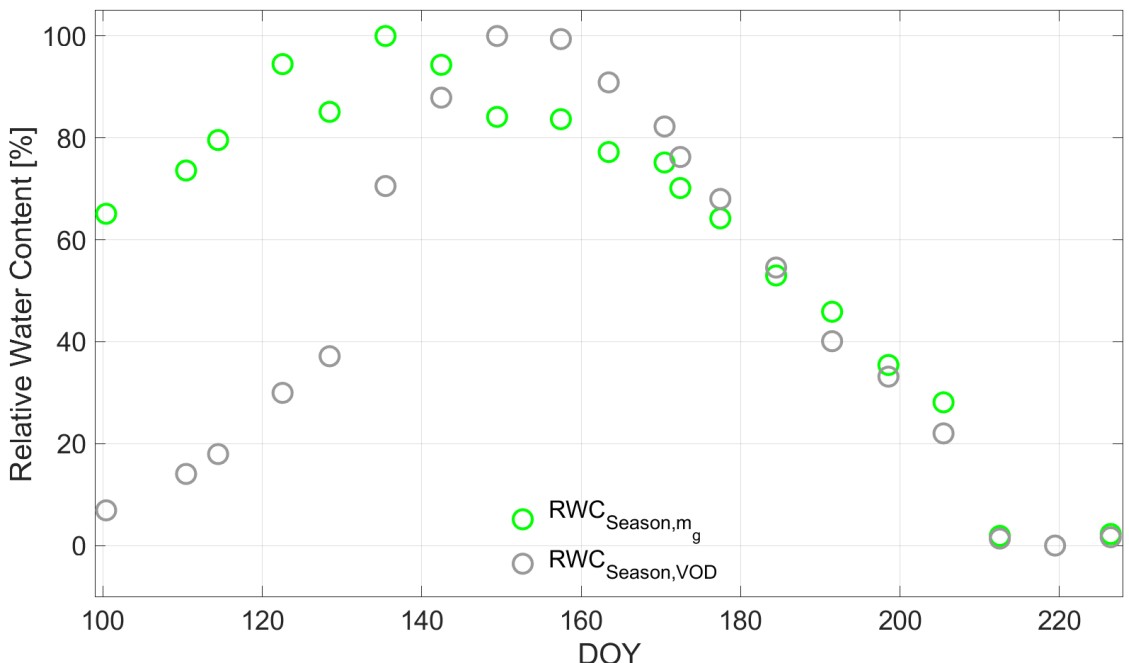

**Figure 11: Seasonal relative water content ($RWC_{Season}$) [%] calculated in (2) with radiometer-derived $m_g$ (green circles) along growing season of 2017 in days of year (DOY) at the winter wheat field in Selhausen, Germany. The gray circles**
**indicate $RWC_{Season}$ calculated directly with the radiometer-derived vegetation optical depth ($VOD$) according to (9).**

    From $RWC_{Season}$, the $VWP$ of the winter wheat can be retrieved using (3) and assuming different change rates of

$VWP$ according to $RWC_{Season}$-dynamics (cf. Fig. 3). In the following the $RWC_{Season}$ from $m_g$ is used in the

analyses. Figure 12 shows in green color the $VWP$ using intermediate change rate and in a gray area between

dashed curves the behavior of the $VWP$ according to the different assumed change rates (blue color: slow change

rate & red color: rapid change rate).

Hence, the influence of $k_1$ and $k_2$ in (3) is evident. The three different change rates (slow, intermediate and rapid)

represent the possible spread of occurring $RWC$-$VWP$ relationships for winter wheat and provide a potentially

occurring $VWP$-value range resulting from the different change rate assumptions.

The common trends of the $VWP$-value range might be interpreted as guided by the precipitation inputs along the growing season (cf. Fig. 9): $SMP$ in the top soil (0-5 [$cm$]) and in the deeper soil layers (30 [$cm$]). The $SMP$ increases to less negative values with each infiltrating precipitation impulse.

Then, $SMP$ gradually decreases in periods of dry downs (between the rain events, e.g. DOY 165 to DOY 175). $VWP$ is not following $SMP$ at any depth from DOY 100 to DOY 140. This might be partly due to the distinct

phenological changes (stem elongation) or the lower temperature regime, since mean daily temperatures can be seen to slowly increase from around 10 [$°C$] to about 20 [$°C$] in these 40 days (cf. Fig. 4). In addition, the vegetation water potential matches the $SMP$-values close to zero in this period (cf. Fig. 12). This indicates a sufficient water supply (close to or at field capacity) of the wheat plants with sufficient canopy hydration.

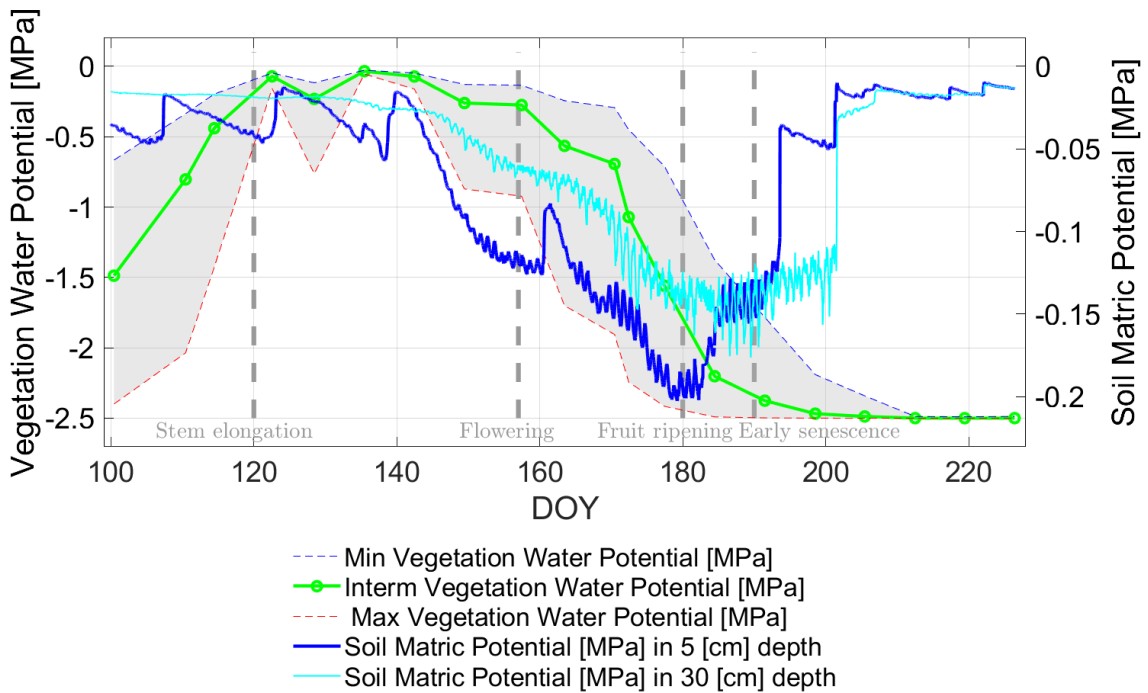

**Figure 12: Comparison of in situ-based soil matric (both blueish curves) and radiometer-based vegetation water (green solid and small dashed curves) potentials ($SMP$ [$MPa$] & $VWP$ [$MPa$]) in the growing period of 2017 along days of year (DOY) at the winter wheat field. The gray-shaded area indicates the potential variability of the estimated $VWP$ depending on applied relationship in (3) (cf. Fig. 3). Vertical dashed lines indicate major phenological phases of the winter wheat.**


A distinct decrease in $VWP$ to more negative values occurs between DOY 140 and DOY 180. In line with results from Choudhury and Idso (1985), this decrease to strongly negative values is concurrent with a period of low rainfall and high evapotranspiration due to higher air temperature with a daily mean of around 20 [$°C$] (cf. Fig. 4) and the maturation of the wheat in fruit development stage towards full biomass stage at DOY 191 (cf. also Tab. 1).

Moreover, the different sampling rates for $SMP$s and $VWP$ should be kept in mind, as $SMP$s and $VWP$ originate from different sources ($SMP$s: in situ soil moisture measurements, $VWP$: L-band radiometer observations).

However, the trend of both potentials (soil & vegetation) for the period of fully developed wheat canopy from DOY 140, with peak LAI at DOY 149 (cf. Fig. 4), to DOY 180 is apparently consistent with the in situ conditions of a diminishing water availability in soil and subsequently for the wheat plants.

Due to the onset of senescence in the wheat stand (latest DOY 200), the water supply of the drying plants degrades in importance, as the fruit (grains) ripen, meaning decreasing its content of liquid in the grains (Steduto et al.,2012; Sarto et al., 2017). Hence, water availability is not the limiting factor anymore and the concurrency of $SMP$ and $VWP$ trends vanishes completely. $SMPs$ at both depths increase to less negative values by a series of irregular rain events starting from DOY 180. $VWP$ reaches the minimum of -2.5 [$MPa$] (cf. Fig. 12) at around DOY 200. This

indicates that water loss in the plant due to senescence processes (strong increase of $R_{RX}$) reached a stage where water content ($m_g$) falls to a minimum. Subsequently, $VWP$ reaches its pre-defined minimum value ($VWP_{min}$ = -2.5 [$MPa$]), which was set for the semi-empirical relationship between water content and water potential (cf. (3) and Fig. 3).

**4.3. Seasonal water dynamics along the SPAS**

Previous results (Figs. 10-12) presented the water filling status and the water potential in soil and wheat vegetation. The analyses are now taken one step further by using the differences of the potentials (between atmosphere & vegetation as well as between vegetation & soil) together with soil, plant and atmosphere resistance estimates to assess water fluxes from soil to atmosphere: $PWU$ and $TR$.

In Figure 13 the $PWU$ [$mm/s$] (bluish curves) is depicted as time-variable uptake (depending on soil and root-

xylem resistances used in (5)) together with its radiometer-based input parameter $VWP$ (green curve). The other input parameter in equation (5) is $SMP$ which is illustrated in Figure 10 for both soil depths.

The blue $PWU$-curves in Figure 13 show an overall decline of water uptake until DOY 122. This results from a sufficient supply by precipitation water ($SMP$ < -0.05 [$MPa$]) under low evaporative conditions (daily mean of $VPD$ < 0.5 [$kPa$] & of air temperature around 10 [$°C$]) and wheat plants in tillering and early stem elongation phase,

but without major biomass development (DOY 122: $LAI$ = 1.24 [-], plant height = 0.25 [$m$] & above ground biomass = 0.24 [$kg/m²$]; all values from Fig. 1 & Tab. 1).

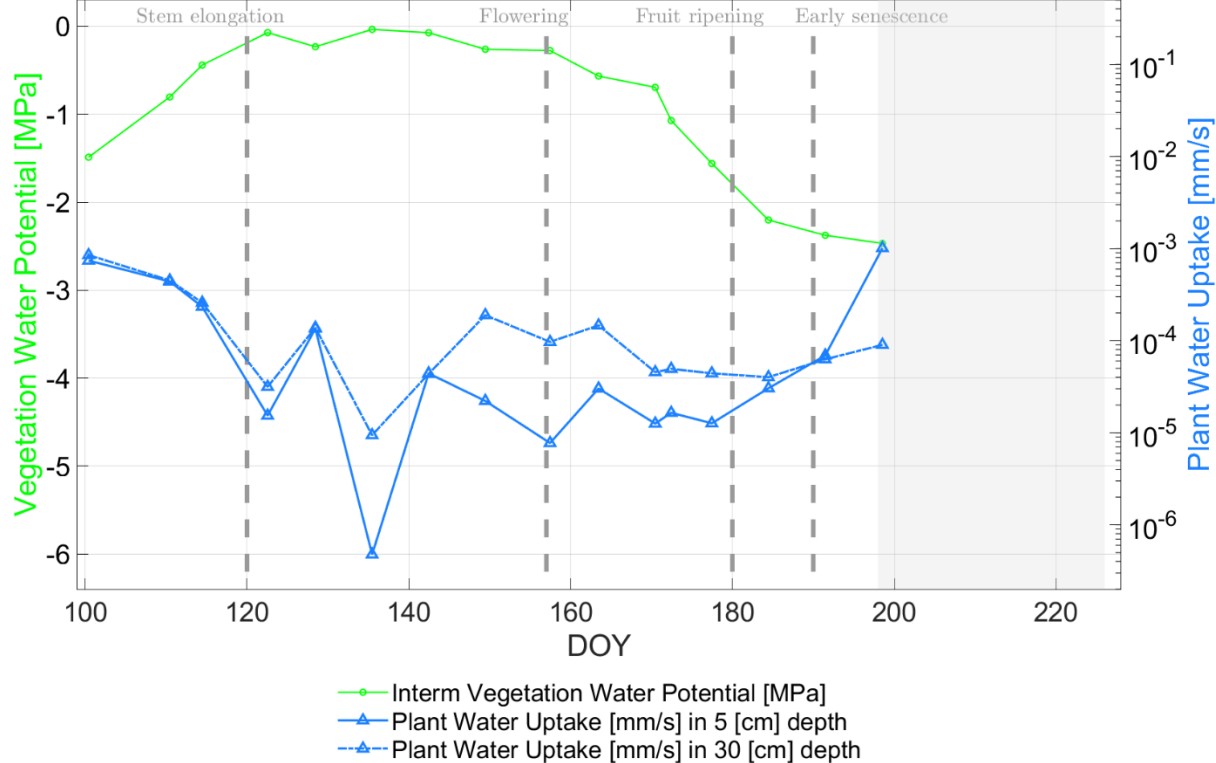

**Figure 13:** Comparison of radiometer-based plant water uptake $PWU$ [$mm/s$] (blueish-colored curves with variable soil resistance at 5 [$cm$] and 30 [$cm$] soil depth) and vegetation water (green solid curve) potential $VWP$ [$MPa$] along the growing season of 2017 for days of year (DOY) at the winter wheat field in Selhausen, Germany. The gray-masked area indicates the senescence phase where the water uptake estimates are not valid anymore due to simplifying assumptions in calculus of the root-xylem resistance. Vertical dashed lines indicate major phenological phases of the winter wheat.

From DOY 122 to DOY 142 (stem elongation phase) the biomass level increased slightly, but due to the wheat stalk development, the vegetation height and $LAI$ changed significantly from 0.25 [$m$] to 0.56 [$m$] and 1.24 [-] to 3.6 [-], respectively (cf. Fig. 1). Moreover, the soil hydraulic conductivity and the respective soil flow resistance $R_S$ (cf. Fig. 5) showed a steady decrease in water flow conditions at 30 [$cm$] depth and a fluctuating behavior due to rain impulses at 5 [$cm$] depth. Hence, the $PWU$-curves (blueish lines in Fig. 13) show the same trends for 30 [$cm$] and 5 [$cm$] depth having a lower level of $PWU$ at shallower soil depth resulting from smaller soil hydraulic conductivity values (cf. Fig. 5). The lowest $PWU$-values in the range of $10^{-6} - 10^{-5}$ [$mm/s$] are supported by analysis of Cai et al. 2018, working at the same test site (Selhausen) and the same crop type (winter wheat), but already in 2016. The root-xylem resistance $R_{RX}$ in Figure 6 indicates a steady decrease until DOY 142, clearly following its driving parameter $LAI$. The assumption behind is an increase in roots and xylem vessels concurrent to leaf growth leading to an increase in flow capacities along plant development (Lynn and Carlson, 1990). Until DOY 142, $PWU$-curves are guided by $R_{RX}$ rather than $R_S$ due to stronger resistance values of the root-xylem system in the early development stages of the winter wheat compared to a sufficiently watered soil for uptake (cf. Figs. 5 & 6).

Afterwards, $PWU$ decreased and increased with different strength from DOY 142 until senescence phase (DOY >

190). First, there is a distinct decrease from DOY 142 to DOY 157 (inflorescence and flowering phase, cf. Tab. 1).

In this phase the soil moisture dropped mainly due to a daily mean air temperature of about 20 [$°C$] and absence of

major precipitation events, leading to a $SMP$-decrease from -0.03 [$MPa$] to -0.12 [$MPa$]. This is accompanied with

a significant increase in soil resistance (cf. Fig. 5) and a decrease in root-xylem resistance (cf. Fig. 6).

From DOY 157 to DOY 180 (flowering and grain development phase) rain input was absent and soil water decreased

with $SMP$–values up to -0.2 [$MPa$] on DOY 180 and a soil conductivity minimum of $1.1 \times 10^{-13}$ [$m/s$] (cf.

Fig. 5). The $VWP$-values strongly dropped from -0.28 [$MPa$] to about -1.75 [$MPa$]. In this period $PWU$ increased,

in the beginning stronger then only slightly, indicating water depletion from the soil (cf. Fig. 9) in the reproductive

stage of the wheat plants.

In the last grain development phase (DOY 180 to DOY 190) before senescence (DOY > 190), the $VWP$ further

decreased to -2.38 [$MPa$], but with a milder slope due to refilling water in soils (note that $SMP$ for both soil depths

raised to -0.14 [$MPa$] during several consecutive rain events as seen in Figure 9 as gray bars). The $PWU$-curves at

5 [$cm$] depth start to rise again from DOY 191 to DOY 198 due to the first strong rain impulses seen first by the top

soil (cf. Fig. 13).

In the senescence phase (DOY 199), the soil moisture was not the limiting factor anymore and $VWP$ dropped to

its pre-defined minimum of -2.5 [$MPa$] as soon as $RWC$ reaches a level of 35 [%], subsequently approaching almost

zero beyond DOY 210 (cf. Fig. 11). In Figure 11 a gray-masked area indicates the senescence phase where the

water uptake estimates are not valid anymore, since root-xylem resistance is considered to be solely dependent on

$LAI$ in (6).

After uptake of water into the wheat plants, we focus in the following on water release from the wheat plants into

the atmosphere. In Figure 14, the estimated transpiration rates $TR$ [$mm/s$] are shown along the growing season

including its input variables $VWP$ and $VPD$. The value range of the $TR$-estimates is similar to the range presented

by Kang et al. 2003 and Zhang et al. 2019 for winter wheat fields, whereby the latter also analyzed different

fertilization levels. The general trend of the $TR$-curves indicates considerable concurrency with the $PWU$-curves

over the entire season in Figure 13. This connection of dynamics, pointing towards steady water flow along the

SPAS, supports the connection of the different water storage compartments in the system.

However, the magnitudes of $TR$ and $PWU$ are different especially around the ripening phase of the wheat (around

DOY 180) due to differently modelled time-variant flow resistances for both processes. This is more decisive for

transpiration (including radiation- and temperature-dependence within $VPD$) having a stronger diurnal cycle than

water uptake (including mainly moisture-dependence within soil matric potential). Hence the temporal dynamics and magnitudes of $TR$ and $PWU$ exhibit no absolute concurrency due to dependence of $TR$ to faster changing atmospheric conditions, while dependence of $PWU$ is on slower changing soil conditions.

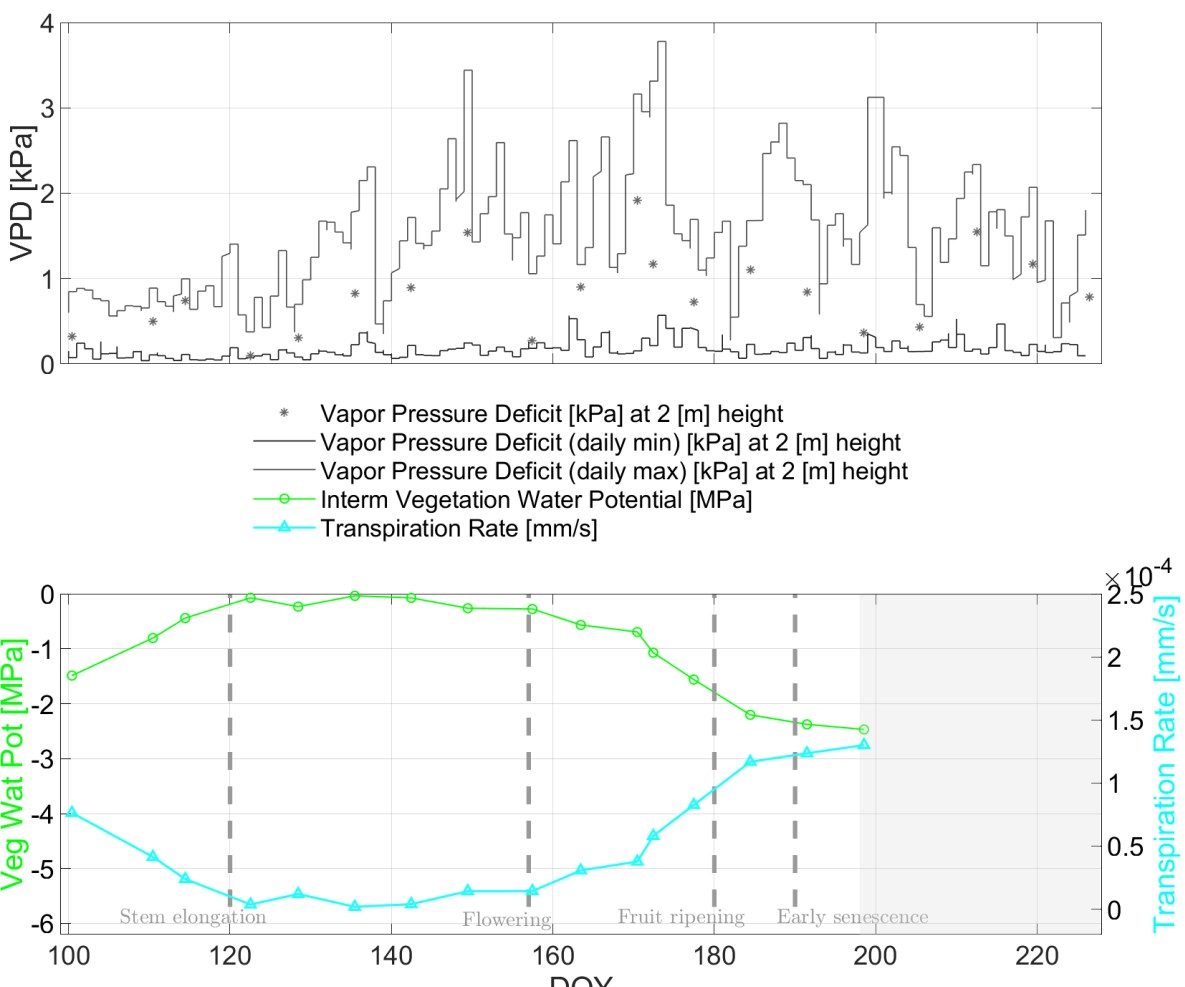

**Figure 14: Top: Minimum and maximum of the vapor pressure deficit ($VPD$) $[kPa]$ for atmospheric water need shown with gray lines. The asterisks indicate the $VPD$–values at time of radiometer measurement. The x-axis labels are the same as for the bottom plot.**
**Bottom: Comparison of radiometer-based transpiration rates $TR$ $[mm/s]$ (cyan curve for variable canopy & aerodynamic resistances as well as a fixed cuticular resistance) and vegetation water (green curve) potential $VWP$ $[MPa]$ along the growing period of 2017 for days of year (DOY) at the winter wheat field in Selhausen, Germany. The gray-masked area indicates the senescence phase where the assumption in the stomatal resistance calculus is not applicable. Vertical dashed lines indicate major phenological phases of the winter wheat.**

Considering both flux estimates, we call to mind that for calculating $PWU$ and $TR$ two substantially different in situ measurements ($SMP$, $VPD$) were paired together with the same radiometer-based $VWP$-input. However, the dynamics of both fluxes ($PWU$, $TR$) are concurrent in trend and show considerable similarity along wheat growing season. This comes with the caveat that the applied resistances of the soil-plant-atmosphere compartments were derived from literature ($R_C$) or form auxiliary in situ information (e.g. soil hydraulic conductivity, leaf area index, net solar radiation and wind speed) applied in empirical models ($R_S$, $R_{RX}$, $R_{ST}$, $R_A$), (Manzoni et al., 2013b,

Monteith & Unsworth, 2013). Nonetheless, $VWP$ as a radiometer-based potential estimate shows considerable similarity in temporal dynamics to the on-site measurement-derived potentials of soil ($SMP$) and atmosphere

($VPD$).

For a first comparison with an independent dataset, the estimated transpiration rates of the space-borne ECOSTRESS sensor were used (cf. section 2 for sensor and product characteristics) and shown in Figure 15. Our results show that the value ranges of the ECOSTRESS estimates are equivalent (showing the same order of magnitude) to the field-based $TR$ estimates at Selhausen. In Figure 15 the $TR$ estimates are presented for several

years (2017, 2019-2021) and with different sample sizes. Note that the heterogeneity of land surface conditions at the spatial resolution of ECOSTRESS (70 [$m$]) does not allow for direct validation against the in situ measurements.

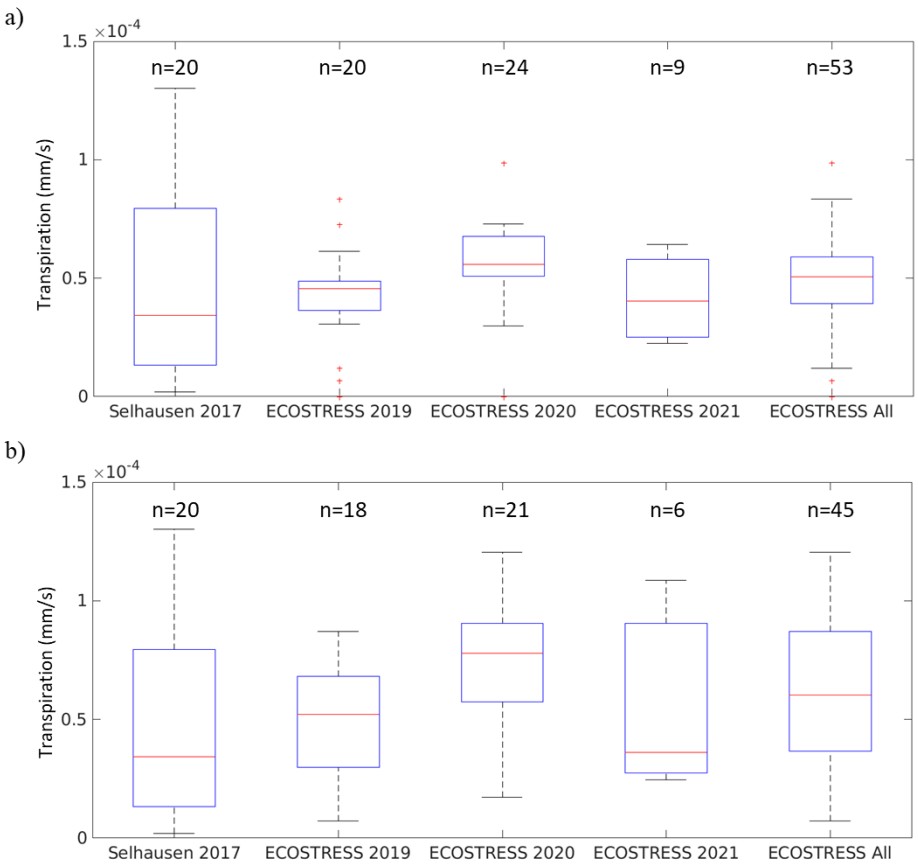

**Figure 15: Boxplots of estimated transpiration rates $TR$ [$mm/s$] comparing the field-based estimates presented for the**
**Selhausen test site (see Fig. 14) and (a) the space-borne ECOSTRESS daily product and (b) the instantaneous product. The number of samples (n) is indicated at the top of each box. For each box, the red line shows the median, top and bottom edges are percentiles 75 and 25, respectively, top and bottom whiskers are percentiles 95 and 5, respectively, and red crosses indicate outliers.**

**5 Summary and discussion of concept**

This concept study presents a radiometer-based approach to estimate time-dynamic flux rates (plant water uptake & transpiration) from water status observations along the SPAS, even if assumptions on water flow properties (soil & plant hydraulic traits, e.g. effective soil porosity, rooting depth and flow resistances) and some ancillary data are necessary. With the proposed approach, analyses of the SPAS are enabled covering the different water pools (soil, plant and atmosphere), their potentials (suction tension) and in-between fluxes (uptake, transpiration). Water pools

are accessible depending on measurement depth and sensing volume of the in situ and remote sensing sensors (cf. section 2). In the case of the in situ soil instruments only the soil moisture and $SMP$ down to 30 [$cm$] were accessible which may not cover the entire water reservoir from which the wheat roots can take up water. The water may also come from deeper layers. However, White et al. (2015) report that for winter wheat (in 17 experiments) the soil depths of 10 [$cm$] and 30 [$cm$] showed a median of the root length density (RLD) above the critical RLD

of 1 [$cm\ cm^{-3}$] for wheat.

In addition to the limitation in sensing depth, the study solely relies on snapshot analysis of radiometer-based vegetation parameters (e.g. $m_g$, $RWC$ and $VWP$) for single days along the growing period, since radiometer measurements were only conducted over one to two hours within 9 am and 2 pm on a measuring day. No analyses about the diurnal cycle of $RWC$, $VWP$, $PWU$ and $TR$ or their sub-daily variability within the wheat canopy can

be provided, but are reported in literature (e.g. Dutt and Gill, 1978). Thus, the study results are representative with respect to their trends along the growing season and not for diel dynamics (Konings et al., 2021).

Moreover, as the setup of the experiment at the Selhausen field laboratory is temporally confined (one growing season), specific in setup (metal mesh for soil signal blockage; only vegetation-related signals included), small-scale (one field) and single-species (only winter wheat), it is at this moment not possible to significantly generalize

the presented results with respect to other temporal and spatial domains, species (traits & phenologies) and scales (cf. Konings et al., 2019; Konings et al., 2021). Furthermore, the generalization of the method is challenging, as for instance, the $RWC$ to $VWP$ conversion (cf. Fig. 3) is plant type-specific (Martinez-Vilalta et al., 2019) and not yet well understood for spatio-temporal resolution conditions of satellite radiometer (passive microwave) remote sensing with resolutions of kilometers in space and days in time (cf. Fig. 2 in Konings et al., 2019). This

means the rate and strength of water supply on potential (suction) reduction is not accessible at field scale and along growing season from the on-site measurements, but in a first attempt deducible from literature survey (Turner and Long, 1980; Turner, 1988; Zweifel and Häsler, 2000; Zweifel et al., 2001; Pearcy et al., 2012).

Since variables, like the soil, wheat canopy and atmospheric resistances, are not directly measured in situ at the field scale, they are estimated in the study using (wheat-specific) mechanistic models for soil, root-xylem, stomatal and aerodynamic resistances fed with (auxiliary) on-site measurements (cf. sections 3.2 & 3.4 for details). However, the mechanistic models are empirically derived and therefore not complete in terms of their inclusion of all potentially occurring flow resistances within the wheat plants, e.g. intercellular air space resistance.

We conducted an uncertainty analysis on the flow resistances and found that introducing up to 20% uncertainty does not change the seasonal trend of the $PWU$- and $TR$-estimates. However, the higher the initial $PWU$- and $TR$-estimates, the stronger was the effect of the included uncertainty. This led to uncertainty-induced changes of $PWU$ and $TR$ of maximum $3.5 \cdot 10^{-4}$ $[mm/s]$ (30 [cm] depth), $4.2 \cdot 10^{-4}$ $[mm/s]$ (5 [cm] depth) and $5.4 \cdot 10^{-5}$ $[mm/s]$, respectively, when including 20% uncertainty.

As mentioned briefly before, the soil emission is completely blocked physically by a metal mesh on the ground (Meyer et al., 2018). This could be recognized as a burden on larger-area monitoring of water dynamics with space-borne radiometer-based retrievals. However, a joint approach combining emission modeling and data inversion (e.g. using zeroth- or first-order radiative transfer models (Wigneron et al., 2017)) was tested in Meyer et al. (2018) on another part of the wheat field without mesh coverage. Soil and vegetation emission signals were separated leading to equivalent estimates of soil moisture and $VOD$ compared to the values from the mesh-covered part ($VOD$) and from the in situ measurements (soil moisture). Hence, soil moisture products from operating satellite radiometer missions, such as SMAP, could be used for wide-area studies (Konings et al., 2017).

Despite all restrictions (due to having no complete measurement portfolio of all SPAS variables), no attempts have been reported in literature which follow a similar approach combining soil, plant and atmosphere information from L-band passive radiometry with (on-site) measurements at the field scale. One exception is the correlation of L-band radiometer-derived $VOD$ with water potential measurements of a tree stand at Harvard forest along one growing season (Holtzman et al., 2020). In any case, there is a need for remote sensing-scale analyses of transfer-functions from $VOD$ via water content to water potential in plants, which must be different from laboratory analyses of single tissues of leaves (Turner, 1988).

In this way, atmosphere and soil information, like the $VPD$ or soil moisture, can already be derived today on global scale from space-borne remote sensing sources, e.g. $VPD$ from Atmospheric Infrared Sounder (AIRS) on-board NASA's Aqua satellite (Feldman et al., 2020) or surface soil moisture from SMAP or SMOS microwave radiometer

missions (Entekhabi et al., 2010; Kerr et al., 2010). In addition, a major part of the additional soil-plant-atmosphere parameters, needed as input in the presented water dynamic calculus and listed in Table 2, can already be estimated from multi-sensor remote sensing, including lidars (vegetation height, e.g. from the Global Ecosystem Dynamics Investigation (GEDI) lidar), optical sensors (solar net radiation,, $LAI$, e.g. from Copernicus Global Land Service using the Sentinel-3 instrument), radars/scatterometers (vegetation volume fraction, $VOD$, e.g. from ASCAT (Liu et al., 2021)) and radiometers ($VOD$). However, this comes with the limitations in spatio-temporal as well as spectral coverage of remote sensing systems, no matter if active (e.g. lidar, radar) or passive (e.g. spectrometer, radiometer) systems are used (Horning et al., 2010). Moreover, it has to be acknowledged that remote sensing acquisitions do not purely sense one variable of the earth system, but normally a mixture of variables (e.g. combination of soil and vegetation variables) (Jackson and Schmugge, 1991; Jagdhuber, 2012). Hence, the quality of retrieved earth system variables (e.g. soil or plant moisture), extracted from remotely sensed observations, depends directly on the sophistication of the signal-to-variable conversion (e.g. Du et al., 2016).

As a matter of fact, L-band radiometry does not measure fluxes directly, but brightness temperatures (Ulaby and Long, 2013). Hence, estimates of the water reservoirs (soil moisture, plant moisture and relative humidity of atmosphere) need to be retrieved by multi-sensor remote sensing. Afterwards, performant estimates of the water potentials need to be established based on the water reservoir estimates (Holtzman, 2021; Jagdhuber et al., 2021) or other techniques (first attempts with GPS sensors in Humphrey (2021)). In the end, the water potential estimates need to be converted to the water fluxes, here the essential auxiliaries are the flow resistances of the soil, vegetation and atmosphere (Pearcy et al., 2012; Nobel, 2020). For these reasons, we advocate that in order to retrieve exact water flow dynamics at remote sensing scales, a plausible solution may come from the combination of soil-plant-atmosphere transport or vegetation growth models and high spatio-temporal resolution remote sensing data from multiple instruments (Wang and Engel, 2002; Palosuo et al., 2011). This multi-source approach will be key for applications requiring quantitative estimates of water fluxes in time and space and should be the subject of further research.

**6 Conclusions and outlook**

The objective of this concept study was to investigate whether observations of a ground-based radiometer over a winter wheat field allow for the estimation of seasonal flux rates of water (plant water uptake and transpiration) along the soil-plant-atmosphere system (SPAS). We started from L-band vegetation optical depth ($VOD$) together with on-site measurements of soil and atmosphere. The major research question was how far these observations contain enough information to derive water potentials and flux rates.

First conclusions can be drawn within the boundaries of the experimental setup, discussed in the former section. Arriving at exact water fluxes in the SPAS of winter wheat needs further information (e.g. about soil, plant and atmosphere resistances) than initially acquired during the campaign in 2017 at the Selhausen (Germany) field laboratory. The campaign was originally not designed for studying water fluxes in the SPAS. Existing on-site measurements of soil (permittivity), plant (height & $LAI$) and atmosphere (net radiation, temperature & relative humidity of air) were used here to estimate water flux rates ($PWU$ & $TR$). Hence, it is imperative to have water status information ($\theta$, $m_g$, $RH$) along the SPAS for estimating water dynamics (cf. Fig. 2). Otherwise, these dynamics cannot be fully assessed.

We were able to obtain reasonable estimates of water potentials and water flux rates in terms of the major trend along season (cf. sections 4.2 - 4.3 and Figs. 12-14). The results reveal the capacity of the setup to assess water dynamics of a wheat field. A first comparison of $TR$ estimates from the presented field-based approach and from the space-borne ECOSTRESS mission indicates similar value ranges (same order of magnitude, mainly between zero and $1.0 \cdot 10^{-4}$ [$mm/s$]). However, the validation of absolute accuracies needs to be tackled in future studies with dedicated in situ measurements of water dynamics (potentials & flux rates). This is especially true for the $PWU$ estimates where no comparison or validation dataset was available in contrast to the $TR$ case.

In these dedicated field laboratory studies, explicitly including ground-based (L-band) radiometry and other remote sensing sensors together with in situ measurements, the focus will be on validation of the water flux dynamics with measurements of soil water potential, leaf water potential, sap flux, stomatal conductance and transpiration rates. Based on the initial results of this study, we advocate that a hybrid-based (remote sensing & earth system models), large-area (up to global) SPAS assessment could potentially be established in the future. For this, the study indicates that with more measurements or wider knowledge about plant characteristics regarding water flux, the approach should be applicable to more plant types and beyond field scale connecting to the spatial mapping

capabilities of space-borne sensors. Future missions, like the Copernicus Imaging Microwave Radiometer (CIMR) of the European Space Agency (ESA) (Kilic et al., 2018), could be potential candidates for addressing research on water fluxes within the SPAS on a global scale. This would exploit the benefit of multiple sensing frequencies allowing for different penetration and transmission capabilities through vegetation canopies (Prigent and Jiménez, 2021; Zhao et al., 2021). Advanced methodologies exploiting the synergies from present and planned satellite constellations will possibly allow a frequent and large-scale mapping of water fluxes through the SPAS.

*Acknowledgments.* The authors thank Dr. Andrew Feldman (MIT), Prof. Dr. Dara Entekhabi (MIT), Dr. Stan Schymanski (LIST) and Prof. Dr. Jordi Martínez-Vilalta (CREAF) for their helpful comments and suggestions supporting this research. They also thank Mr. Mark Lützner for language editing. The authors want to acknowledge MIT for supporting this research with the MIT-Germany Seed Fund "Global Water Cycle and Environmental Monitoring using Active and Passive Satellite-based Microwave Instruments" and with the MIT-Belgium UCL Seed Fund "Early Detection of Plant Water Stress Using Remote Sensing". The ELBARA-II radiometer was provided by the Terrestrial Environmental Observatories (TERENO) initiative funded by the Helmholtz Association of German Research Centers. D.C. has received funding by "la Caixa" Foundation (ID 100010434), under agreement LCF/PR/MIT19/51840001, by the MIT-MISTI, by the XXXIII Ramón Areces Postdoctoral Fellowship, and by grants PID2020-114623RB-C32 and MDM-2016-0600 funded by MCIN/AEI/10.13039/501100011033 and by the European Regional Development Fund (ERDF, EU). M.P. was supported by grant RTI2018-096765-A-100 funded by MCIN/AEI/ 10.13039/501100011033 and by the European Regional Development Fund (ERDF, EU).

*Author Contributions.* Conceptualization, T.J., F.J. and M.P.; Methodology, T.J., A.F., F.J., D.C. and M.P.; Software, T.J., F.J., A.F. and T.M.; Formal Analysis, All; Data Curation, T.M., F.J., T.J. and A.F.; Writing-Original Draft Preparation, T.J., A.F., F.J., D.C., M.J.B. and M.P.; Writing-Review & Editing, All; Visualization, T.J., A.F., F.J., M.J.B. and T.M.; Supervision, M.P..

*Data availability.* All data were recorded by Forschungszentrum Jülich, Institute of Bio- and Geosciences, Agrosphere (IGB-3), and are partly available upon request from François Jonard (f.jonard@fz-Juelich.de). TERENO data are available at www.tereno.net. ECOSTRESS datasets are freely available at https://ecostress.jpl.nasa.gov/data/.

*Competing interests.* The authors declare that they have no conflict of interest.

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
