# Peer review of "Towards Estimation of Seasonal Water Dynamics of Winter Wheat from Ground-Based L-Band Radiometry: A Concept Study"

_Biogeosciences, 2021_

## Referee Comment (RC2)

**Review of Biogeosciences manuscript bg-2021-71,**
**"Towards Estimation of Seasonal Water Dynamics of Winter Wheat from Ground-Based L-Band Radiometry"**

by Thomas Jagdhuber, François Jonard, Anke Fluhrer, David Chaparro, Martin J. Baur, Thomas Meyer and María Piles

**Summary:**

The manuscript presents a radio-meter based approach along with on-site measurements to estimate seasonal flux rates of water over a winter wheat field. The paper is well written, and the manuscript exhibits useful results. There are just a few aspects that need to be addressed before publication. First, the paper lacks other sources of data (e.g., satellite products and/or field laboratory data) to validate the employed empirical models and results. I'd suggest the authors at least include a few other observations to validate the overall utilized approach. Second, the paper requires some further modifications and/or clarifications in different parts. Based on these shortcomings, I recommend a minor revision. The authors should consider the following comments in their revision.

**Major Comments:**

**Comment (1):** Line 105: Why this particular plant has been selected for this study? What are the characteristics that distinguish it from other plants? How does selecting a plant and its hydraulic traits influence the final conclusions of the research? The authors need to comment on these.

**Comment (2):** Equation 6: This model seems to have some empirical coefficients. Are these coefficients plant-type dependent? In Lynn and Carlson (1990), Fig. 16 is depicted for corn. How can that impact the used model in this study? The authors need to comment on these.

**Comment (3):** Figure 11 and Line 420: Something that perplexes me is that the LAI is changing nonlinearly in the whole duration of the measurements according to Figure 6 implying that the total biomass is changing. If this is true, the comparison shown in this study does not seem valid (based on Line 420) and does not add anything to the paper.

**Comment (4):** Figure 11, and a general comment: In general, one downside of the paper is that it does not compare the obtained results with other remote sensing products and/or laboratory analysis. This is significant for validation of the employed empirical models and results. In particular, authors can compare their derived RWC$_{VOD}$ (Fig. 11) or soil moisture with satellite

products. Although the resolution might be different, it is expected to see generally a similar trend that can further validate the employed methods.

**Comment (5):** Line 560: How can such water flow estimations be done solely using remote sensing data? The authors could add some discussions on this and the deficiencies of remote sensing approaches to fully capture the water flow dynamics. I noticed that this has been somewhat discussed in lines 610-620 but more discussions focusing on the limitations and deficiencies of such remote sensing data would be useful especially for large-scale studies.

**Minor Comments:**

**Comment (1):** Line 125: How far is the climate station from the measurement site?

**Comment (2):** Figure 1: How much is VWC correlated with LAI?

---

## Author Comment (AC1)

Response to Reviewer 1

**Towards Estimation of Seasonal Water Dynamics of Winter Wheat from Ground-Based L-Band Radiometry (Manuscript # BG-2021-71)**

| Comments | Responses/Actions |
|---|---|
| In this paper the authors seek to show that L-band radiometry can improve water dynamics estimation based on the Soil-Plant-Atmosphere System (SPAS). The methodology presented in the paper is relevant to the special issue and current L-band missions such as SMAP, and builds upon previous L-band research in Vegetation Optical Depth (VOD). While the method utilizing L-band radiometry and existing physical models to estimate wheat water dynamics is described in some detail, I have two major concerns: | Many thanks for confirming the relevance of the manuscript for the special issue. According to the reviewer comments, we will work on all raised issues with special focus on the two major comments:
 ▪ Validation of plant water dynamics
 ▪ Role of in situ measurements in the study. |
| 1. The field data used does not contain in situ measurements for the target variables Transpiration Rate (TR) and Plant Water Uptake (PWU), leaving the authors to discuss results in vague terms of what 'might be a first indication to the feasibility' of their method without any validation. In the absence of any strong validation data, the paper could be a short communication rather than a full-length research paper. | Validation of plant water dynamics:
 We agree that the presented estimates of transpiration rate ($TR$) and plant water uptake ($PWU$) were not tracked by a set of in situ measurements from the dedicated field laboratory experiment along the growing season of 2017 (Meyer et al., 2018). The experiment was originally not designed for this purpose, but for estimating vegetation optical depth ($VOD$) and gravimetric plant water content from L-band microwave radiometry at the field scale and for one entire growing season of 2017 (Meyer et al., 2018; Meyer et al., 2019).

 One of the main innovations of the presented path finder research study is to elaborate a concept, foremost a methodology, to concert classical in situ measurements and $VOD$ for finding a way to arrive synergistically (in situ with microwave remote-sensing combined) at estimated $PWU$ and $TR$. This is a conceptual step forward in water dynamics estimation incorporating $VOD$ in a field experimental setup leading to the projection of a future majorly remote sensing-based methodology to retrieve $PWU$ and $TR$. |

| | |
|---|---|
| | We want to acknowledge this fact by adapting the title of our study and in this way preparing the reader for a concept-focused, rather than a validation-based, study. Suggestion for the new title is: "Towards Estimation of Seasonal Water Dynamics of Winter Wheat From Ground-Based L-Band Radiometry: A Concept Study". |
| | Moreover, note we explicitly stress in the manuscript (in Sections: Introduction (l.38-39), and Conclusions (l.638-639, 650-652)) that its scientific contribution is on the concept and methodology of estimating water dynamics by retrieving L-band radiometer-derived estimates and orchestrating them with on-site measurements for arriving at estimates of plant water dynamics. To our knowledge, this is the first time that an end-to-end SPAS analysis is conducted using mechanistic models and input data available from in-situ and remote sensors. |
| | We agree with the reviewer that this research study cannot serve as a validation study, meaning as a validation of an already existing methodology. Still, following the reviewer suggestion, we will consider different approaches with the aim of including an initial assessment of our estimated water potential and water dynamics ($PWU$, $TR$) with independently measured/derived entities of these variables in the revised version of the manuscript. To this end, we will investigate the following options: |
| |    1.  Comparison with space-borne $VOD$ from radiometer missions (e.g. SMAP or SMOS). |
| |    2.  Comparison with evapotranspiration data from the remote sensing-based EcoSTRESS mission (starting from 2018): https://ecostress.jpl.nasa.gov/. |
| |    3.  Comparison with Penman-Monteith-based calculus of evapotranspiration using on-site measurements (in situ & remote sensing). |
| |    4.  Comparison with values of wheat water dynamics from literature. |
| 2. If I understand correctly, $m_g$ used in Figure 2 is derived from L-band retrieved VOD. While lines 130 through 132 mention that VWC was measured using destructive sampling during the study, there is no mention of sampled values being used in the processing workflow to derive later values outside of the comparison in Figure 10. Figures 13 and 14, therefore, appear to compare variables that are | Role of in situ measurements for $m_g$:

 In situ measured $VWC$ was used to calculate in situ $m_g$. The details are described in Meyer et al., 2019 and read as follows:

 "Finally, to be able to compare our retrievals of $m_g$ with a reference dataset, the in situ $VWC$ was converted to $m_g$ by calculating first the dry matter fraction ($m_d$) as defined by Mätzler, 1994 (i.e., $m_d$= *dry mass/ fresh mass*) and subtracting it afterwards from 1 (i.e., $m_g$= 1 - $m_d$). This calculated $m_g$ will be called in situ measured $m_g$ in our study."

 We will update the manuscript by including a description on how in situ $m_g$-values were calculated.

 These in situ $m_g$-values are used in Figure 10 to be compared against L-band radiometer-derived $m_g$-values. Both datasets are independent from each other and serve as a first validation effort. We will clarify this in the |

| both derived from L-band measurements, which results in a circular comparison and leaves the method unvalidated. | updated version of the manuscript.

In Figure 2 the different variables are not assigned to certain acquisition techniques (in situ or remote sensing). Figure 2 introduces the general work flow to arrive from storage components to water fluxes. In order to make it more informative, we will update it by using different colors to indicate L-band radiometry-derived (green color), in-situ-derived (gray color) and jointly-derived variables (blue color).

[Figure]

Figure 2: Processing workflow for estimation of soil, vegetation and atmosphere water potentials ($SMP$= Soil Matric Potential, $VWP$ = Vegetation Water Potential, $VPD$ = Vapor Pressure Deficit) and water fluxes ($PWU$ = Plant Water Uptake, $TR$ = Transpiration Rate) from storage variables ($\theta$ = Soil Moisture, $m_g$ = Vegetation Water Content (gravimetric), $RH$ = Relative Air Humidity); Green variables are derived from radiometer observations, while gray ones are calculated from in situ measurements; Blue variables are derived jointly from radiometer and in situ observations.

Finally, Figures 13 and 14 show estimates of plant water uptake and transpiration rate. They are jointly estimated from a combination of in situ measurements and L-band radiometry. |
| Without comparison to values derived from sampled VWC, the statement on line 569 that 'the presented results indicate the unique potential of using passive microwave observations with on-site information of soil and atmosphere to estimate seasonal water dynamics' remains | We will change the statement and clarify that in situ measured $VWC$ was used to calculate in situ $m_g$. The details are described in Meyer et al., 2019 and read as follows:
"Finally, to be able to compare our retrievals of $m_g$ with a reference dataset, the in situ $VWC$ was converted to $m_g$ by calculating first the dry matter fraction ($m_d$) as defined by Mätzler, 1994 (i.e., $m_d$= *dry mass/ fresh mass*) and subtracting it afterwards from 1 (i.e., $m_g$= 1 - $m_d$). This calculated $m_g$ will be called in situ measured $m_g$ in our study."
We will update the manuscript by including these details, especially how in situ $m_g$-values were calculated and used in our study. |

| | |
|---|---|
| unjustified and is based upon both target variables derived from L-band measurements that are 'overall concurrent and similar in trend' to their like derived counterparts. | |
| How, if at all, in-situ destructive measurements of VWC were used in the study. | The details about the on-site and in situ measurements are provided in Meyer et al., 2018.
 In situ measured $VWC$ was used to calculate in situ $m_g$. The details are described in Meyer et al., 2019. From the reviewer comments, we realize this is an important point that needs to be further elaborated and clarified in the manuscript. We will update the manuscript accordingly. |
| If in-situ measurements were used, provide a more rigorous validation and comparison to L-band based results, instead of vague sentences such as on line 550 'VWP seems to be appropriate and fitting …'. | We will change the statement in line 550 to be more specific: "Nonetheless, $VWP$ as a radiometer-based potential estimate shows considerable similarity in temporal dynamics to the on-site measurement-derived potentials of soil ($SMP$) and atmosphere ($VPD$)"
 Although the in situ data availability is limited for this concept-based path finder research, we will update the manuscript to include quantitative measures from comparison to in-situ data when possible.
 In this study, in situ -based gravimetric water content $m_g$ is available and shown in Figure 10 together with its radiometer-based counterparts. However, validation using both (from in situ & from radiometry) was already done in Meyer et al., 2019 leading to a correlation of R²=0.89. |
| **Specific Comments** | |
| Soil moisture measurements are only at 5cm and 30cm, however wheat root zone can go to 100cm (as mentioned on line 279). Additional justification is required to state how 5 and 30 cm is sufficient to capture seasonal water dynamics. This would presumably affect Soil Matric Potential and PWU estimates. | In situ soil moisture measurements were solely available at 5 cm and 30 cm depth during the growing season in 2017. Both measurements are included in the analysis and fully reported in the manuscript. Unfortunately, soil moisture below 30 cm depth and rooting depth of the wheat plants were not measured in situ. The root zone until 100 cm depth was adopted from literature.
 Interestingly, White et al. in (2015) showed in the Figure below that for winter wheat in 17 experiment, the soil depths of 10cm and 30cm (upper most two boxes) exhibited a median of the root length density (RLD) above the critical RLD of 1 cm cm$^{-3}$ for wheat. |

[Figure]

**Fig. 1.** Mean RLD (root length density; filled circles, full line) to 100 cm depth for winter wheat in 17 experiments across the UK from 2007 to 2013, compared with published reference values [from Gregory *et al*. (1978b) and Barraclough *et al*. (1989, 1991); open circles, dashed line]. The cRLD of 1 cm cm⁻³ for wheat is shown (dotted). The box and whisker plots at each soil depth show the median (mid-line), interquartile range (boxes), and the minimum and maximum ('whiskers').

Nonetheless, rooting behavior and resulting water uptake might be very much site dependent. Thus, the representativeness of the results in White et al., 2015 for the case in Selhausen might be quite limited.

The reviewer comment made us realize, it is important to acknowledge this potentially limiting aspect for $SMP$ and follow-on parameters ($PWU$) estimation. We will include a discussion on this in the updated version of the manuscript.

In addition, we could have access to soil moisture (TDR) and soil matric potential ($SMP$) measurements from two rhizotron facilities next to the test field (facility 1 at 100 m distance to radiometer and facility 2 at 80 m distance to radiometer). The datasets are available from the responsible rhizotron-operator Prof. Dr. Andrea Schnepf, a direct and well-known colleague of Prof. Jonard (co-author). Although the relatively short distance to the radiometer should not lead to large differences in soil characteristics (e.g. texture, bulk density), this needs to be confirmed.
The advantage of using this new data would be the availability of $SMP$ and soil moisture at an hourly temporal resolution at three different plots and in six different depths (10, 20, 40, 60. 80, 120 cm). This may allow for a more detailed estimation of $PWU$ from 10 cm to 120 cm depth. We plan to explore the feasibility of this option and update the manuscript accordingly.

| | |
|---|---|
| Figure 11 and related discussion: Comparison of | The reason for presenting Figure 11 and including the statement at line 420 (see Figure and statement below) is to show that VOD carries |

| | |
|---|---|
| $RWC_{,season,\ VOD}$ and $RWC_{season,mg}$ seems to be superfluous and does not add to the paper. A statement on the shortcomings of directly calculating RWC from VOD (e.g. because plant biomass changes) would suffice. | influences from vegetation water content AND vegetation biomass & structure.

Hence, we want to convey the message, especially to the readers with interest in vegetation water content estimation with remotely sensed $VOD$ that $RWC_{Season,VOD}$, directly calculated with $VOD$ from (9) carries a biomass imprint (gray curve in Figure 11), while $RWC_{Season,mg}$ does not, because $m_g$ was extracted from $VOD$ before $RWC$-calculus. We believe it is relevant to stress this fact, since $VOD$ is increasingly being used as a direct indicator of either biomass or vegetation water content depending on the study focus (biomass: Malon et al., 2020; Rodriguez-Fernandez et al., 2018; Tian et al., 2016; vegetation water content: Xu et al., 2021; Holtzman et al., 2021). Figure 11 and associated text helps us convey this 'caution' message.

Statement at line 420:
"However, in periods of constant biomass, meaning times where only the water content in the plants would change, $RWC_{Season}$ could be directly estimated from $VOD$ (Rao et al., 2019; Holtzman et al., 2020)."

Figure 11:

[Figure]

Figure 11: Seasonal Relative Water Content ($RWC_{Season}$) [%] calculated in (2) with radiometer-derived $m_g$ (green circles) along growing season of 2017 in days of year (DOY) at the winter wheat field in Selhausen, Germany. The gray circles indicate $RWC_{Season}$ calculated directly with the radiometer-derived vegetation optical depth ($VOD$) according to (9). |
| Figure 9 and related discussion: Figure 9 does not add to the paper. That soil permittivity varies with precipitation impulse is a given and neither permittivity nor Soil Matric Potential (SMP) are derived from L-band in this study. SMP as plotted in Figure 12 alongside | We will review section 4.1 (including Figure 9 and related text) on "water status in the soil" in order to update and shorten the content discarding redundant or trivial statements. |

| | |
|---|---|
| Vegetation Water Potential is sufficient. | |
| Lines 616-617: It is stated that wind speed can be remotely sensed by radar/scatterometers and radiometers. Please provide references for how to derive wind speed on land from these instruments. | We will revise the text paragraph and cancel the statement about satellite-based (radar, radiometer) sensed wind speed estimation, as retrievals are almost exclusively conducted over water and not over land. Land heterogeneity does not allow to easily isolate a clear wind-only signal contribution. Many thanks for pointing this out. |
| Lines 461-462: Please provide a reference and expand on the meaning of the statement 'Due to the onset of senescence … water availability is not the limiting factor any more' | In the late wheat development stages (onset of senescence), the water supply of the drying plants degrades in importance, as the fruit (grains) needs to ripen, meaning to decrease its content of liquid in the grains (Steduto et al.,2012; Sarto et al., 2017). We will further elaborate this point and include references. |
| **Technical Corrections** | |
| Multiple grammatical errors in this paper | We will correct the grammatical errors. |
| Line 84: 'microwave remote sensing techniques should be capable to obtain …' | We will revise this. |
| Line 265: 'Van den Honert in 1948 was one of the first realizing and showing …' | We will revise this. |
| Line 657: 'We advocate in future a fully remote sensing-based, wide area (up to global) SPAS assessment can be a major achievement …' as well as several typos. | We will revise this. |
| This paper would benefit from a thorough review by a copy editor. | We will conduct a thorough review. |
| References: Holtzman, Nataniel M., et al. "L-band vegetation optical depth as an indicator of plant water potential in a temperate deciduous forest stand." Biogeosciences 18.2 (2021): 739-753  Mätzler, C. Microwave (1–100 GHz) dielectric model of leaves. *IEEE Trans. Geosci. Remote Sens., 32*, 947–949, 1994. | |

Meyer, T., Weihermüller, L., Vereecken, H., andJonard, F.: Vegetation Optical Depth and Soil Moisture Retrieved from L-Band Radiometry over the Growth Cycle of a Winter Wheat, Remote Sensing, 10(10), 1637, 2018

Meyer, T., Jagdhuber, T., Piles, M., Fink, A., Grant, J., Vereecken, H., and Jonard, F.: Estimating Gravimetric Water Content of a Winter Wheat Field from L-Band Vegetation Optical Depth. Remote Sensing, Remote Sensing11(20), 2353, 2019.

Mialon, Arnaud, et al. "Evaluation of the Sensitivity of SMOS L-VOD to Forest Above-Ground Biomass at Global Scale." Remote Sensing 12.9 (2020): 1450.

Rodríguez-Fernández, Nemesio J., et al. "An evaluation of SMOS L-band vegetation optical depth (L-VOD) data sets: high sensitivity of L-VOD to above-ground biomass in Africa." Biogeosciences 15.14 (2018): 4627-4645.

Sarto, M. V. M., Sarto, J. R. W., Rampim, L., Rosset, J. S., Bassegio, D., da Costa, P. F., & Inagaki, A. M. (2017). Wheat phenology and yield under drought: a review. *Australian Journal of Crop Science*, *11*(8), 941.

Steduto, P., Hsiao, T. C., Fereres, E., & Raes, D. (2012). Crop yield response to water (Vol. 1028). Rome: Food and Agriculture Organization of the United Nations.

Tian, Feng, et al. "Remote sensing of vegetation dynamics in drylands: Evaluating vegetation optical depth (VOD) using AVHRR NDVI and in situ green biomass data over West African Sahel." Remote Sensing of Environment 177 (2016): 265-276.

White, C. A., Sylvester-Bradley, R., & Berry, P. M. (2015). Root length densities of UK wheat and oilseed rape crops with implications for water capture and yield. Journal of Experimental Botany, 66(8), 2293-2303.

Xu, Xiangtao, et al. "Leaf surface water, not plant water stress, drives diurnal variation in tropical forest canopy water content." New Phytologist (2021).

---

## Author Response (AR1)

Dear Editors,

We are pleased to re-submit the reviewed manuscript entitled now "Towards Estimation of Seasonal Water Dynamics of Winter Wheat from Ground-Based L-Band Radiometry: A Concept Study" for publication in the inter-journal special issue of the EGU journals Biogeosciences and HESS about "Microwave remote sensing for improved understanding of vegetation–water interactions". We strongly believe the updated manuscript is appropriate for publication in this special issue.

In the updated manuscript we have addressed all points of the reviewers according to the point-by-point answers to the reviewer comments, which were approved by the editors for implementation. We also re-submitted a color-coded version of the answers-to-reviewer documents, where comments & answers in green text color are fully addressed and implemented as described in the updated manuscript. Comments and answers in black text color are implemented specifically after extending the data analysis and thorough review. For each of the few black-colored comments and answers, an additional explanation was added describing in detail the individual implementations and changes. A short report of the investigated major point and the subsequent changes in the manuscript are given hereafter:

We considered different approaches with the aim of including an initial assessment of our estimated water potential and water dynamics (plant water uptake $PWU$, transpiration rate $TR$) with independently measured/derived entities of these variables in the revised version of the manuscript. To this end, we investigated the following options:

- Comparison with space-borne $VOD$ from radiometer missions (SMAP MT-DCA product):
  We assessed the MT-DCA $VOD$-product of the SMAP mission for the region observed by the satellite in its native resolution (kilometer-scale) containing the test site (meter-scale). Our analyses showed that the $VOD$-values of both sources could not be compared in a reasonable and fair manner, due to the distinct spatial representativity of the measurements, i.e. the mismatch of the coarse spatial resolution of space-borne radiometers and the very high resolution of the field-based radiometer. As demonstrated by our colleague Thomas Meyer and co-authors in 2018, the field-based radiometer measurements and the retrieved $VOD$ show a distinct polarization dependence at the small scale due to the vertical orientation of the winter wheat stalks. However, this orientation effect is not affecting space-borne $VOD$ retrievals due to the large size of the resolution cells containing rather many land cover types of different shape and orientation.

  Meyer, T., Weihermüller, L., Vereecken, H., and Jonard, F.: Vegetation Optical Depth and Soil Moisture Retrieved from L-Band Radiometry over the Growth Cycle of a Winter Wheat, Remote Sensing, 10(10), 1637, 2018.

- Comparison with evapotranspiration data from the remote sensing-based ECOSTRESS mission (starting from 2018): https://ecostress.jpl.nasa.gov/:
  Three years of ECOSTRESS data (2019, 2020 & 2021) were extracted over the test site with a spatial resolution of 70 meters and were analyzed in detail. A range of ECOSTRESS-derived transpiration rates was found to compare well to our estimated $TR$. These results were included in the updated manuscript as a new Figure (15) and in several text paragraphs which were added to different sections of the manuscript (from abstract to conclusions).

- Comparison with Penman-Monteith-based calculus of evapotranspiration using on-site measurements:

  We used the FAO-based version of the Penman-Monteith equation (Allen et al., 1998) to estimate evapotranspiration ($ET$) values from on-site measurements (including temperature, radiation and wind speed). However, two problems arose with this calculus which disqualified this approach for an independent comparison: 1. The input data is also used, at least partially, in the proposed approach of the manuscript. 2. The disentanglement of evaporation and transpiration is needed, but complicated to conduct rigorously with the existing in situ data.

- Comparison with values of wheat water dynamics from literature

  We included several literature sources in the results section as reference of the value ranges of water dynamics ($PWU$ and $TR$) of winter wheat reported in previous studies. These studies support that our retrieved values are within realistic ranges.

- Inclusion of soil matric potential data from a rhizotron facility under corn vegetation:

  We had access to soil matric potential ($SMP$) measurements from a rhizotron facility (80 m distance to the ground-based radiometer placement) under a corn field close to the Selhausen (winter wheat) test field. The datasets were made partially available by the site operator (Prof. Schnepf, FZ Jülich). This $SMP$ data comes nominally at an hourly temporal resolution (April-August 2017) at three different locations and in six different depths (10, 20, 40, 60. 80, 120 cm). We explored the feasibility for a more detailed estimation of $PWU$ from 10 cm to 120 cm depth and found that many of the ground-based measurements taken at different depths and different times showed unrealistic to non-physical values. Hence, the dataset needed further refinement and quality control. In its present status it did not qualify for inclusion in the manuscript.

In addition, we have contacted the copy-editing office of the Biogeosciences journal and made arrangements to further improve our manuscript with their professional support, even after all our best efforts (being non-native speakers) to correct grammatical and wording errors in the updated version.

The updated manuscript has been approved by all authors. All authors are free of competing interests.

Thank you for your consideration. We look forward to hearing from you.

Best regards,

Thomas Jagdhuber, and all co-authors: François Jonard, Anke Fluhrer, David Chaparro, Martin J. Baur, Thomas Meyer and María Piles

**Towards Estimation of Seasonal Water Dynamics of Winter Wheat from Ground-Based L-Band Radiometry (Manuscript # BG-2021-71)**

| Comments | Responses/Actions |
|---|---|
| In this paper the authors seek to show that L-band radiometry can improve water dynamics estimation based on the Soil-Plant-Atmosphere System (SPAS). The methodology presented in the paper is relevant to the special issue and current L-band missions such as SMAP, and builds upon previous L-band research in Vegetation Optical Depth (VOD). While the method utilizing L-band radiometry and existing physical models to estimate wheat water dynamics is described in some detail, I have two major concerns: | Many thanks for confirming the relevance of the manuscript for the special issue. According to the reviewer comments, we worked on all raised issues with special focus on the two major comments:
▪ Validation of plant water dynamics
▪ Role of in situ measurements in the study. |
| 1. The field data used does not contain in situ measurements for the target variables Transpiration Rate (TR) and Plant Water Uptake (PWU), leaving the authors to discuss results in vague terms of what 'might be a first indication to the feasibility' of their method without any validation. In the absence of any strong validation data, the | Validation of plant water dynamics:
We agree that the presented estimates of transpiration rate ($TR$) and plant water uptake ($PWU$) were not tracked by a set of in situ measurements from the dedicated field laboratory experiment along the growing season of 2017 (Meyer et al., 2018). The experiment was originally not designed for this purpose, but for estimating vegetation optical depth ($VOD$) and gravimetric plant water content from L-band microwave radiometry at the field scale and for one entire growing season of 2017 (Meyer et al., 2018; Meyer et al., 2019).

One of the main innovations of the presented path finder research study is to elaborate a concept, foremost a methodology, to concert classical in situ measurements and $VOD$ for finding a way to arrive synergistically (in situ with microwave remote-sensing combined) at estimated $PWU$ and $TR$. This is a conceptual step forward in water dynamics estimation incorporating |

| | |
|---|---|
| paper could be a short communication rather than a full-length research paper. | $VOD$ in a field experimental setup leading to the projection of a future majorly remote sensing-based methodology to retrieve $PWU$ and $TR$.

We want to acknowledge this fact by adapting the title of our study and in this way preparing the reader for a concept-focused, rather than a validation-based, study. Suggestion for the new title is: "Towards Estimation of Seasonal Water Dynamics of Winter Wheat From Ground-Based L-Band Radiometry: A Concept Study".

Moreover, note we explicitly stress in the manuscript (in Sections: Introduction (l.38-39), and Conclusions (l.638-639, 650-652)) that its scientific contribution is on the concept and methodology of estimating water dynamics by retrieving L-band radiometer-derived estimates and orchestrating them with on-site measurements for arriving at estimates of plant water dynamics. To our knowledge, this is the first time that an end-to-end SPAS analysis is conducted using mechanistic models and input data available from in-situ and remote sensors.
We agree with the reviewer that this research study cannot serve as a validation study, meaning as a validation of an already existing methodology. Still, following the reviewer suggestion, we have considered different approaches with the aim of including an initial assessment of our estimated water potential and water dynamics ($PWU$, $TR$) with independently measured/derived entities of these variables in the revised version of the manuscript. To this end, we have investigated the following options:

    1.  Comparison with space-borne $VOD$ from radiometer missions (e.g. SMAP or SMOS).
        Extended explanation:
        We assessed the MT-DCA $VOD$-product of the SMAP mission for the wider region around the test site. Due to the coarse spatial resolution of space-borne radiometers (in terms of kilometers) in contrast to the very high resolution of the field-based radiometer (in terms of meters), the $VOD$-values of both sources could not be compared in a reasonable and fair manner due to the strong spatial scale gap. Especially, our colleague Thomas Meyer and co-authors demonstrated in 2018 that the field-based radiometer measurements and the retrieved $VOD$ show a distinct polarization dependence on the small scale due to the vertical orientation of the winter wheat stalks. This orientation effect is not affecting space-borne $VOD$ retrievals due to the large size of the resolution cells containing rather many land cover types of different shape and orientation.

Meyer, T., Weihermüller, L., Vereecken, H., andJonard, F.: Vegetation Optical Depth and Soil Moisture Retrieved from L-Band Radiometry over the Growth Cycle of a Winter Wheat, Remote Sensing, 10(10), 1637, 2018. |

| | 2. Comparison with evapotranspiration data from the remote sensing-based ECOSTRESS mission (starting from 2018): https://ecostress.jpl.nasa.gov/.
Extended explanation:
Due to the high spatial resolution of 70 meters, three years of ECOSTRESS data (2019, 2020 & 2021) were analyzed in detail and a range of transpiration rates was found fitting to the estimated $TR$ in the manuscript. A Figure (15) and several text paragraphs were added to different sections of the manuscript. Note that, due to the irregular distribution of samples along time from ECOSTRESS, a full time-series of satellite-derived $ET$ data was not available, and building a comparison of time dynamics between satellite and in situ estimates was not feasible.

3. Comparison with Penman-Monteith-based calculus of evapotranspiration using on-site measurements (in situ & remote sensing).
Extended explanation:
We used the FAO-based version of the Penman-Monteith equation (Allen et al., 1998) to estimate evapotranspiration ($ET$) values from on-site measurements (including temperature, radiation and wind speed). However, two problems arose with this calculus which disqualify this approach for an independent comparison: 1. The input data is also used, at least partly, in the proposed approach of the manuscript. 2. The disentanglement of evaporation and transpiration is needed, but complicated to conduct rigorously with existing in situ data.

4. Comparison with values of wheat water dynamics from literature.
Extended explanation:
We added several references exemplarily to present value ranges of water dynamics in literature for winter wheat. These references support that our retrieved value ranges appear realistic:

Cai, G., Vanderborght, J., Langensiepen, M., Schnepf, A., Hüging, H. and Vereecken, H., 2018. Root growth, water uptake, and sap flow of winter wheat in response to different soil water conditions. *Hydrology and Earth System Sciences*, *22*(4), pp.2449-2470.

Kang, S., Gu, B., Du, T. and Zhang, J., 2003. Crop coefficient and ratio of transpiration to evapotranspiration of winter wheat and maize in a semi-humid region. *Agricultural water management*, *59*(3), pp.239-254.

Zhang, T., Hou, M., Liu, L. and Tian, F., 2019. Estimation of transpiration and canopy cover of winter wheat under different fertilization levels using thermal infrared and visible imagery. *Computers and Electronics in Agriculture*, *165*, p.104936. |

| 2. If I understand correctly, m_g used in Figure 2 is derived from L-band retrieved VOD. While lines 130 through 132 mention that VWC was measured using destructive sampling during the study, there is no mention of sampled values being used in the processing workflow to derive later values outside of the comparison in Figure 10. Figures 13 and 14, therefore, appear to compare variables that are both derived from L-band measurements, which results in a circular comparison and leaves the method unvalidated. | Role of in situ measurements for $m_g$: |
|---|---|

Role of in situ measurements for $m_g$:

In situ measured $VWC$ was used to calculate in situ $m_g$. The details are described in Meyer et al., 2019 and read as follows:
"Finally, to be able to compare our retrievals of $m_g$ with a reference dataset, the in situ $VWC$ was converted to $m_g$ by calculating first the dry matter fraction ($m_d$) as defined by Mätzler, 1994 (i.e., $m_d$= *dry mass/ fresh mass*) and subtracting it afterwards from 1 (i.e., $m_g$= 1 - $m_d$). This calculated $m_g$ will be called in situ measured $m_g$ in our study."

We updated the manuscript detailing how in situ $m_g$-values were calculated.

These in situ $m_g$-values are compared against L-band radiometer-derived $m_g$-values in Figure 10. Both datasets are independent from each other and their comparison serves as a first validation effort. We clarified this in the updated version of the manuscript.

In Figure 2 the different variables are not assigned to certain acquisition techniques (in situ or remote sensing). Figure 2 introduces the general work flow to estimate water fluxes starting from storage components. In order to make it more informative, we updated it by using different colors to indicate L-band radiometry-derived (green color), in-situ-derived (gray color) and jointly-derived variables (blue color).

[Figure]

Figure 2: Processing workflow for estimation of soil, vegetation and atmosphere water potentials ($SMP$= Soil Matric Potential, $VWP$ = Vegetation Water Potential, $VPD$ = Vapor Pressure Deficit) and water fluxes ($PWU$ = Plant Water Uptake, $TR$ = Transpiration Rate) from storage variables ($\theta$ = Soil Moisture, $m_g$ = Vegetation Water Content (gravimetric), $RH$ = Relative Air Humidity); Green variables are derived from radiometer observations, while gray ones are calculated from in situ measurements; Red variables are derived jointly from radiometer and in situ observations.

| | |
|---|---|
| | Finally, Figures 13 and 14 show estimates of plant water uptake and transpiration rate. They are jointly estimated from a combination of in situ and remotely sensed data. |
| Without comparison to values derived from sampled VWC, the statement on line 569 that 'the presented results indicate the unique potential of using passive microwave observations with on-site information of soil and atmosphere to estimate seasonal water dynamics' remains unjustified and is based upon both target variables derived from L-band measurements that are 'overall concurrent and similar in trend' to their like derived counterparts. | We changed the statement and clarified that in situ measured $VWC$ was used to calculate in situ $m_g$. The details of the procedure are described in Meyer et al., 2019 and read as follows:
"Finally, to be able to compare our retrievals of $m_g$ with a reference dataset, the in situ $VWC$ was converted to $m_g$ by calculating first the dry matter fraction ($m_d$) as defined by Mätzler, 1994 (i.e., $m_d$= dry mass/ fresh mass) and subtracting it afterwards from 1 (i.e., $m_g$= 1 - $m_d$). This calculated $m_g$ will be called in situ measured $m_g$ in our study."
We updated the manuscript detailing how in situ $m_g$-values were calculated and used in our study. |
| How, if at all, in-situ destructive measurements of VWC were used in the study. | Full details about the on-site and in situ measurements are provided in Meyer et al., 2018.
In situ measured $VWC$ was used to calculate in situ $m_g$. This procedure is described in Meyer et al., 2019. From the reviewer comments, we realized this is an important point that needs to be further elaborated and clarified in the manuscript. We updated the manuscript accordingly. |
| If in-situ measurements were used, provide a more rigorous validation and comparison to L-band based results, instead of vague sentences such as on line 550 'VWP seems to be appropriate and fitting … '. | We changed the statement in line 550 to be more specific: "Nonetheless, $VWP$ as a radiometer-based potential estimate shows considerable similarity in temporal dynamics to the on-site measurement-derived potentials of soil ($SMP$) and atmosphere ($VPD$)"
Although the in situ data availability is limited for this concept-based path finder research, we updated the manuscript to include quantitative measurements from comparison to in-situ data:
Figure 10 now compares in situ -based gravimetric water content $m_g$ with its radiometer-based counterparts. Note that validation using both (from in situ & from radiometry) was already done in Meyer et al., 2019 leading to a correlation of R²=0.89. |
| **Specific Comments** | |
| Soil moisture measurements are only at 5cm and 30cm, however wheat root | In situ soil moisture measurements were solely available at 5 cm and 30 cm depth during the growing season in 2017. Both measurements are included in the analysis and fully reported in the manuscript. Unfortunately, soil moisture below 30 cm depth and rooting depth of the wheat plants were |

| | |
|---|---|
| zone can go to 100cm (as mentioned on line 279). Additional justification is required to state how 5 and 30 cm is sufficient to capture seasonal water dynamics. This would presumably affect Soil Matric Potential and PWU estimates. | not measured in situ. The root zone until 100 cm depth was adopted from literature. |

not measured in situ. The root zone until 100 cm depth was adopted from literature.

Interestingly, White et al. in (2015) showed in the Figure below that for winter wheat in 17 experiments, the soil depths of 10cm and 30cm (upper most two boxes) exhibited a median of the root length density (RLD) above the critical RLD of 1 cm cm$^{-3}$ for wheat.

[Figure]

**Fig. 1.** Mean RLD (root length density; filled circles, full line) to 100 cm depth for winter wheat in 17 experiments across the UK from 2007 to 2013, compared with published reference values [from Gregory et al. (1978b) and Barraclough et al. (1989, 1991); open circles, dashed line]. The cRLD of 1 cm cm$^{-3}$ for wheat is shown (dotted). The box and whisker plots at each soil depth show the median (mid-line), interquartile range (boxes), and the minimum and maximum ('whiskers').

Nonetheless, rooting behavior and resulting water uptake might be very much site dependent. Thus, the representativeness of the results in White et al., 2015 for the case in Selhausen might be quite limited.

The reviewer comment made us realize, it is important to acknowledge this potentially limiting aspect for $SMP$ and follow-on parameters ($PWU$) estimation.

Extended explanation:

We could have access to soil matric potential ($SMP$) measurements from a rhizotron facility (80 m distance to radiometer) under a corn field close to the Selhausen (winter wheat) test field. The datasets were made partially available by the site operator (Prof. Schnepf, FZ Jülich). This $SMP$ data comes nominally at an hourly temporal resolution (April-August 2017) at three different locations and in six different depths (10, 20, 40, 60. 80, 120 cm). We explored the feasibility for a more detailed estimation of $PWU$ from 10 cm to 120 cm depth and found that sensors in different depth and different times showed unrealistic to non-physical values. Hence, the dataset in its momentary status did not qualify for inclusion in the manuscript.

| | |
|---|---|
| Figure 11 and related discussion: Comparison of RWC,season, VOD and RWCseason,mg seems to be superfluous and does not add to the paper. A statement on the shortcomings of directly calculating RWC from VOD (e.g. because plant biomass changes) would suffice. | The reason for presenting Figure 11 and including the statement at line 420 (see Figure and statement below) is to show that $VOD$ carries influences from vegetation water content AND vegetation biomass & structure. Hence, we want to convey the message, especially to the readers with interest in vegetation water content estimation with remotely sensed $VOD$, that $RWC_{Season,VOD}$, directly calculated with $VOD$ from (9) carries a biomass imprint (gray curve in Figure 11), while $RWC_{Season,mg}$ does not, because $m_g$ was extracted from $VOD$ before $RWC$-calculus. We believe it is relevant to stress this fact, since $VOD$ is increasingly being used as a direct indicator of either biomass or vegetation water content depending on the study focus (biomass: Malon et al., 2020; Rodriguez-Fernandez et al., 2018; Tian et al., 2016; vegetation water content: Xu et al., 2021; Holtzman et al., 2021). Figure 11 and associated text helps us convey this 'caution' message.

Statement at line 420:
"However, in periods of constant biomass, meaning times when only the water content in the plants would change, $RWC_{Season}$ could be directly estimated from $VOD$ (Rao et al., 2019; Holtzman et al., 2020)."

Figure 11:

[Figure]

Figure 11: Seasonal Relative Water Content ($RWC_{Season}$) [%] calculated in (2) with radiometer-derived $m_g$ (green circles) along growing season of 2017 in days of year (DOY) at the winter wheat field in Selhausen, Germany. The gray circles indicate $RWC_{Season}$ calculated directly with the radiometer-derived vegetation optical depth ($VOD$) according to (9). |
| Figure 9 and related discussion: Figure 9 does not add to the paper. That soil permittivity varies with precipitation impulse is a given and neither permittivity nor Soil Matric Potential (SMP) are derived from L-band in this study. SMP as | We reviewed section 4.1 (including Figure 9 and related text) on "water status in the soil" in order to update and shorten the content discarding redundant or trivial statements.

Extended explanation:
We shortened the content, but kept Figure 9 and related explanations. Both are essential to understand, how the calculated (model-based) soil matric potential looks like in comparison to its driving input variable (soil permittivity). This concise overview provides the basis to better understand the later estimated plant water uptake, where (SMP) plays an essential role. |

| | |
|---|---|
| plotted in Figure 12 alongside Vegetation Water Potential is sufficient. | |
| Lines 616-617: It is stated that wind speed can be remotely sensed by radar/scatterometers and radiometers. Please provide references for how to derive wind speed on land from these instruments. | We revised these lines and cancelled the statement about satellite-based (radar, radiometer) sensed wind speed estimation, as retrievals are almost exclusively conducted over water and not over land. Land heterogeneity does not allow to easily isolate a clear wind-only signal contribution. Many thanks for pointing this out. |
| Lines 461-462: Please provide a reference and expand on the meaning of the statement 'Due to the onset of senescence … water availability is not the limiting factor any more' | In the late wheat development stages (onset of senescence), the water supply of the drying plants degrades in importance, as the fruit (grains) needs to ripen, meaning to decrease its content of liquid in the grains (Steduto et al.,2012; Sarto et al., 2017). In the revised version of the manuscript we further elaborated this point and included references. |
| **Technical Corrections** | |
| Multiple grammatical errors in this paper | We corrected for the grammatical errors together with a native-speaker colleague at DLR (group leader: M.-Eng. Mark Lützner). |
| Line 84: 'microwave remote sensing techniques should be capable to obtain …' | We revised this. |
| Line 265: 'Van den Honert in 1948 was one of the first realizing and showing …' | We revised this. |
| Line 657: 'We advocate in future a fully remote sensing-based, wide area (up to global) SPAS assessment can be a major achievement …' as well as several typos. | We revised this. |
| This paper would benefit from a thorough review by a copy editor. | We conducted a thorough review and made arrangements with the copy-editing team of the journal for further improvements and optimizations towards publication. |
| References: | |

Holtzman, Nataniel M., et al. "L-band vegetation optical depth as an indicator of plant water potential in a temperate deciduous forest stand." Biogeosciences 18.2 (2021): 739-753

Mätzler, C. Microwave (1–100 GHz) dielectric model of leaves. *IEEE Trans. Geosci. Remote Sens.*, *32*, 947–949, 1994.

Meyer, T., Weihermüller, L., Vereecken, H., andJonard, F.: Vegetation Optical Depth and Soil Moisture Retrieved from L-Band Radiometry over the Growth Cycle of a Winter Wheat, Remote Sensing, 10(10), 1637, 2018

Meyer, T., Jagdhuber, T., Piles, M., Fink, A., Grant, J., Vereecken, H., and Jonard, F.: Estimating Gravimetric Water Content of a Winter Wheat Field from L-Band Vegetation Optical Depth. Remote Sensing, Remote Sensing11(20), 2353, 2019.

Mialon, Arnaud, et al. "Evaluation of the Sensitivity of SMOS L-VOD to Forest Above-Ground Biomass at Global Scale." Remote Sensing 12.9 (2020): 1450.

Rodríguez-Fernández, Nemesio J., et al. "An evaluation of SMOS L-band vegetation optical depth (L-VOD) data sets: high sensitivity of L-VOD to above-ground biomass in Africa." Biogeosciences 15.14 (2018): 4627-4645.

Sarto, M. V. M., Sarto, J. R. W., Rampim, L., Rosset, J. S., Bassegio, D., da Costa, P. F., & Inagaki, A. M. (2017). Wheat phenology and yield under drought: a review. *Australian Journal of Crop Science*, *11*(8), 941.

Steduto, P., Hsiao, T. C., Fereres, E., & Raes, D. (2012). Crop yield response to water (Vol. 1028). Rome: Food and Agriculture Organization of the United Nations.

Tian, Feng, et al. "Remote sensing of vegetation dynamics in drylands: Evaluating vegetation optical depth (VOD) using AVHRR NDVI and in situ green biomass data over West African Sahel." Remote Sensing of Environment 177 (2016): 265-276.

White, C. A., Sylvester-Bradley, R., & Berry, P. M. (2015). Root length densities of UK wheat and oilseed rape crops with implications for water capture and yield. Journal of Experimental Botany, 66(8), 2293-2303.

Xu, Xiangtao, et al. "Leaf surface water, not plant water stress, drives diurnal variation in tropical forest canopy water content." New Phytologist (2021).

Response to Reviewer 2

**Towards Estimation of Seasonal Water Dynamics of Winter Wheat from Ground-Based L-Band Radiometry (Manuscript # BG-2021-71)**

| Comments | Responses/Actions |
|---|---|
| The manuscript presents a radio-meter based approach along with on-site measurements to estimate seasonal flux rates of water over a winter wheat field. The paper is well written, and the manuscript exhibits useful results. There are just a few aspects that need to be addressed before publication. First, the paper lacks other sources of data (e.g., satellite products and/or field laboratory data) to validate the employed empirical models and results. I'd suggest the authors at least include a few other observations to validate the overall utilized approach. Second, the paper requires some further modifications and/or clarifications in different parts. Based on these shortcomings, I recommend a minor revision. The authors should consider the following comments in their revision. | Dear Dr. Mostafa Momen, Many thanks for your encouraging and positive feedback, we are grateful you found this study useful and appropriate for this special issue and for the BG community. Concerning the aspects to address, we closely followed your advice and included other sources of data to compare and validate the employed empirical models and our obtained results. We also incorporated further modifications and clarifications in response to your suggested major and minor comments. |
| **Major Comments:** | |
| **Comment (1):** Line 105:

Q1: Why this particular plant has been selected for this study?

Q2: What are the characteristics that distinguish it from other plants?

Q3: How does selecting a plant and its hydraulic traits influence the final conclusions of the research?

The authors need to comment on these. | We will add several text paragraphs to the manuscript to address the three issues (Q1-Q3) raised here. Please find our answers as follows:

Q1: In 2017 winter wheat (*Triticum aestivum*) was grown in the crop rotation of the farmers at the Selhausen test site. We had access to this test field and the on-site measurements. The winter wheat at Selhausen grew well without too much care (no irrigation) or inputs (fertilizers). It was also not affected by diseases.
Moreover, this wheat monoculture has the advantage, that growth stages between individual plants are nearly completely synchronized and the canopy is very homogenous. The benefit here is that measurements of individual plants are very likely representative for all other plants and can be scaled to the whole canopy. In a more complex study design, a direct comparison between remote sensing and in situ measurements would be even more difficult. |

The described experimental work, together with first estimations of $VOD$ and the gravimetric water content of wheat ($m_g$) were the focus of previous research (Meyer et al., 2018; Meyer et al., 2019). We build on these results here and present a concept study for the estimation of water fluxes in the SPAS.

Most notably, a main motivation for analyzing wheat comes from its importance for food production being one of the major crop types cultivated around the globe. A concise infographic of the FAO (Food and Agriculture Organization of the United Nations) summarizes the main impact of wheat as one of the top commercial crops: http://www.fao.org/assets/infographics/FAO-Infographic-wheat-en.pdf

Q2: Key developmental stages of winter wheat (*Triticum aestivum*) are published by H. A. Bruns & L. I. Croy and indicate that this agricultural crop has a distinct phenological cycle in the yearly growing period. Detailed information on global distribution, botany, growth and physiology of winter wheat are presented in Curtis et al., 2002 (http://www.fao.org/3/y4011e/y4011e00.htm).

These distinct growth stages are particularly interesting, since they allow us investigating whether and to what extent L-band radiometry is a technology suitable to capture them. Taking the other extreme, a tree in a system where nearly no change in biomass happens, would not allow conducting these analyses.

We added a paragraph in the introduction motivating the focus of our study on winter wheat.

Q3: We used a field-based measurement setup (including several in situ and radiometer observations) that monitored a winter wheat (*Triticum aestivum*) field at the Selhausen (Germany) test site of the FZ Jülich for the 2017 growing season.

The final conclusions of our research study are bound to this setup as well as to the selected plant type (winter wheat), its characteristics and traits. A transferability to another setup as well as to another plant type and its individual traits may not be possible, or only partially.

This will depend on the similarity between setups as well as phenotypes, phenological status and traits of the plant subject to study compared to the one used in the present study.

| | |
|---|---|
| **Comment (2):** Equation 6: This model seems to have some empirical coefficients. Are these coefficients plant-type dependent?
In Lynn and Carlson (1990), Fig. 16 is depicted for corn.
How can that impact the used model in this study?
The authors need to comment on these. | We added a comment (text paragraph) on the revised manuscript specifying that the coefficients are empirically derived from a field study on corn, published in Lynn and Carlson (1990). We acknowledge that the relationship for wheat may be different than that of corn, but that we adopted it due to its simplicity (linear correlation with LAI) that allows us to dynamize the root-xylem resistance along the growing season, while keeping the amount of needed input variables constant. |
| **Comment (3):** Figure 11 and Line 420: Something that perplexes me is that the LAI is changing nonlinearly in the whole duration of the measurements according to Figure 6 implying that the total biomass is changing. If this is true, the comparison shown in this study does not seem valid (based on Line 420) and does not add anything to the paper. | Above ground biomass is shown together with other in situ measurements (LAI, vegetation height & vegetation water content) in Figure 1.
Figure 1 particularly illustrates how the total biomass changes along the growing season, as indicated by the reviewer.
However, the reason for presenting Figure 11 and including the statement at line 420 (see Figure and statement below) is to show that VOD carries influences from both vegetation water content and vegetation biomass & structure.
Hence, we want to convey the message, especially to the readers with interest in vegetation water content estimation by remotely sensed $VOD$, that $RWC_{Season,VOD}$, directly calculated with $VOD$ from (9) carries a biomass imprint (gray curve in Figure 11), while $RWC_{Season,mg}$ does not, because $m_g$ was extracted from $VOD$ before $RWC$-calculus. This is especially important, since $VOD$ is being increasingly used as a direct indicator of either biomass or vegetation water content depending on the study focus (biomass: Malon et al., 2020; Rodriguez-Fernandez et al., 2018; Tian et al., 2016; vegetation water content: Xu et al., 2021; Holtzman et al., 2021). This is in line with the study by Momen et al., 2017, where the reviewer investigated water and biomass effects on $VOD$. We added these references to the respective chapter in the manuscript.

Statement at line 420:
"However, in periods of constant biomass, meaning times where only the water content in the plants would change, $RWC_{Season}$ could be directly estimated from $VOD$ (Rao et al., 2019; Holtzman et al., 2020)." |

Figure 11:

[Figure]

Figure 11: Seasonal Relative Water Content ($RWC_{Season}$) [%] calculated in (2) with radiometer-derived $m_g$ (green circles) along growing season of 2017 in days of year (DOY) at the winter wheat field in Selhausen, Germany. The gray circles indicate $RWC_{Season}$ calculated directly with the radiometer-derived vegetation optical depth ($VOD$) according to (9).

| | |
|---|---|
| **Comment (4):** Figure 11, and a general comment: In general, one downside of the paper is that it does not compare the obtained results with other remote sensing products and/or laboratory analysis. This is significant for validation of the employed empirical models and results. In particular, authors can compare their derived RWCVOD (Fig. 11) or soil moisture with satellite products. Although the resolution might be different, it is expected to see generally a similar trend that can further validate the employed methods. | As suggested by the reviewer, we investigated the best way to compare and validate our obtained results with other available remote sensing products and/or laboratory analysis, despite the given inconsistencies in spatial and temporal resolutions of the different approaches and sensors. We compared our water potential estimates and the water dynamics ($PWU$, $TR$) with independently measured/derived entities of these variables, considering the following approaches:

1. **Comparison with space-borne VOD from radiometer missions (e.g. SMAP or SMOS).**
 **Extended explanation:**
We assessed the MT-DCA $VOD$-product of the SMAP mission for the wider region around the test site. Due to the coarse spatial resolution of space-borne radiometers (in terms of kilometers) in contrast to the very high resolution of the field-based radiometer (in terms of meters), the $VOD$-values of both sources could not be compared in a reasonable and fair manner due to the strong spatial scale gap. Especially, our colleague Thomas Meyer and co-authors demonstrated in 2018 that the field-based radiometer measurements and the retrieved $VOD$ show a distinct polarization dependence on the small scale due to the vertical orientation of the winter wheat stalks. This orientation effect is not affecting space-borne $VOD$ retrievals due to the large size of the |

resolution cells containing rather many land cover types of different shape and orientation.

Meyer, T., Weihermüller, L., Vereecken, H., andJonard, F.: Vegetation Optical Depth and Soil Moisture Retrieved from L-Band Radiometry over the Growth Cycle of a Winter Wheat, Remote Sensing, 10(10), 1637, 2018.

2. **Comparison with evapotranspiration data from the remote sensing-based ECOSTRESS mission (starting from 2018): https://ecostress.jpl. nasa. gov/.**
   Extended explanation:
   Due to the high spatial resolution of 70 meters, three years of ECOSTRESS data (2019, 2020 & 2021) were analyzed in detail and a range of transpiration rates was found fitting to the estimated $TR$ in the manuscript. A Figure (15) and several text paragraphs were added to different sections of the manuscript. Note that, due to the irregular distribution of samples along time from ECOSTRESS, a full time-series of satellite-derived $ET$ data was not available, and building a comparison of time dynamics between satellite and in situ estimates was not feasible.

3. **Comparison with Penman-Monteith-based calculus of evapotranspiration using on-site measurements (in situ & remote sensing).**
   Extended explanation:
   We used the FAO-based version of the Penman-Monteith equation (Allen et al., 1998) to estimate evapotranspiration ($ET$) values from on-site measurements (including temperature, radiation and wind speed). However, two problems arose with this calculus which disqualify this approach for an independent comparison: 1. The input data is also used, at least partly, in the proposed approach of the manuscript. 2. The disentanglement of evaporation and transpiration is needed, but complicated to conduct rigorously with existing in situ data.

4. **Comparison with values of wheat water dynamics from literature.**
   Extended explanation:

| | We added several references exemplarily to present value ranges of water dynamics in literature for winter wheat. These references support that our retrieved value ranges appear realistic: |
|---|---|
| | Cai, G., Vanderborght, J., Langensiepen, M., Schnepf, A., Hüging, H. and Vereecken, H., 2018. Root growth, water uptake, and sap flow of winter wheat in response to different soil water conditions. *Hydrology and Earth System Sciences*, *22*(4), pp.2449-2470. |
| | Kang, S., Gu, B., Du, T. and Zhang, J., 2003. Crop coefficient and ratio of transpiration to evapotranspiration of winter wheat and maize in a semi-humid region. *Agricultural water management*, *59*(3), pp.239-254. |
| | Zhang, T., Hou, M., Liu, L. and Tian, F., 2019. Estimation of transpiration and canopy cover of winter wheat under different fertilization levels using thermal infrared and visible imagery. *Computers and Electronics in Agriculture*, *165*, p.104936. |
| | However, we would like to note that this research study cannot contain a thorough validation study of the proposed concept. This will be subject of future research in which we plan to design dedicated measurement campaigns to validate and explore the practical application of the here introduced methodology for a wider range of vegetation types and climate conditions. |
| **Comment (5):** Line 560: How can such water flow estimations be done solely using remote sensing data? The authors could add some discussions on this and the deficiencies of remote sensing approaches to fully capture the water flow dynamics. I noticed that this has been somewhat discussed in lines 610-620 but more discussions focusing on the limitations and deficiencies of such remote sensing data would be useful especially for large-scale studies. | In order to discuss possible limitations and challenges on the use of large-scale remote sensing to fully capture water flow dynamics, we added the following text paragraph to the discussion section, connected to lines 610-620:
"…This would enable a wide-area (up to global) assessment of the SPAS in the end." However, this comes with the limitations in spatio-temporal as well as spectral coverage of remote sensing systems, no matter if active (e.g. lidar, radar) or passive (e.g. spectrometer, radiometer) systems are used. Moreover, it has to be acknowledged that remote sensing acquisitions do not purely sense one variable of the earth system, but normally a mixture of variables (e.g. combination of soil and vegetation variables). Hence, the quality of retrieved |

| | |
|---|---|
| | Earth system variables (e.g. soil or plant moisture), extracted from remotely sensed observations, depends directly on the sophistication of the signal-to-variable conversion. |
| | Moreover, L-band radiometry does not measure fluxes per se. Hence, we need valid estimates of the water reservoirs (soil moisture, plant moisture and relative humidity of atmosphere). Afterwards, we need performant estimates of the water potentials. In the end, we need to transit to the water fluxes, here the essential auxiliaries are the flow resistances of the soil, vegetation and atmosphere. These resistances are challenging to assess with remote sensing due to multi-factorial (inter-) dependencies. |
| | For these reasons, we advocate that in order to retrieve exact water flow dynamics, a plausible solution may come from the combination of earth system/vegetation growth models and high spatio-temporal resolution remote sensing data from multiple instruments. This multi-source approach could be key for applications needing quantitative estimates of water fluxes and will be the subject of further research. |
| **Minor Comments:** | |
| **Comment (1):** Line 125: How far is the climate station from the measurement site? | The used climate stations are located directly next to the test field (60 m from radiometer) and on a neighboring field (about 400 m from the radiometer). The second station is used only for assessing wind speed and net radiation as measurements of the closer station would be biased by interfering man-made infrastructure and measurement devices, which are located close by. |
| | We added an informative sentence to section 2 (test site and experimental data) to report this on-site setup. |
| **Comment (2):** Figure 1: How much is VWC correlated with LAI? | We calculated the Pearson's correlation coefficient R between the in situ measured vegetation water content ($VWC$) and leaf area index ($LAI$) along the growing season at the wheat field (see Figure 1 for individual data sets). It amounts to R=0.94. We added this result to the revised version of the manuscript. |
| **References**

Bruns, H. A., & Croy, L. I.: Key developmental stages of winter wheat, Triticum aestivum. Economic botany, 37(4), 410-417, 1983.

Curtis, B. C., Rajaram, S., & Gómez Macpherson, H.: Bread wheat: improvement and production. Food and Agriculture Organization of the United Nations (FAO), 2002. | |

Meyer, T., Weihermüller, L., Vereecken, H., andJonard, F.: Vegetation Optical Depth and Soil Moisture Retrieved from L-Band Radiometry over the Growth Cycle of a Winter Wheat, Remote Sensing, 10(10), 1637, 2018

Meyer, T., Jagdhuber, T., Piles, M., Fink, A., Grant, J., Vereecken, H., and Jonard, F.: Estimating Gravimetric Water Content of a Winter Wheat Field from L-Band Vegetation Optical Depth. Remote Sensing, Remote Sensing, 11(20), 2353, 2019.

Holtzman, Nataniel M., et al. "L-band vegetation optical depth as an indicator of plant water potential in a temperate deciduous forest stand." Biogeosciences 18.2 (2021): 739-753

Mialon, Arnaud, et al. "Evaluation of the Sensitivity of SMOS L-VOD to Forest Above-Ground Biomass at Global Scale." Remote Sensing 12.9 (2020): 1450.

Momen, Mostafa, et al. "Interacting effects of leaf water potential and biomass on vegetation optical depth." Journal of Geophysical Research: Biogeosciences 122.11 (2017): 3031-3046.

Rodríguez-Fernández, Nemesio J., et al. "An evaluation of SMOS L-band vegetation optical depth (L-VOD) data sets: high sensitivity of L-VOD to above-ground biomass in Africa." Biogeosciences 15.14 (2018): 4627-4645.

Tian, Feng, et al. "Remote sensing of vegetation dynamics in drylands: Evaluating vegetation optical depth (VOD) using AVHRR NDVI and in situ green biomass data over West African Sahel." Remote Sensing of Environment 177 (2016): 265-276.

Xu, Xiangtao, et al. "Leaf surface water, not plant water stress, drives diurnal variation in tropical forest canopy water content." New Phytologist (2021).

---

## Editor Decision (ED1)

Dear Dr. Jagdhuber and Colleagues,

Thank you for your detailed responses to both reviewers. In light of the reviewers' comments and your replies, the paper will be reconsidered after major revisions. Please apply all of the changes that you have discussed in your response to comments. Additionally, please pay close attention to the grammatical errors that were present throughout the document, as they make the technical aspects of the manuscript more difficult to review. The full responses to comments are listed below.

Kind regards,

Julia Green

**Towards Estimation of Seasonal Water Dynamics of Winter Wheat from Ground-Based L-Band Radiometry (Manuscript # BG-2021-71)**

| Comments | Responses/Actions |
|---|---|
| In this paper the authors seek to show that L-band radiometry can improve water dynamics estimation based on the Soil-Plant-Atmosphere System (SPAS). The methodology presented in the paper is relevant to the special issue and current L-band missions such as SMAP, and builds upon previous L-band research in Vegetation Optical Depth (VOD). While the method utilizing L-band radiometry and existing physical models to estimate wheat water dynamics is described in some detail, I have two major concerns: | Many thanks for confirming the relevance of the manuscript for the special issue. According to the reviewer comments, we will work on all raised issues with special focus on the two major comments:
 ▪ Validation of plant water dynamics
 ▪ Role of in situ measurements in the study. |
| 1. The field data used does not contain in situ measurements for the target variables Transpiration Rate (TR) and Plant Water Uptake (PWU), leaving the authors to discuss results in vague terms of what 'might be a first indication to the feasibility' of their method without any validation. In the absence of any strong validation data, the paper could be a short communication rather than a full-length research paper. | Validation of plant water dynamics:
 We agree that the presented estimates of transpiration rate ($TR$) and plant water uptake ($PWU$) were not tracked by a set of in situ measurements from the dedicated field laboratory experiment along the growing season of 2017 (Meyer et al., 2018). The experiment was originally not designed for this purpose, but for estimating vegetation optical depth ($VOD$) and gravimetric plant water content from L-band microwave radiometry at the field scale and for one entire growing season of 2017 (Meyer et al., 2018; Meyer et al., 2019).

 One of the main innovations of the presented path finder research study is to elaborate a concept, foremost a methodology, to concert classical in situ measurements and $VOD$ for finding a way to arrive synergistically (in situ with microwave remote-sensing combined) at estimated $PWU$ and $TR$. This is a conceptual step forward in water dynamics estimation incorporating $VOD$ in a field experimental setup leading to the projection of a future majorly remote sensing-based methodology to retrieve $PWU$ and $TR$. |

| | |
|---|---|
| | We want to acknowledge this fact by adapting the title of our study and in this way preparing the reader for a concept-focused, rather than a validation-based, study. Suggestion for the new title is: "Towards Estimation of Seasonal Water Dynamics of Winter Wheat From Ground-Based L-Band Radiometry: A Concept Study". |
| | Moreover, note we explicitly stress in the manuscript (in Sections: Introduction (l.38-39), and Conclusions (l.638-639, 650-652)) that its scientific contribution is on the concept and methodology of estimating water dynamics by retrieving L-band radiometer-derived estimates and orchestrating them with on-site measurements for arriving at estimates of plant water dynamics. To our knowledge, this is the first time that an end-to-end SPAS analysis is conducted using mechanistic models and input data available from in-situ and remote sensors. |
| | We agree with the reviewer that this research study cannot serve as a validation study, meaning as a validation of an already existing methodology. Still, following the reviewer suggestion, we will consider different approaches with the aim of including an initial assessment of our estimated water potential and water dynamics ($PWU$, $TR$) with independently measured/derived entities of these variables in the revised version of the manuscript. To this end, we will investigate the following options:
1. Comparison with space-borne $VOD$ from radiometer missions (e.g. SMAP or SMOS).
2. Comparison with evapotranspiration data from the remote sensing-based EcoSTRESS mission (starting from 2018): https://ecostress.jpl.nasa.gov/.
3. Comparison with Penman-Monteith-based calculus of evapotranspiration using on-site measurements (in situ & remote sensing).
4. Comparison with values of wheat water dynamics from literature. |
| 2. If I understand correctly, $m_g$ used in Figure 2 is derived from L-band retrieved VOD. While lines 130 through 132 mention that VWC was measured using destructive sampling during the study, there is no mention of sampled values being used in the processing workflow to derive later values outside of the comparison in Figure 10. Figures 13 and 14, therefore, appear to compare variables that are | Role of in situ measurements for $m_g$:
In situ measured $VWC$ was used to calculate in situ $m_g$. The details are described in Meyer et al., 2019 and read as follows:
"Finally, to be able to compare our retrievals of $m_g$ with a reference dataset, the in situ $VWC$ was converted to $m_g$ by calculating first the dry matter fraction ($m_d$) as defined by Mätzler, 1994 (i.e., $m_d$= *dry mass/ fresh mass*) and subtracting it afterwards from 1 (i.e., $m_g$= 1 - $m_d$). This calculated $m_g$ will be called in situ measured $m_g$ in our study."

We will update the manuscript by including a description on how in situ $m_g$-values were calculated.

These in situ $m_g$-values are used in Figure 10 to be compared against L-band radiometer-derived $m_g$-values. Both datasets are independent from each other and serve as a first validation effort. We will clarify this in the |

| | |
|---|---|
| both derived from L-band measurements, which results in a circular comparison and leaves the method unvalidated. | updated version of the manuscript.

In Figure 2 the different variables are not assigned to certain acquisition techniques (in situ or remote sensing). Figure 2 introduces the general work flow to arrive from storage components to water fluxes. In order to make it more informative, we will update it by using different colors to indicate L-band radiometry-derived (green color), in-situ-derived (gray color) and jointly-derived variables (blue color).

[Figure]

Figure 2: Processing workflow for estimation of soil, vegetation and atmosphere water potentials ($SMP$= Soil Matric Potential, $VWP$ = Vegetation Water Potential, $VPD$ = Vapor Pressure Deficit) and water fluxes ($PWU$ = Plant Water Uptake, $TR$ = Transpiration Rate) from storage variables ($\theta$ = Soil Moisture, $m_g$ = Vegetation Water Content (gravimetric), $RH$ = Relative Air Humidity); Green variables are derived from radiometer observations, while gray ones are calculated from in situ measurements; Blue variables are derived jointly from radiometer and in situ observations.

Finally, Figures 13 and 14 show estimates of plant water uptake and transpiration rate. They are jointly estimated from a combination of in situ measurements and L-band radiometry. |
| Without comparison to values derived from sampled VWC, the statement on line 569 that 'the presented results indicate the unique potential of using passive microwave observations with on-site information of soil and atmosphere to estimate seasonal water dynamics' remains | We will change the statement and clarify that in situ measured $VWC$ was used to calculate in situ $m_g$. The details are described in Meyer et al., 2019 and read as follows:
"Finally, to be able to compare our retrievals of $m_g$ with a reference dataset, the in situ $VWC$ was converted to $m_g$ by calculating first the dry matter fraction ($m_d$) as defined by Mätzler, 1994 (i.e., $m_d$= *dry mass/ fresh mass*) and subtracting it afterwards from 1 (i.e., $m_g$= 1 - $m_d$). This calculated $m_g$ will be called in situ measured $m_g$ in our study."
We will update the manuscript by including these details, especially how in situ $m_g$-values were calculated and used in our study. |

| | |
|---|---|
| unjustified and is based upon both target variables derived from L-band measurements that are 'overall concurrent and similar in trend' to their like derived counterparts. | |
| How, if at all, in-situ destructive measurements of VWC were used in the study. | The details about the on-site and in situ measurements are provided in Meyer et al., 2018.
In situ measured $VWC$ was used to calculate in situ $m_g$. The details are described in Meyer et al., 2019. From the reviewer comments, we realize this is an important point that needs to be further elaborated and clarified in the manuscript. We will update the manuscript accordingly. |
| If in-situ measurements were used, provide a more rigorous validation and comparison to L-band based results, instead of vague sentences such as on line 550 'VWP seems to be appropriate and fitting …'. | We will change the statement in line 550 to be more specific: "Nonetheless, $VWP$ as a radiometer-based potential estimate shows considerable similarity in temporal dynamics to the on-site measurement-derived potentials of soil ($SMP$) and atmosphere ($VPD$)"
Although the in situ data availability is limited for this concept-based path finder research, we will update the manuscript to include quantitative measures from comparison to in-situ data when possible.
In this study, in situ -based gravimetric water content $m_g$ is available and shown in Figure 10 together with its radiometer-based counterparts. However, validation using both (from in situ & from radiometry) was already done in Meyer et al., 2019 leading to a correlation of R²=0.89. |
| **Specific Comments** | |
| Soil moisture measurements are only at 5cm and 30cm, however wheat root zone can go to 100cm (as mentioned on line 279). Additional justification is required to state how 5 and 30 cm is sufficient to capture seasonal water dynamics. This would presumably affect Soil Matric Potential and PWU estimates. | In situ soil moisture measurements were solely available at 5 cm and 30 cm depth during the growing season in 2017. Both measurements are included in the analysis and fully reported in the manuscript. Unfortunately, soil moisture below 30 cm depth and rooting depth of the wheat plants were not measured in situ. The root zone until 100 cm depth was adopted from literature.
Interestingly, White et al. in (2015) showed in the Figure below that for winter wheat in 17 experiment, the soil depths of 10cm and 30cm (upper most two boxes) exhibited a median of the root length density (RLD) above the critical RLD of 1 cm cm$^{-3}$ for wheat. |

[Figure]

**Fig. 1.** Mean RLD (root length density; filled circles, full line) to 100 cm depth for winter wheat in 17 experiments across the UK from 2007 to 2013, compared with published reference values [from Gregory *et al.* (1978b) and Barraclough *et al.* (1989, 1991); open circles, dashed line]. The cRLD of 1 cm⁻³ for wheat is shown (dotted). The box and whisker plots at each soil depth show the median (mid-line), interquartile range (boxes), and the minimum and maximum ('whiskers').

Nonetheless, rooting behavior and resulting water uptake might be very much site dependent. Thus, the representativeness of the results in White et al., 2015 for the case in Selhausen might be quite limited.

The reviewer comment made us realize, it is important to acknowledge this potentially limiting aspect for $SMP$ and follow-on parameters ($PWU$) estimation. We will include a discussion on this in the updated version of the manuscript.

In addition, we could have access to soil moisture (TDR) and soil matric potential ($SMP$) measurements from two rhizotron facilities next to the test field (facility 1 at 100 m distance to radiometer and facility 2 at 80 m distance to radiometer). The datasets are available from the responsible rhizotron-operator Prof. Dr. Andrea Schnepf, a direct and well-known colleague of Prof. Jonard (co-author). Although the relatively short distance to the radiometer should not lead to large differences in soil characteristics (e.g. texture, bulk density), this needs to be confirmed.
The advantage of using this new data would be the availability of $SMP$ and soil moisture at an hourly temporal resolution at three different plots and in six different depths (10, 20, 40, 60. 80, 120 cm). This may allow for a more detailed estimation of $PWU$ from 10 cm to 120 cm depth. We plan to explore the feasibility of this option and update the manuscript accordingly.

| | |
|---|---|
| Figure 11 and related discussion: Comparison of | The reason for presenting Figure 11 and including the statement at line 420 (see Figure and statement below) is to show that VOD carries |

| RWC$_{,season, VOD}$ and RWC$_{season,mg}$ seems to be superfluous and does not add to the paper. A statement on the shortcomings of directly calculating RWC from VOD (e.g. because plant biomass changes) would suffice. | influences from vegetation water content AND vegetation biomass & structure.

Hence, we want to convey the message, especially to the readers with interest in vegetation water content estimation with remotely sensed $VOD$ that $RWC_{Season,VOD}$, directly calculated with $VOD$ from (9) carries a biomass imprint (gray curve in Figure 11), while $RWC_{Season,mg}$ does not, because $m_g$ was extracted from $VOD$ before $RWC$-calculus. We believe it is relevant to stress this fact, since $VOD$ is increasingly being used as a direct indicator of either biomass or vegetation water content depending on the study focus (biomass: Malon et al., 2020; Rodriguez-Fernandez et al., 2018; Tian et al., 2016; vegetation water content: Xu et al., 2021; Holtzman et al., 2021). Figure 11 and associated text helps us convey this 'caution' message.

Statement at line 420:
"However, in periods of constant biomass, meaning times where only the water content in the plants would change, $RWC_{Season}$ could be directly estimated from $VOD$ (Rao et al., 2019; Holtzman et al., 2020)."

Figure 11:

[Figure]

Figure 11: Seasonal Relative Water Content ($RWC_{Season}$) [%] calculated in (2) with radiometer-derived $m_g$ (green circles) along growing season of 2017 in days of year (DOY) at the winter wheat field in Selhausen, Germany. The gray circles indicate $RWC_{Season}$ calculated directly with the radiometer-derived vegetation optical depth ($VOD$) according to (9). |
| Figure 9 and related discussion: Figure 9 does not add to the paper. That soil permittivity varies with precipitation impulse is a given and neither permittivity nor Soil Matric Potential (SMP) are derived from L-band in this study. SMP as plotted in Figure 12 alongside | We will review section 4.1 (including Figure 9 and related text) on "water status in the soil" in order to update and shorten the content discarding redundant or trivial statements. |

| | |
|---|---|
| Vegetation Water Potential is sufficient. | |
| Lines 616-617: It is stated that wind speed can be remotely sensed by radar/scatterometers and radiometers. Please provide references for how to derive wind speed on land from these instruments. | We will revise the text paragraph and cancel the statement about satellite-based (radar, radiometer) sensed wind speed estimation, as retrievals are almost exclusively conducted over water and not over land. Land heterogeneity does not allow to easily isolate a clear wind-only signal contribution. Many thanks for pointing this out. |
| Lines 461-462: Please provide a reference and expand on the meaning of the statement 'Due to the onset of senescence … water availability is not the limiting factor any more' | In the late wheat development stages (onset of senescence), the water supply of the drying plants degrades in importance, as the fruit (grains) needs to ripen, meaning to decrease its content of liquid in the grains (Steduto et al.,2012; Sarto et al., 2017). We will further elaborate this point and include references. |
| **Technical Corrections** | |
| Multiple grammatical errors in this paper | We will correct the grammatical errors. |
| Line 84: 'microwave remote sensing techniques should be capable to obtain …' | We will revise this. |
| Line 265: 'Van den Honert in 1948 was one of the first realizing and showing …' | We will revise this. |
| Line 657: 'We advocate in future a fully remote sensing-based, wide area (up to global) SPAS assessment can be a major achievement …' as well as several typos. | We will revise this. |
| This paper would benefit from a thorough review by a copy editor. | We will conduct a thorough review. |
| References:

Holtzman, Nataniel M., et al. "L-band vegetation optical depth as an indicator of plant water potential in a temperate deciduous forest stand." Biogeosciences 18.2 (2021): 739-753

Mätzler, C. Microwave (1–100 GHz) dielectric model of leaves. *IEEE Trans. Geosci. Remote Sens., 32*, 947–949, 1994. | |

Meyer, T., Weihermüller, L., Vereecken, H., andJonard, F.: Vegetation Optical Depth and Soil Moisture Retrieved from L-Band Radiometry over the Growth Cycle of a Winter Wheat, Remote Sensing, 10(10), 1637, 2018

Meyer, T., Jagdhuber, T., Piles, M., Fink, A., Grant, J., Vereecken, H., and Jonard, F.: Estimating Gravimetric Water Content of a Winter Wheat Field from L-Band Vegetation Optical Depth. Remote Sensing, Remote Sensing11(20), 2353, 2019.

Mialon, Arnaud, et al. "Evaluation of the Sensitivity of SMOS L-VOD to Forest Above-Ground Biomass at Global Scale." Remote Sensing 12.9 (2020): 1450.

Rodríguez-Fernández, Nemesio J., et al. "An evaluation of SMOS L-band vegetation optical depth (L-VOD) data sets: high sensitivity of L-VOD to above-ground biomass in Africa." Biogeosciences 15.14 (2018): 4627-4645.

Sarto, M. V. M., Sarto, J. R. W., Rampim, L., Rosset, J. S., Bassegio, D., da Costa, P. F., & Inagaki, A. M. (2017). Wheat phenology and yield under drought: a review. *Australian Journal of Crop Science*, *11*(8), 941.

Steduto, P., Hsiao, T. C., Fereres, E., & Raes, D. (2012). Crop yield response to water (Vol. 1028). Rome: Food and Agriculture Organization of the United Nations.

Tian, Feng, et al. "Remote sensing of vegetation dynamics in drylands: Evaluating vegetation optical depth (VOD) using AVHRR NDVI and in situ green biomass data over West African Sahel." Remote Sensing of Environment 177 (2016): 265-276.

White, C. A., Sylvester-Bradley, R., & Berry, P. M. (2015). Root length densities of UK wheat and oilseed rape crops with implications for water capture and yield. Journal of Experimental Botany, 66(8), 2293-2303.

Xu, Xiangtao, et al. "Leaf surface water, not plant water stress, drives diurnal variation in tropical forest canopy water content." New Phytologist (2021).

**Towards Estimation of Seasonal Water Dynamics of Winter Wheat from Ground-Based L-Band Radiometry (Manuscript # BG-2021-71)**

| Comments | Responses/Actions |
|---|---|
| The manuscript presents a radio-meter based approach along with on-site measurements to estimate seasonal flux rates of water over a winter wheat field. The paper is well written, and the manuscript exhibits useful results. There are just a few aspects that need to be addressed before publication. First, the paper lacks other sources of data (e.g., satellite products and/or field laboratory data) to validate the employed empirical models and results. I'd suggest the authors at least include a few other observations to validate the overall utilized approach. Second, the paper requires some further modifications and/or clarifications in different parts. Based on these shortcomings, I recommend a minor revision. The authors should consider the following comments in their revision. | Dear Mostafa Momen, Many thanks for your encouraging and positive feedback, we are grateful you found this study useful and appropriate for this special issue and for the BG community. Concerning the aspects to address, we will closely follow your advice and include other sources of data to compare and validate the employed empirical models and results. Moreover, we will also incorporate further modifications and clarifications in response to your suggested major and minor comments. |
| **Major Comments:** | |
| **Comment (1):** Line 105:

Q1: Why this particular plant has been selected for this study?

Q2: What are the characteristics that distinguish it from other plants?

Q3: How does selecting a plant and its hydraulic traits influence the final conclusions of the research?

The authors need to comment on these. | We will add several text paragraphs to the manuscript to address the three issues (Q1-Q3) raised here. Please find our answers as follows:

Q1: In 2017 winter wheat (*Triticum aestivum*) was grown in the crop rotation of the farmers at the Selhausen test site. We have access to this test field and the on-site measurements. The winter wheat at Selhausen grew well without too much care (no irrigation) or inputs (fertilizers). It was also not affected by diseases.
Moreover, this wheat monoculture has the advantage, that growth stages between individual plants are nearly completely synchronized and the canopy is very homogenous. The benefit here is that measurements of individual plants are very likely representative for all other plants and can be scaled to the whole canopy. In a more complex study design, a direct comparison between remote sensing and in situ measurements would be even more difficult. |

The described experimental work, together with first estimations of $VOD$ and the gravimetric water content of wheat ($m_g$) were the focus of previous research (Meyer et al., 2018; Meyer et al., 2019). We build on these results here and present a concept study for the estimation of water fluxes in the SPAS.

Most notably, a main motivation for analyzing wheat comes from its importance for food production being one of the major crop types cultivated around the globe. A concise infographic of the FAO (Food and Agriculture Organization of the United Nations) summarizes the main impact of wheat as one of the top commercial crops:

http://www.fao.org/assets/infographics/FAO-Infographic-wheat-en.pdf

Q2: Key developmental stages of winter wheat (*Triticum aestivum*) are published by H. A. Bruns & L. I. Croy and indicate that this agricultural crop has a distinct phenological cycle in the yearly growing period. Detailed information on global distribution, botany, growth and physiology of winter wheat are presented in Curtis et al., 2002 (http://www.fao.org/3/y4011e/y4011e00.htm).

These distinct growth stages are particularly interesting, since they allow us investigating whether and to what extent L-band radiometry is a technology suitable to capture them. Taking the other extreme, a tree in a system where nearly no change in biomass happens, would not allow conducting these analyses.

We will add an introductory paragraph to the manuscript characterizing winter wheat as the investigated crop type of our study.

Q3: We used a field-based measurement setup (including several in situ and radiometer observations) that monitored a winter wheat (*Triticum aestivum*) field at the Selhausen (Germany) test site of the FZ Jülich for the 2017 growing season.

The final conclusions of our research study are bound to this setup as well as the selected plant type (winter wheat), its characteristics and traits. A transferability to another setup as well as to another plant type and its individual traits may not or only partially be possible.

This will depend on the similarity between setups as well as phenotypes, phenological status and traits of the plant subject to study compared to the one used in the

| | present study. |
|---|---|
| **Comment (2):** Equation 6: This model seems to have some empirical coefficients. Are these coefficients plant-type dependent? In Lynn and Carlson (1990), Fig. 16 is depicted for corn. How can that impact the used model in this study? The authors need to comment on these. | We will add a comment (text paragraph) on the revised manuscript specifying that the coefficients are empirically derived from a field study on corn, published in Lynn and Carlson (1990). We will acknowledge that the relationship for wheat may be different than that of corn, but that we adopted it due to its simplicity (linear correlation with LAI) that allows us to dynamize the root-xylem resistance along the growing season, while keeping the amount of needed input variables constant. |
| **Comment (3):** Figure 11 and Line 420: Something that perplexes me is that the LAI is changing nonlinearly in the whole duration of the measurements according to Figure 6 implying that the total biomass is changing. If this is true, the comparison shown in this study does not seem valid (based on Line 420) and does not add anything to the paper. | Above ground biomass is shown together with other in situ measurements (LAI, vegetation height & vegetation water content) in Figure 1. Figure 1 particularly illustrates how the total biomass changes across the growing season, as indicated by the reviewer. However, the reason for presenting Figure 11 and including the statement at line 420 (see Figure and statement below) is to show that VOD carries influences from both vegetation water content and vegetation biomass & structure. Hence, we want to convey the message, especially to the readers with interest in vegetation water content estimation by remotely sensed $VOD$, that $RWC_{Season,VOD}$, directly calculated with $VOD$ from (9) carries a biomass imprint (gray curve in Figure 11), while $RWC_{Season,mg}$ does not, because $m_g$ was extracted from $VOD$ before $RWC$-calculus. This is especially important, since $VOD$ is being increasingly used as a direct indicator of either biomass or vegetation water content depending on the study focus (biomass: Malon et al., 2020; Rodriguez-Fernandez et al., 2018; Tian et al., 2016; vegetation water content: Xu et al., 2021; Holtzman et al., 2021). This is in line with the study in Momen et al., 2017, where the reviewer investigated water and biomass effects on $VOD$. We will add these references to the respective chapter in the manuscript. Statement at line 420: "However, in periods of constant biomass, meaning times where only the water content in the plants would change, $RWC_{Season}$ could be directly estimated from $VOD$ (Rao et al., 2019; Holtzman et al., 2020)." |

Figure 11:

[Figure]

Figure 11: Seasonal Relative Water Content ($RWC_{Season}$) [%] calculated in (2) with radiometer-derived $m_g$ (green circles) along growing season of 2017 in days of year (DOY) at the winter wheat field in Selhausen, Germany. The gray circles indicate $RWC_{Season}$ calculated directly with the radiometer-derived vegetation optical depth ($VOD$) according to (9).

| **Comment (4):** Figure 11, and a general comment: In general, one downside of the paper is that it does not compare the obtained results with other remote sensing products and/or laboratory analysis. This is significant for validation of the employed empirical models and results. In particular, authors can compare their derived RWCVOD (Fig. 11) or soil moisture with satellite products. Although the resolution might be different, it is expected to see generally a similar trend that can further validate the employed methods. | As suggested by the reviewer, we will investigate the best way to compare and validate our obtained results with other available remote sensing products and/or laboratory analysis, despite the given inconsistencies in spatial and temporal resolutions of the different approaches and sensors. We plan to compare our water potential estimates and the water dynamics ($PWU$, $TR$) with independently measured/derived entities of these variables, considering the following approaches: |
|---|---|

1. Comparison with space-borne $VOD$ from radiometer missions (e.g. SMAP or SMOS).
2. Comparison with evapotranspiration data from the remote sensing-based EcoSTRESS mission (starting from 2018): https://ecostress.jpl.nasa.gov/.
3. Comparison with Penman-Monteith-based calculus of evapotranspiration using on-site measurements (in situ & remote sensing).
4. Comparison with values of wheat water dynamics from literature.

However, we would like to note that this research study cannot contain a thorough validation study of the proposed concept. This will be subject of future research in which we plan to design dedicated measurement campaigns to validate and explore the practical application of the here introduced methodology for a wider range of vegetation types and climate conditions.

| | |
|---|---|
| **Comment (5):** Line 560: How can such water flow estimations be done solely using remote sensing data?
The authors could add some discussions on this and the deficiencies of remote sensing approaches to fully capture the water flow dynamics. I noticed that this has been somewhat discussed in lines 610-620 but more discussions focusing on the limitations and deficiencies of such remote sensing data would be useful especially for large-scale studies. | In order to discuss possible limitations and challenges on the use of large-scale remote sensing for fully capturing the water flow dynamics, we will add the following text paragraph to the discussion section, connected to lines 610-620:
"…This would enable a wide-area (up to global) assessment of the SPAS in the end." However, this comes with the limitations in spatio-temporal as well as spectral coverage of remote sensing systems, no matter if active (e.g. lidar, radar) or passive (e.g. spectrometer, radiometer) systems are used. Moreover, it has to be acknowledged that remote sensing acquisitions do not purely sense one variable of the Earth system, but normally a mixture of variables (e.g. combination of soil and vegetation variables). Hence, the quality of retrieved Earth system variables (e.g. soil or plant moisture), extracted from remotely sensed observations, depends directly on the sophistication of the signal-to-variable conversion by the established retrieval algorithm.
Moreover, L-band radiometry does not measure fluxes per se. Hence, we need valid estimates of the water reservoirs (soil moisture, plant moisture and relative humidity of atmosphere). Afterwards, we need performant estimates of the water potentials. In the end, we need to transit to the water fluxes, here the essential auxiliaries are the flow resistances of the soil, vegetation and atmosphere. These resistances are challenging to assess with remote sensing due to multi-factorial (inter-)dependencies.
For these reasons, in order to retrieve exact water flow dynamics, the most plausible solution will probably come from the combination of Earth system/vegetation growth models and high spatio-temporal resolution remote sensing data from multiple instruments. This multi-source approach will be key for applications needing quantitative estimates of water fluxes and will be the subject of further research. |
| **Minor Comments:** | |
| **Comment (1):** Line 125: How far is the climate station from the measurement site? | The used climate stations are located directly next to the test field (60 m from radiometer) and on a neighboring field (about 400 m from the radiometer). The second station is used only for assessing wind speed and net radiation as measurements of the closer station would be biased by interfering man-made infrastructure and measurement devices, which are located close by.
We will add an informative sentence to the section 2 (test site and experimental data) to report this on-site |

| | setup. |
|---|---|
| **Comment (2):** Figure 1: How much is VWC correlated with LAI? | We calculated the Pearson's correlation coefficient R between the in situ measured vegetation water content ($VWC$) and leaf area index ($LAI$) along the growing season at the wheat field (see Figure 1 for individual data sets). It amounts to R=0.94. We will add a statement close to the description of Figure 1. |

**References**

Bruns, H. A., & Croy, L. I.: Key developmental stages of winter wheat, Triticum aestivum. Economic botany, 37(4), 410-417, 1983.

Curtis, B. C., Rajaram, S., & Gómez Macpherson, H.: Bread wheat: improvement and production. Food and Agriculture Organization of the United Nations (FAO), 2002.

Meyer, T., Weihermüller, L., Vereecken, H., andJonard, F.: Vegetation Optical Depth and Soil Moisture Retrieved from L-Band Radiometry over the Growth Cycle of a Winter Wheat, Remote Sensing, 10(10), 1637, 2018

Meyer, T., Jagdhuber, T., Piles, M., Fink, A., Grant, J., Vereecken, H., and Jonard, F.: Estimating Gravimetric Water Content of a Winter Wheat Field from L-Band Vegetation Optical Depth. Remote Sensing, Remote Sensing, 11(20), 2353, 2019.

Holtzman, Nataniel M., et al. "L-band vegetation optical depth as an indicator of plant water potential in a temperate deciduous forest stand." Biogeosciences 18.2 (2021): 739-753

Mialon, Arnaud, et al. "Evaluation of the Sensitivity of SMOS L-VOD to Forest Above-Ground Biomass at Global Scale." Remote Sensing 12.9 (2020): 1450.

Momen, Mostafa, et al. "Interacting effects of leaf water potential and biomass on vegetation optical depth." Journal of Geophysical Research: Biogeosciences 122.11 (2017): 3031-3046.

Rodríguez-Fernández, Nemesio J., et al. "An evaluation of SMOS L-band vegetation optical depth (L-VOD) data sets: high sensitivity of L-VOD to above-ground biomass in Africa." Biogeosciences 15.14 (2018): 4627-4645.

Tian, Feng, et al. "Remote sensing of vegetation dynamics in drylands: Evaluating vegetation optical depth (VOD) using AVHRR NDVI and in situ green biomass data over West African Sahel." Remote Sensing of Environment 177 (2016): 265-276.

Xu, Xiangtao, et al. "Leaf surface water, not plant water stress, drives diurnal variation in tropical forest canopy water content." New Phytologist (2021).

---

## Author Response (AR2)

Response to Reviewer 3

**Towards Estimation of Seasonal Water Dynamics of Winter Wheat from Ground-Based L-Band Radiometry: A Concept Study (Manuscript # BG-2021-71)**

| Comments | Responses/Actions |
|---|---|
| The topic is really interesting and fits the purpose of the journal. The objectives to derive the water uptake and TR are challenging. The manuscript is well structure, the method and the data are well described. There are some points that need to be addressed before considering this work for publication. | Many thanks for the reviewer's encouraging evaluation. We addressed all points, raised by the reviewer, in the following responses and changed the manuscript accordingly. |
| Generally, the discussion contains many sentences that can be part of the conclusion, which conclusion then seems like an abstract. The discussion needs to be re-worked. | We worked through the discussion and the conclusion sections and re-named, re-organized, re-wrote and specified the individual text paragraphs. Redundant text paragraphs were removed. Moreover, we specified a paragraph in the discussion section about uncertainty of key variables (flow resistances) in the retrieval algorithm to inform about their influence on $PWU$- & $TR$-estimation. |
| Plus, there are a lot of repetitive ideas (a few examples : lines 667 page 31 and 683 page 32 ; line 598 page 28 and lines 634 page 30) which make the discussion incomplete with vague comments. Another example is section lines 681-685 : « Our results show the potential of combining » These sentences sound general without precisions (to do what ? Our results ..where ; to existing results but which results ? Which parameters ? | We removed the repetitions in the discussion sections and re-formulated vague statements wherever found. As mentioned directly by the reviewer:
 ▪ We adapted and coordinated the paragraphs in lines 667 and 683 to keep their common idea, but removed the repetitive text parts.
 ▪ We cancelled the paragraph in line 634 on page 30 in the discussion section, as it is equivalent to the statement in lines 598 on page 28 of the results section.
 ▪ We removed the vague statement in lines 681-685.

 We also cancelled or moved the following statements:
 ▪ We cancelled the repetitive statement in lines 626-628.
 ▪ We cancelled a repetitive ECOSTRESS statement in lines 643-645.
 ▪ We also cancelled the repeated statement on limitations in line 657.
 ▪ We deleted the repetitive comment on the transpiration from top and bottom of leaves in lines 662-664.
 ▪ We cut out the repeated statement on wide-area retrieval in lines 672—673.
 ▪ We cancelled the re-occurring statement on the challenge to estimate resistances at remote sensing scales in lines 704-705. |

| | ▪ We moved the statements lines 710-718 (end of discussion section) to the conclusion section. |
|---|---|
| But most importantly, it lacks an analysis on the influence of each variables (uncertainties). The methods use many parameters and equations to compute the TR and PWU. Some of the parameters are difficult to obtain as mentioned in the manuscript , but one can not evaluate their importance on the final results. Is it worth having a precise value for these or not ? For instance paragraph lines 657-664, the discussion points out issues but there is no link with existing studies on the topic nor the influence of these parameters on the results. | We re-wrote and specified the paragraph in the discussion section (lines 657-664) to quantify the uncertainty of the $PWU$- & $TR$-estimation regarding the flow resistances.

For this, we first conducted an uncertainty analysis on the influence of the major input variables following table 2. We found that especially the flow resistances, necessary to calculate the water fluxes ($PWU$ & $TR$), are critical and not directly measurable or assessible at field-scale, neither by remote sensing nor by in situ measurements. Hence, we focused on the uncertainty of the resistances on flux estimation. We changed the resistance values by $\pm1\%$, $\pm5\%$, $\pm10\%$, $\pm15\%$ and $\pm20\%$ of value in order to understand these uncertainty effects on the flux estimates. In Figs. R1 to R3 below, the uncertainty-induced change ($\pm10\%$ & $\pm20\%$) is shown exemplary for plant water uptake in 5 [cm] and 30 [cm] depth as well as for the transpiration rate.

[Figure]

Fig. R1: Change in plant water uptake in 5 [cm] depth due to uncertainty ($\pm10\%$ & $\pm20\%$) in resistance values compared to original values.

[Figure]

Fig. R2. Change in plant water uptake in 30 [cm] depth due to uncertainty ($\pm10\%$ & $\pm20\%$) in resistance values compared to original values. |

[Figure]

Fig. R3. Change in transpiration rate due to uncertainty ($\pm$10% & $\pm$20%) in resistance values compared to original values.

Figs. R1 to R3 reveal that the seasonal trend for $PWU$ and $TR$ is not changed when the knowledge about the resistance values is more and more uncertain. Moreover, the larger the $PWU$- and $TR$-values, the stronger the uncertainty affects the estimates. This led to uncertainty-induced maximum changes of $PWU$s and $TR$ of $3.5 \cdot 10^{-4}$ $[mm/s]$ (30 $[cm]$ depth), $4.2 \cdot 10^{-4}$ $[mm/s]$ (5 $[cm]$ depth) and $5.4 \cdot 10^{-5}$ $[mm/s]$, respectively, when including 20% uncertainty.

| | |
|---|---|
| Validation of PWU is difficult but there is nothing in the manuscript to evaluate the significance of the derived results. | We added a statement concerning the difficulty to evaluate the significance of the achieved results in the end of the conclusion section:
 "A first comparison of $TR$ estimates from the presented field-based approach and from the space-borne ECOSTRESS mission indicates similar value ranges (same order of magnitude, mainly between zero and $1.0 \cdot 10^{-4}$ $[mm/s]$). However, the validation of absolute accuracies needs to be tackled in future studies with dedicated in situ measurements of water dynamics (potentials & flux rates). This is especially true for the $PWU$ estimates where no comparison or validation dataset was available in contrast to the $TR$ case." |
| **Minor Corrections** | |
| Page 673/674, page 31, « Algorithms …. » what is the point of this sentence ? | Many thanks. This sentence was misleading. We cancelled it. |
| Figure 11 : the 2 derived RWC are delayed by 15 days. What can explain the differences ? Which one is used in equation (3) (line 480) | 1. The two derived RWC curves are temporally delayed, where $RWC_{Season}$ from $m_g$ leads by 2-3 weeks compared to $RWC_{Season}$ from $VOD$. One explanation could be that $VOD$ also follows the biomass dynamics and not only the water dynamics, whereas $m_g$ is only sensitive to the water dynamics. The water content in the plants peak around DOY 140 (see Figure 10), whereas the biomass peaks around DOY 155. This may explain the temporal delay.

 2. We added a sentence to the paragraph to clarify that $RWC_{Season}$ from $m_g$ is used in the analyses: |

| | |
|---|---|
| | "From $RWC_{Season}$, the $VWP$ of the winter wheat can be retrieved using (3) and assuming different change rates of $VWP$ according to $RWC_{Season}$-dynamics (cf. Fig. 3). In the following the $RWC_{Season}$ from $m_g$ is used in the analyses. Figure 12 shows in green color the $VWP$ using intermediate change rate and in a gray area between dashed curves the behavior of the $VWP$ according to the different assumed change rates (blue color: slow change rate & red color: rapid change rate)." |
| figure 14 top : There is no x-axis ; what is the difference with figure 4 top. Generally the paper contains a lot of figures, it is recommanded | The x-axis is the same for both plots in figure 14, but only plotted once for the bottom plot. We added a sentence to the caption of figure 14 for clarification: "The x-axis labels are the same as for the bottom plot." |
| Figure 2 : all acronyms are defined but RWC | Thank you very much. We added the definition of $RWC$ to Figure 2 which is the relative water content of vegetation. |